# Salinity control on Na incorporation into calcite tests of the planktonic foraminifera *Trilobatus sacculifer* – Evidence from culture experiments and surface sediments

Jacqueline Bertlich[1], Dirk Nürnberg[1], Ed C. Hathorne[1], Lennart J. de Nooijer[2], Eveline M. Mezger[2], Markus Kienast[3], Steffanie Nordhausen[1], Gert-Jan Reichart[2, 4], Joachim Schönfeld[1], Jelle Bijma[5]

[1]GEOMAR Helmholtz Centre for Ocean Research Kiel, Wischhofstr. 1-3, 24148 Kiel, Germany
[2]Department of Ocean System Sciences, Royal Netherlands Institute for Sea Research, and Utrecht University, Texel, Netherlands
[3]Department of Oceanography, Dalhousie University, 1355 Oxford Street, PO Box 15000, Halifax, Nova Scotia, B3H 4R2, Canada
[4]Department of Earth Sciences, Faculty of Geosciences, Utrecht University, Heidelberglaan 2, 3584 CS Utrecht, Netherlands
[5]Marine Biogeosciences, Alfred-Wegener-Institut Helmholtz-Zentrum für Polar- und Meeresforschung, Am Handelshafen 12, 27570 Bremerhaven, Germany

*Correspondence to*: Jacqueline Bertlich (jbertlich@geomar.de)

**Abstract.** The quantitative reconstruction of past seawater salinity has yet to be achieved and the search for a direct and independent salinity proxy is ongoing. Recent culture and field studies show a significant positive correlation of Na/Ca with salinity in benthic and planktonic foraminiferal calcite. For accurate paleoceanographic reconstructions, consistent and reliable calibrations are necessary, which are still missing. In order to assess the reliability of foraminiferal Na/Ca as a direct proxy for seawater salinity, this study presents electron microprobe Na/Ca data, measured on cultured specimens of *Trilobatus sacculifer*. The culture experiments were conducted over a wide salinity range of 26 to 45, while temperature was kept constant. To further understand potential controlling factors of Na incorporation, measurements were also performed on foraminifera cultured at various temperatures in the range of 19.5 °C to 29.5 °C under constant salinity conditions. Foraminiferal Na/Ca values positively correlate with seawater salinity ($Na/Ca_{T.\ sacculifer}$ = 0.97 + 0.115 · Salinity, R = 0.97, $p <$ 0.005). Temperature on the other hand exhibits no statistically significant relationship with Na/Ca values indicating salinity to be one of the dominant factors controlling Na incorporation. The culturing results are corroborated by measurements on *T. sacculifer* from Caribbean and Gulf of Guinea surface sediments, indicating no dissolution effect on Na/Ca in foraminiferal calcite with increasing water depth up to >4 km. In conclusion, planktonic foraminiferal Na/Ca can be applied as a potential proxy for reconstructing sea surface salinities, although species-specific calibrations might be necessary.

## 1 Introduction

The combination of foraminiferal Mg/Ca and stable oxygen isotopes ($\delta^{18}O$) measured on the same specimens is a well-established approach to estimate both past sea surface temperatures (SST) and $\delta^{18}O$ of seawater ($\delta^{18}O_{sw}$) (Elderfield and

Ganssen, 2000; Anand et al., 2003; Flower et al., 2004). When corrected for past changes in global ice volume and individual temperature effects, the ice volume free $\delta^{18}O_{ivf-sw}$ indicates past changes in salinity (S), caused by changes in evaporation, precipitation and freshwater fluxes such as river discharge (Rohling and Bigg, 1998; Rosenthal et al., 2000;

Weldeab et al., 2007). Quantitative assessments of past sea surface salinities (SSS) may be erroneous, however, as the relationship between $\delta^{18}O_{sw}$ and salinity varies in space and most likely time too, due to variable freshwater contributions, for instance, from river discharge or meltwater flux (Rohling and Bigg, 1998).

Weldeab et al. (2007) applied Ba/Ca in planktonic foraminifera as an independent tool for quantitative reconstructions of salinity near rivers. Ba/Ca of river water is an order of magnitude higher than that of seawater (Weldeab et al., 2007;

Schmidt and Lynch-Stieglitz, 2011; Hönisch et al., 2011). Therefore, high foraminiferal Ba/Ca values provide evidence of increased continental run-off due to precipitation changes in the hinterland (Hall and Chan, 2004; Hoffmann et al., 2014; Bahr et al., 2018). $Ba/Ca_{foram}$ is apparently not influenced by temperature changes within 7 °C and hence linearly correlated to Ba changes in seawater, but its applicability as salinity proxy is still limited to regions close to river mouths (Weldeab et al., 2007; Bahr et al., 2013). Another theoretical and indirect approach to reconstruct salinity changes is based on the

correlation of stable oxygen isotopes ($\delta^{18}O$) and stable hydrogen isotope ratios ($\delta D$) between surface waters and the hydrological cycle (Schouten et al., 2006; Rohling, 2007). However, these different ratios must be measured on different archives such as foraminifera calcite ($\delta^{18}O$) and alkenones ($\delta D$), whereby salinity reconstructions are inherently limited by an uncertainty of at least one salinity unit (Rohling, 2007).

Because of the limitations in current approaches for the quantitative reconstruction of past salinities, the search for a direct

salinity proxy is ongoing. Recent culture experiments demonstrated a significant positive correlation between seawater salinity and Na/Ca in foraminiferal calcite of the intertidal benthic foraminifer *Ammonia tepida* (Wit et al., 2013) and the planktonic species *Globigerinoides ruber* (pink) (Allen et al., 2016). A field calibration study of living specimens (*G. ruber* (white) and *Trilobatus sacculifer*) from the Red Sea confirms the positive correlation between $Na/Ca_{foram}$ and salinity, although measured Na/Ca values were much higher (Mezger et al., 2016) compared to our culture studies.

Further studies are, therefore, essential to assess previous results and to understand the main controlling factors on foraminiferal Na incorporation. It is important to know how certain species react to extreme environmental changes and what their tolerance limits are to confirm the robustness of Na/Ca as a salinity proxy over a broad salinity range. Here, we present electron microprobe measurements on *Trilobatus sacculifer*, which were cultured over a salinity range between 26 and 45 (Bijma et al., 1990b; Nürnberg et al., 1996). Salinity ranges used in previous culture studies were rather limited, with

salinites of >30 and <40 (Wit et al. (2013): S 30–39; Allen et al. (2016): S 33–40). The high spatial resolution of the microprobe technique used here provides further insights into the inter- and intra-specimen variability of Na/Ca and potential mechanisms involved in Na incorporation. Temperature experiments with cultured *T. sacculifer* grown at temperatures from 19.5 °C to 29.5 °C and constant salinity (Bijma et al., 1990b; Nürnberg et al., 1996) were performed to check for a potential temperature effect on Na incorporation, as previously suggested by Allen et al. (2016) and Mezger et al. (2016).

In addition, early diagenetic processes can significantly alter the geochemical signal of fossil foraminifera, while the empty tests settle through the water column and are buried in deep-sea sediments (Elderfield and Ganssen, 2000; Regenberg et al., 2006). Previous studies have shown that foraminiferal Mg/Ca, Cd/Ca, and Ba/Ca are affected by calcite dissolution (McCorkle et al., 1995; Brown and Elderfield, 1996; Dekens et al., 2002; Regenberg et al., 2006, 2014). To assess potential geochemical alteration we compare Na/Ca$_{foram}$ results of *T. sacculifer* from culture experiments and from Caribbean and Gulf
of Guinea surface sediments.

## 2 Material and Methods

### 2.1 Foraminiferal species studied

The planktonic foraminifer *T. sacculifer* is a spinose, symbiont-bearing species and its chemical composition (e.g. Mg/Ca, Nürnberg et al., 1996; δ$^{18}$O, Elderfield and Ganssen, 2000) is widely used in paleoceanography to reconstruct environmental
conditions at the ocean's surface. Here, we use the new genus name *Trilobatus* instead of *Globigerinoides* following Spezzaferri et al. (2015), since the latter genus is polyphyletic in its traditional perception based on the presence of supplementary apertures on the spiral side. The species *T. sacculifer* and *T. trilobus* are the same biological species and genetically identical (Hemleben et al., 1987; Bijma et al., 1994; André et al., 2013; Spezzaferri et al., 2015). They depict different morphotypes that developed during certain ontogenetic stages and were therefore not separated in the present study.

In the Caribbean, *T. sacculifer* preferentially inhabits water depths between 10−30 m (Jones, 1968; Hemleben et al., 1987). The average habitat depth of this species is between 31−50 m in the eastern Caribbean (Jentzen et al., 2018), with highest abundances from 0−20 m (Schmuker and Schiebel, 2002). In the eastern tropical Atlantic, *T. sacculifer* calcifies ontogenetically in the mixed layer at 0−30 m, and slightly deeper in the western tropical Atlantic (0−80 m) within the photic zone above the pycnocline (Ravelo and Fairbanks, 1992; Steph et al., 2009; Jentzen et al., 2018). In the subtropical eastern
North Atlantic, *T. sacculifer* is most abundant between ~30 and 60 m (Rebotim et al., 2017). Prior to gametogenesis at the end of its life cycle, a process related to reproduction, *T. sacculifer* descends through the water column and precipitates gametogenic calcite after discarding spines (Bé, 1980; Bijma et al., 1990a; Bijma and Hemleben, 1994). The formation of a final sac-like chamber is a sign that gametogenesis is impending within 24 hours (Bé, 1980, Hemleben et al., 1989; Erez, 2003).

Since *T. sacculifer* preferentially calcifies in the mixed layer at high light intensities, an averaged habitat depth of 30 m is assumed for this study. Further, *T. sacculifer* is present throughout the year and shell fluxes do not vary significantly with seasonal changes in the tropic and sub-tropic water masses, when the annual SST is ≥25 °C (Bijma et al., 1990a, b; Bijma and Hemleben, 1994; Schmuker and Schiebel, 2002; Lin et al., 2004; Jonkers and Kučera, 2015). Accordingly, annual salinity and temperature values of 30 m water depth were taken from the World Ocean Atlas (WOA) 2013 (Zweng et al.,
2013; Locarnini et al., 2013; Schlitzer, 2015) and correlated to geochemical results of this study.

**2.2 Culture experiments**

Two sets of culture experiments are described. For both, specimens of the planktonic foraminiferal species *T. sacculifer* (originally named as *G. sacculifer* in Bijma et al., 1990b and Nürnberg et al., 1996) were collected by scuba divers at 3−8 m water depth, 1−2 miles off the south coast of Curaçao (Bijma et al., 1990b) and off the west coast of Barbados (Hemleben et al., 1987), respectively. The test length of collected foraminifera thus sampled did not exceed ~110−500 µm, but only specimens with a similar test size were chosen for each experiment (Bijma et al., 1990b). For salinity experiments, specimens were grown in filtered seawater from the sampling site at salinities of 26, 41, 44 and 45 (abbreviated as S 26−45 below). The different salinities were achieved by evaporating natural seawater at 50 °C or diluting with distilled water. The temperature was held constant at 26.5 °C during salinity experiments. Detailed information on culture protocols and experiments is given in Bijma et al. (1990b) and Nürnberg et al. (1996).

For the temperature experiment, specimens were grown in unfiltered seawater from the sampling site at temperatures of 19.5 °C, 23.5 °C, 26.5 °C and 29.5 °C (all ±0.25 °C), while keeping salinity changes to a minimum (Hemleben et al., 1987). The experiments were run during different seasons between 1980 through 1984 and because salinity varied in the winter and summer season between ~36 to ~31 at the sampling sites, caused by Orinoco and Amazon freshwater discharges, the temperature experiments were divided into 33 salinity (S 33) and 36 salinity (S 36) groups (see Hemleben et al., 1987 for more details). The salinity increased by 0.5−0.8 during the 29.5 °C experiment by evaporation in the water bath (Nürnberg et al., 1996). Electron microprobe measurements were carried out on newly grown chambers at defined salinity and temperature conditions. The final, newly grown chamber, the penultimate, and prior chambers are labeled as F (for final), F-1, F-2, F-3 and so forth.

**2.3 Surface sediment samples**

Caribbean

Foraminiferal tests collected from Caribbean surface sediment samples were used to gain information about the natural interspecific variation of Na/Ca in an open ocean setting with only minor annual changes in salinity and temperature (Figure 1, Table 1). Undisturbed surface sediments were retrieved using a multicorer (MUC) during RV *SONNE* cruise SO164 (Nürnberg et al., 2003). Sample locations and their respective bottom water depths are shown in Figure 1a and Table 1. Planktonic foraminifera were selected from the uppermost sediment layer (0−1 cm) and handpicked from the 315−400 µm size fraction. This narrow size fraction was chosen to avoid possible size-effects, as known for Mg/Ca values (e.g. Elderfield et al., 2002), or a bias of ontogenetic variations (e.g. Lin et al., 1997). Further, only foraminifera with intact, unbroken, and visually (using a binocular microscope) unaltered tests were selected for analysis. This procedure also holds for the Gulf of Guinea surface sediment samples.

The ages of Caribbean surface sediments are late Holocene, specifically they are most likely younger than 2000–3000 years (AMS[14]C ages before present, BP) (Regenberg et al., 2009; Table 1). During this time period, Caribbean sea surface temperatures (SST) varied by ±1–2 °C at most (Keigwin, 1996; Watanabe et al., 2001; Haase-Schramm et al., 2003).

Surface waters at the sampling locations are composed of Caribbean Surface Water (CSW), characterized by lower salinities throughout the year (S ~33.7–36.3) than the underlying Subtropical Underwater mass at depths of 100–300 m, reaching a salinity maximum of ~37.2 (Corredor and Morell, 2001; Schmuker and Schiebel, 2002; Haase-Schramm et al., 2003; Schönfeld et al., 2011). The CSW is mainly fed by trade wind driven water masses of the tropical and subtropical Atlantic, entering the NE Caribbean through the Lesser Antilles (Gordon, 1967). Moreover, the CSW is seasonally influenced by local precipitation and freshwater input from the Orinoco and Amazon rivers in response to shifts of the Intertropical Convergence Zone (Busalacchi and Picaut, 1983; Gordon, 1967). In spite of seasonal variations, the annual sea surface salinity across the Caribbean is restricted to variations between 35.1 and 36.3 (±0.4) at 30 m depth (Schlitzer, 2015; Zweng et al., 2013, Figure 1c). Annual SST varies between 26.8 °C and 27.7 °C (Locarnini et al., 2013; Figure 1d).

Gulf of Guinea

In addition to the Caribbean samples, surface sediments were chosen from six core locations (GIK16860 to GIK16808) in the Gulf of Guinea, located in the eastern equatorial Atlantic (Figures 1b-d, Table 1). The surface sediments were retrieved by giant box corers from January to February 1988 during RV *METEOR* cruise M6–5 (Lutze et al., 1988; Altenbach et al., 2003). Foraminiferal samples were taken from the uppermost sediment layer (0–1 cm). For geochemical analyses, specimens of *T. sacculifer* were handpicked from the 300–400 µm size fraction. Radiocarbon ages are not available for these surface sediments, but the study of Regenberg et al. (2009) estimates ages of 2430–2730 ±40–50 years BP for E-Atlantic surface sediments proximal to the Gulf of Guinea.

The Gulf of Guinea is influenced by high rates of monsoon-controlled precipitation and large freshwater discharges from the adjacent Sanaga and Niger rivers, with highest input during June to September (Altenbach et al., 2003; Weldeab et al., 2007). The annual SSS varies from east to west at surface waters from ~32 to 35 and increases within the same range from ~32 to 36 with water depth (0–60 m), but is strongly dependent on proximity to river mouths (Schlitzer et al., 2015; Zweng et al., 2013; Figures 1b, c). The annual SST varies around 27.6 °C (±0.6 °C) and decreases with depth to ~19 °C at 60 m (Schlitzer et al., 2015; Locarnini et al., 2013; Figure 1d).

Dissolution effect on foraminiferal Na/Ca

To examine the possible effect of calcite dissolution on partial $Na^+$ removal from foraminiferal calcite with increasing water depth, we here provide the calcite saturation state $\Delta[CO_3^{2-}]$ of bottom waters at all sampling sites from Caribbean and Gulf of Guinea surface sediments (Figure 6b). The $\Delta[CO_3^{2-}]$ is defined as the difference between the in situ carbonate-ion concentration ($[CO_3^{2-}]$) and $[CO_3^{2-}]$ at calcite saturation, and has been shown to be crucial for the selective ion removal of

$Mg^{2+}$ from foraminiferal calcite (Brown and Elderfield, 1998; Regenberg et al., 2006). The present day calcite saturation horizon, where $\Delta[CO_3^{2-}]$ is 0 µmol kg$^{-1}$, is at ~4.7 km in the Caribbean and ~4.4 km in the Gulf of Guinea (Regenberg et al., 2006, 2014, Figure 6b). According to Regenberg et al. (2014), the onset of planktonic foraminiferal Mg/Ca dissolution occur globally below a critical calcite saturation level in bottom waters of $\Delta[CO_3^{2-}]$ ~21.3 ±6.6 µmol kg$^{-1}$. This critical threshold is reached at water depths around ~3.5 km in the Caribbean and ~2 km in the Gulf of Guinea, respectively. Below, Mg/Ca values decline linearly by ~0.05 ±0.02 mmol mol$^{-1}$ per µmol kg$^{-1}$ (Regenberg et al., 2006, 2014). Conversely, $Mg/Ca_{foram}$ values are stable at supersaturated bottom waters with $\Delta[CO_3^{2-}]$ >40 µmol kg$^{-1}$, or above 400 m water depths (Regenberg et al., 2006, 2014). Although a clear dependence of foraminiferal Mg/Ca with water depth and $\Delta[CO_3^{2-}]$ is shown in previous studies, no data are available yet about the Na/Ca to $\Delta[CO_3^{2-}]$ relationship over time and with water depth in foraminiferal calcite. Hence, we assume a similar $\Delta[CO_3^{2-}]_{critical}$ threshold (~21 µmol kg$^{-1}$) for our Na/Ca results, where the onset of partial Na$^+$ removal could possibly start, which needs further study.

Bottom water $\Delta[CO_3^{2-}]$ values for Caribbean surface sediments presented here are taken from preexisting datasets for the same core locations from Regenberg et al. (2006). To calculate $[CO_3^{2-}]_{in\ situ}$, Regenberg et al. (2006) used total alkalinity (TA) and TCO$_2$ data from the World Ocean Circulation Experiment (WOCE) (RV *KNORR* cruise 1997, section A22, campaign 316N151_4, cchdo.ucsd.edu). Calculated bottom water $\Delta[CO_3^{2-}]$ values for the Gulf of Guinea water depth profile were adopted from Weldeab et al. (2016) and Weldeab (2012) for overlapping core locations. Missing values were calculated with the program CO2SYS by Pierrot et al. (2006) using the constants K1 and K2 following Mehrbach et al. (1973), refitted by Dickson and Millero (1987), and the constant $K_{SO4}$ by Dickson (1990). To calculate the in situ carbonate concentration, the respective data (TA, TCO$_2$, *p*CO$_2$, pH, salinity, temperature, pressure, total phosphate and silicate) were taken from WOCE (RV *L'ATLANTE* cruise, section A13, campaign 35A3CITHER3_2, cchdo.ucsd.edu). The $\Delta[CO_3^{2-}]$ at saturation was calculated after Jansen et al. (2002).

## 2.4 Sample preparation and electron microprobe analysis of cultured foraminifera

Three individual foraminifera per salinity experiment and five individual foraminifera for every temperature experiment were selected from the same size fraction (250–400 µm). Only specimens without a sac-like final chamber were chosen for geochemical analysis to avoid a potential impact from other factors than those targeted here. Newly grown chambers under defined laboratory conditions were compared to chambers grown in situ before placing them in culture (Hemleben et al., 1987). For the electron microprobe analyses, all cultured specimens were mounted in epoxy resin (Nürnberg et al., 1996). To expose a fresh and planar surface of test cross sections, the sample mounts were polished with diamond polishing pastes with a 1 µm grain size in the last step. The samples were coated with a carbon coating of 20 nm to reduce charging effects and hence to increase the spatial resolution of the obtained microprobe element maps.

The measurements were performed on a JEOL JXA 8200 electron microprobe at GEOMAR. In order to map and quantify Na distribution patterns in both single chambers and chamber wall profiles of *T. sacculifer*, quantitative wavelength-dispersive spectrometry (WDS) was applied. Na/Ca values were derived from elemental maps, reaching a higher spatial resolution than point measurements (Figure 2). Na/Ca values of newly grown calcite were derived from at least 6 single maps along the inner (non-gametogenic) part of the relevant chamber, varying in size between 5 μm x 5 μm and 10 μm x 20 μm (Figure 2). Calcium intensities (wt %) deviating more than 10 % from the maximum Ca intensity within one map were discarded to ensure that measurements affected by pores or cracks within the test were excluded. Further to avoid a mixed signal, maps with accumulated higher Na concentrations ($\pm 2\sigma$) at calcite spines and spine bases were not integrated into the calibration as well (referring to Branson et al., 2016). Every pixel within a map is 1 μm in diameter and has an element intensity. Pixels were averaged for the entire map (Figure 2). The maps were generated with a focused electron beam, adjusted with an acceleration voltage of 15 kV, a beam current of 80 nA, a spot diameter and step size of 1 μm. The dwell time was set at 50 ms per measurement with 5 accumulations, resulting in a total measurement time of 250 ms on the peak position. Na and Ca intensities were measured with the crystals TAPH (K$\alpha$) and PETH (K$\alpha$). The background intensities were measured separately at almost the same sample location with exactly the same settings and 250 ms accumulation time. Absolute Na/Ca values were calculated using a constant calibration factor for each element after subtracting the background intensities from the total element intensities. The obtained net intensities were converted to concentrations and corrected to known concentrations (in wt %) from referenced materials (Table 2). Measurements on standard materials were performed with 10 μm x 10 μm maps (number of pixels: n = 100). The precision of the element analysis and the uncertainty between single measurements is given by the relative standard deviation of averaged element maps (RSD = (standard deviation (1$\sigma$) / mean value) $\cdot$ 100) in %, which is less than 5 % for used standard materials (Table 2). To check for matrix effects, the in house standard A2 (Ohde et al., 2003), a natural modern coral, with a similar elemental composition to foraminiferal calcite was also measured. The accuracy of analyses is expressed as the relative error in %, representing the deviation between the measured and the reference values (Table 2).

The intra-test variability is expressed as RSD in % and is calculated from averaged Na/Ca maps within chamber wall cross sections of one foraminifer. The inter-test variability (RSD in %) is derived from mean Na/Ca values of one foraminifer and exhibit differences between specimens within the same experiment. Pearson's correlation coefficient R describes within the range of −1 to 1 the correlation between trace elements and the experimental settings, here salinity and temperature. A positive or negative correlation occurs if R is >0 or R <0. The appending *p*-value indicates its significance. If *p* is less than 0.05 (5 %) the results are statistically significant. The Na/Ca results for every culture experiment were tested for normality with the Shapiro-Wilk test and corresponding *p*-values (Appendix A). If *p* is less or equal to 0.05 (95 % confidence interval), the data are not normally distributed. The range and distribution of averaged Na/Ca maps for each salinity and temperature experiment is shown as box and whisker plots, indicating the first and third quartiles (Figure 3). The horizontal lines inside the box represent the mean and median. Minimum and maximum values are shown by vertical lines (whiskers) outside the

box. All Na/Ca values of culture experiments are presented with the standard error of the mean (SEM = (standard deviation $\sigma / \sqrt{n}$)).

## 2.5 ICP−OES sample preparation and analysis of surface sediment samples

Traditional measurements of the elemental composition of foraminiferal calcite employ chemical cleaning procedures as pioneered by Boyle (1981). These were modified for Na/Ca after Barker et al. (2003) as follows. Each sample consisting of 25−35 specimens from the same size fraction (Caribbean: 315−400 µm; Gulf of Guinea: 300−400 µm) were cracked under a binocular and placed in acid cleaned PP micro-centrifuge tubes before being rinsed under ultrasonic agitation with ultrapure water (<18 MΩ) three times, ethanol and again ultrapure water, to remove clays. To eliminate metal oxides in the reductive

cleaning step, 100 µL of a buffered solution of hydrous hydrazine (1 M hydrazine mixed with 0.25 M citric acid in 16 M ammonia, following the protocol of Boyle and Keigwin (1985)) was added to each vial and reacted in a hot (90 °C) water bath for 30 minutes. The samples were thoroughly rinsed to remove these chemicals with ultrapure water before being transferred to fresh acid cleaned tubes. Residual organic matter was subsequently removed by a 0.3 % $H_2O_2$ solution (250 µL added to each vial), kept again for 20 min in a hot water bath and rinsed three times with ultrapure water. A final leaching

step quickly rinsed with 0.001 M $HNO_3$ before the samples were finally rinsed once more with ultrapure water. The remaining water was removed and samples were diluted right before the measurements in 500 µL 0.075 M $HNO_3$.

The elemental analyses were performed simultaneously with a VARIAN 720 Inductively Coupled Plasma − Optical Emission Spectroscopy (ICP−OES) at GEOMAR. The device is equipped with a cooled cyclonic spray chamber combined with a microconcentric nebulizer (200 µL min$^{-1}$ sample uptake), optimized for signal stability. The most intense element

lines without interferences were chosen for elemental analysis. These wavelengths were 70.60 nm for Ca, 279.55 nm for Mg and 589.59 nm for Na. The measurement routine repeats 5 times for every sample and standard solution and the RSD of respective measurements give an estimation of the internal error, which is 0.31 % RSD for Na and 0.22 % RSD for Ca. To correct for analytical drift of the instrument, the reference material ECRM 752-1, commonly used for foraminiferal Mg/Ca analysis (reference value: 3.762 mmol mol$^{-1}$ in Greaves et al., 2008), was measured after a set of 10 foraminifera samples

(RSD: 0.06 %). As there are no Na values in the literature for the ECRM 752-1, we additionally measured the coral reference material JCp−1, with reference values of 0.586 (±0.003 SD) wt % $Na_2O$ and 53.52 (±0.29 SD) wt % CaO in Okai et al. (2002). The JCp−1 was measured after a set of 10 foraminifera samples, resulting in a average Na/Ca value of 18.75 mmol mol$^{-1}$ ±0.11 mmol mol$^{-1}$ (number of measurements = 3) during our ICP-OES analysis with a precision of 0.3 % (RSD). The accuracy of analysis is 5.5 %, which corresponds to a variation of ±0.54 mmol mol$^{-1}$ from the reference JCp-1 Na/Ca value.

## 3 Results

### 3.1 Sodium incorporation into foraminiferal calcite

Salinity experiments

Averaged Na/Ca values for the salinity experiments are presented in Table 3. Datasets for T = 26.5 °C of both temperature experiments (S 36, S 33) were included into the Na/Ca–salinity calibration. Newly grown chambers of *T. sacculifer* have, on average, Na/Ca values between 3.86 and 6.40 mmol mol$^{-1}$ over a range of 19 salinity units. Foraminiferal Na/Ca values correlate positively with sea surface salinity: Na/Ca$_{foram}$ = 0.97 + 0.115 · Salinity (R = 0.97, *p* <0.005) and increase by 2.25 % per salinity unit, corresponding to an absolute Na/Ca value of 0.12 mmol mol$^{-1}$ (Figure 4). Between salinities of 26 to 36, there is a marked increase in Na/Ca$_{foram}$ by 1–1.25 mmol mol$^{-1}$, based on the lowest and highest Na/Ca value of each experiment. The smallest possible range is detected between salinities of 33 to 36 where Na/Ca values increase by ~0.5 mmol mol$^{-1}$, which is 0.1 mmol mol$^{-1}$ larger than the 95 % confidence interval.

Single values and the calculated intra- and inter-test variability for each experiment are presented in tabular form in the Appendix B. The inter-test Na/Ca variability (RSD in %) ranges from 5.5 % to 6.5 % (±0.29–0.34 mmol mol$^{-1}$) between single foraminifera from the same salinity experiment, which is larger than the analytical error of the electron microprobe (RSD ≤0.2 %–1.2 %, Table 2). The intra-test Na/Ca variability (between different maps on one chamber) is twice as large as the inter-test variability and varies between 7.4 % and 13.3 % (±0.39–0.70 mmol mol$^{-1}$, Appendix B).

The largest ranges of Na/Ca within one experiment are observed among foraminifera cultured at salinities >41 (Figure 3a). Intra-test map analyses deviate within chamber cross sections of one foraminifer by ±1.1 mmol mol$^{-1}$, while inter-test Na/Ca values between foraminifera deviate by ±0.25 mmol mol$^{-1}$ (S = 41). For the remaining experiments at salinities <41, Na/Ca values deviate within the same test by ±0.35–0.65 mmol mol$^{-1}$ and vary among foraminifera by ±0.22–0.55 mmol mol$^{-1}$, which equals the range of the 95 % confidence interval. The culture experiment conducted at a salinity of 44 deviates from previous results: Na/Ca shows a decrease of ~0.5 mmol mol$^{-1}$ compared to the experiments at salinities below 41 (Figures 3a, 4). According to Nürnberg et al. (1996), all specimens of the S 44-experiment underwent gametogenesis and were enriched in Mg/Ca. Because of differences in elemental composition and calcite structure between gametogenic calcite and ontogenetic calcite (Erez, 2003), the results of the S 44-experiment were hence excluded from the regression shown in Figure 4. Still, even when including foraminifera with gametogenic calcite in our calculations, Na/Ca increases by 2 % per salinity unit (Na/Ca$_{foram}$ = 1.46 + 0.097 · Salinity, R = 0.90, *p* <0.01). Na/Ca values measured on chambers (F-3, F-4) grown in situ before sampling and culturing reflect the salinity of ambient seawater in the Caribbean (S ~35.9) and correspond to the results of culture experiments at a salinity of 36 (Zweng et al., 2013; Figure 4). Nonetheless, Na/Ca values of chambers grown in situ were not included into the calibration, because the calcification conditions are not exactly known.

Temperature experiments

Within the 36 salinity group, averaged Na/Ca values of newly grown foraminiferal calcite, precipitated at temperatures from 19.5 to 29.5 °C, range from 3.96 to 5.49 mmol mol$^{-1}$ (Table 4, Figure 5). In the 33 salinity group, averaged Na/Ca vary between 4.10 to 5.78 mmol mol$^{-1}$ (Table 5, Figure 5). No significant ($p$ <0.69 (S 36), $p$ <0.14 (S 33)) correlation is found between Na/Ca and temperature for both experiments (Figure 5).

The inter- and intra-test variability of Na/Ca for the different treatments is shown in the Appendix B. Box and whisker plots demonstrate the Na/Ca distribution for each temperature experiment (Figure 3b). The inter-test variability of Na/Ca (between single foraminifera of the same experiment), expressed as RSD, varies from 7.3–9.8 % (S 36) and 3.2–11.5 % (S 33), corresponding to absolute values of ±0.38–0.51 mmol mol$^{-1}$ (S 36) and ±0.17–0.6 mmol mol$^{-1}$ (S 33). Na/Ca values vary within newly formed chambers of one test (intra-test) between 1.8–16.2 % (S 36; ±0.09–0.85 mmol mol$^{-1}$) and 3.6–10.6 % (S 33; ±0.19–0.55 mmol mol$^{-1}$). Notably, the highest Na/Ca intra-test distribution (up to 16 %) only occurs in experiments below 24 °C (S 36), which is in contrast to the smallest intra-test variation of 1.8 %, observed for treatments close to the natural habitat conditions of *T. sacculifer* (S 36, 26.5 °C; Figure 3b, Appendix B). This corresponds to the lowest intra- and inter-test variability of ~1.7 %, measured on tests, which grew in situ in the open ocean before culturing started (see also Sect. 2.3).

Surface sediments

The ICP–OES–derived Na/Ca values of *T. sacculifer* from both the Caribbean and Gulf of Guinea surface sediments match the results from single map analyses (electron microprobe) of chamber cross sections from cultured foraminifera (Figures 4, 6a). All surface sediment-related data are provided in Table 1.

Caribbean

Among all Caribbean stations, the surface water (0–60 m water depth) salinity varies insignificantly by less than ±0.2 throughout the year (see section 2.3, Fig. 1c). Temperature changes are also minor (with a maximum of ±0.3 °C at 0–60 m water depth, Fig. 1d). Sediment surface foraminiferal Na/Ca results in relation to salinity and temperature are shown in Figures 4, 5 and 6a. Accordingly, all Na/Ca values are presented with horizontal error bars. These indicate the salinity and temperature variations between 0 and 60 m water depths, representing the main habitat migration range of *T. sacculifer* (Sect. 2.1). As a result of minimal parameter changes with depth, horizontal error bars could be as small as their symbol sizes.

Maximum Na/Ca values (5.68 ±0.3 mmol mol$^{-1}$) were measured at stations SO164–7–3 and –18–1 northwest of Haiti, showing the highest annual SSS at 30 m water depth (~36.3) of all sampling sites (Figures 1a, c). Lowest Na/Ca values of 4.64 and 4.79 ±0.2 mmol mol$^{-1}$ were found southwest and southeast of Puerto Rico (stations SO164–23–3 and –22–2), with an annual SSS of ~35.4 at 30 m water depth (Figures 1a, c). In this respect, station SO164-24-3, which is close to the Puerto Rico stations, is exceptional as high Na/Ca values of 5.51 mmol mol$^{-1}$ were recorded at apparently low SSS (35.37).

Nevertheless, a $Na/Ca_{T.\ sacculifer}$ increase of about 0.2 mmol mol$^{-1}$ per salinity unit is noticeable between the Puerto Rico stations and SO164−07−3/18−1.

All Caribbean stations exhibit Na/Ca values within the error of the confidence interval of the Na/Ca-salinity relationship derived from culture experiments (Figures 4, 6a). This observation is also consistent with Na/Ca values of in situ grown chambers in the open ocean (~27.4 °C, ~S 35.7, WOA13) before culturing started (Figure 6a, indicated by yellow reversed

triangles). The Na/Ca variability of averaged test concentrations (20−30 foraminifera per station) from Caribbean surface sediments varies between 5.5−13 % in RSD (±0.2 to 0.7 mmol mol$^{-1}$) among stations (Figure 6a). These variations are consistent with results of single foraminifera analysis, as the inter-test variability indicates (salinity experiments: RSD of 5.5−6.5 %, ±~0.3 mmol mol$^{-1}$; temperature experiments: RSD of 3.2−11.5 %, ±~0.2−0.6 mmol mol$^{-1}$) (Figures 3, 4).

Gulf of Guinea

In contrast to the Caribbean, all sampling sites in the Gulf of Guinea are proximal to the Niger river mouth (Fig. 1b) with the the upper 0−60 m being influenced by riverine freshwater input. Salinities vary from ~32 to ~35 at the surface and increase with water depth by 1.5−2 at stations close to freshwater discharge (GIK 16860-1, 16864-1, 16865-1) (Fig. 1c). Temperature variations at surface waters are minor (~27−28 °C) compared to the significantly decreasing temperatures with water depth

(0−60 m) by ~27.5 to 19 °C (Fig. 1d). Na/Ca results are therefore presented with horizontal error bars, in order to account for the salinity (Fig. 4, 6) or temperature (Fig. 5) variations, respectively, in the preferred habitat depth (0−60 m) of *T. sacculifer*. Gulf of Guinea Na/Ca results also lie within the 95 % confidence interval of our salinity culture experiments, even when taking into account their uncertainty marked by horizontal error bars (Fig. 6a). In comparison to the foraminiferal Na/Ca variability of Caribbean surface sediments, samples of the Gulf of Guinea show the smallest Na/Ca variability of ≤2

% (≤0.1 mmol mol$^{-1}$) at all stations (Fig. 6a). The lowest Na/Ca value of 4.77 ±0.03 mmol mol$^{-1}$ in *T. sacculifer* was measured at station GIK 16865-1 close to the Niger and Sanaga river mouths with SSS of 35 at 30 m water depth (Figures 1b, c), while stations next to it at the same salinity level present higher Na/Ca values (+0.15−0.24 mmol mol$^{-1}$).

In the Gulf of Guinea temperature variations are higher (±~4 °C) than changes in salinity with water depth and between sampling sites. Despite this, Na/Ca values are in line with results of temperature culture experiments, demonstrating the

limited impact of temperature on Na incorporation (Figure 5).

Calcite dissolution effect on Na/Ca in foraminiferal tests

Overall, Na/Ca values of *T. sacculifer* from both Caribbean and Gulf of Guinea surface sediments, show no significant correlation with increasing water depth, corresponding to the $\Delta[CO_3^{2-}]$ versus water depth profile at respective sampling

stations (Table 1, Figure 6b). Na/Ca values of both sampling regions lie within the 95 % confidence interval (±0.25 mmol mol$^{-1}$) of our salinity culture experiments (not affected by dissolution) and show the same range of inter-test variability (Figures 4, 5 and 6a, Table 1).

In detail, although the onset of Mg/Ca dissolution in planktonic foraminifera starts below the critical $\Delta[CO_3^{2-}]$ threshold of
~21 µmol kg$^{-1}$ (defined by Regenberg et al., 2006, 2014), stated at ~3.5 km in the Caribbean, no similar trend is obvious
from our foraminiferal Na/Ca values at bottom waters until 4 km water depth (Figure 6b - Caribbean). Only three sampling
stations (Caribbean: SO164−01−3, −22−2, −23−3) and their respective Na/Ca values are below the critical $\Delta[CO_3^{2-}]$ threshold
for Mg/Ca dissolution, tempting to a decline in Na/Ca close to the calcite saturation horizon (~4.7 km). Corresponding
Na/Ca$_{T. sacculifer}$ values of the latter stations show the lowest values of all Caribbean surface sediment samples. Although these
Na/Ca values do not intersect with any error bars of values above 4 km, they are still within the error of the 95% confidence
interval of cultured *T. sacculifer* (compare Figures 6a and 6b).

Foraminiferal Na/Ca values of Gulf of Guinea sediments show no clear trend versus $\Delta[CO_3^{2-}]$ (Figure 6b). Compared to the
Caribbean, the critical $\Delta[CO_3^{2-}]$ of ~21 µmol kg$^{-1}$ for Mg/Ca dissolution (Regenberg et al., 2014) is already reached at 2.5−3
km water depth in the Gulf of Guinea, but all stations are at supersaturated bottom waters >35 µmol kg$^{-1}$ (Table 1) above 2.5
km. Further, the lowest Na/Ca$_{foram}$ value (GIK 16865-1) was measured at samples collected close to the Niger river discharge
(Figure 1b). Those samples were taken from the deepest station at this site (~2.5 km water depth, Table 1). Instead, both
stations next to it (GIK 16860-1/16864-1) show higher Na/Ca values (+~0.2 mmol mol$^{-1}$), collected at 0.2 and 1.5 km water
depth. Nevertheless, three of four stations show the same Na/Ca$_{foram}$ value of 4.92 mmol mol$^{-1}$ between water depths of
around 1.5 to 2.2 km, pointing to Na/Ca values unaffected by dissolution.

## 4 Discussion

### 4.1 Inter- and intra-test variability

The inter- and intra-test element distribution revealed by high-resolution electron microprobe mapping provides important
information on test Na/Ca composition and heterogeneity, which is fundamental for proxy development. In this study, we
explore the variability (expressed as RSD in %) from averaged Na/Ca electron microprobe maps within chamber wall cross
sections of one foraminifer and among individual specimens. From our culture experiments the Na/Ca inter- and intra-test
variability varies between approximately 2 and 16 % (Appendix B). This is in line with previous studies, reporting an inter-
test variability of 9−17 % (RSD), measured by laser ablation ICP-MS (Wit et al., 2013; Mezger et al., 2016).

Compared to the Mg/Ca variability (≥30 %) observed in symbiont-bearing foraminiferal species, the Na/Ca inter-test
variability of *T. sacculifer* is relatively low, half that determined for Mg/Ca (Sadekov et al., 2005; Dueñas-Bohórquez et al.,
2011b). Previous studies on (trace) metal distribution in various species of planktonic foraminifera showed for instance
prominent Mg/Ca banding, alternating in high- and low Mg-calcite bands within chamber walls. These variations are likely
linked to different calcification mechanisms, with the inner primary calcite enriched in Mg and the outer thickening calcite
depleted (together named as ontogenetic calcite) (Erez, 2003; Eggins et al., 2003; Sadekov et al., 2005; Dueñas-Bohórquez et
al., 2011b; Hathorne et al., 2009; Spero et al., 2015). Potential factors controlling Mg/Ca values in banding, which could

change the biological activity of foraminifera and/or their symbionts, are changes in calcification temperatures (due to

habitat migration or seasonal change), seawater pH, pH in the calcifying microenvironment, carbonate ion concentrations or diurnal day/night cycles (Eggins et al., 2004; Sadekov et al., 2005; Dueñas-Bohórquez et al., 2011b; Spero et al., 2015; Davis et al., 2017; Fehrenbacher et al., 2017).

These potential factors affecting Mg heterogeneity could also bias Na incorporation. In the following, possible effects on the Na/Ca inter- and intra-test variability are discussed. The reduced Na/Ca variability compared to higher Mg/Ca variations

implies less pronounced Na/Ca banding as found for *Orbulina universa* (Branson et al., 2016) and possible differences in calcification pathways (e.g. ion transport mechanisms to the calcification site in the cell). Using time-of-flight secondary ionization mass spectrometry, only one fine band (<500 nm) enriched in Na was found in all specimens, identified close to the primary organic sheet where the spines of foraminifera grow from (Branson et al., 2016). This is consistent with observations from Erez (2003), who reported higher Na concentrations in benthic foraminifera in proximity of the primary

organic sheet, located between the inner and outer calcite. Branson et al. (2016) demonstrated that Na is not associated with higher Mg concentrations within tests of *Orbulina universa*, which could be also valid for *T. sacculifer* and other planktonic species.

Nevertheless, although the high Mg variability (>30 %) is strongly biologically controlled, changes of ambient environmental conditions, such as temperature or salinity, could also bias Mg/Ca values in foraminifera (Nürnberg et al.,

1996, Bentov and Erez, 2005, 2006, Dueñas-Bohórquez et al., 2009). Transferred to our study, higher Na/Ca intra-test variations within single wall cross sections of each salinity and temperature experiment (Appendix B) may occur due to possibly higher ecological stress levels of (individual) foraminifera in culture. This hypothesis, which would imply a strong dependence of Na incorporation on environmental changes, is supported by the appearance of the lowest Na/Ca range in those tests, which grew at culture settings close to the natural habitat of *T. sacculifer* (26.5 °C, S 36) (Figure 3).

The inhibition of chamber formation or gametogenesis, as observed in cultured foraminifera at different salinities (Bijma et al., 1990b), may be a prime indication of the biological response of foraminifera to environmental stress. Only within one culture experiment at a salinity of 41, foraminifera precipitated more than 2 to 3 new chambers, while only one additional chamber grew at salinities below and above 41. Precipitation of gametogenic calcite (GAM) was only observed at salinity experiments of S 44. Compared to salinity culture experiments, foraminifera were able to precipitate 3 to 4 new chambers in

all temperature experiments (Hemleben et al., 1987; Bijma et al., 1990b; Nürnberg et al., 1996).

At the end of their lifecycle foraminifera precipitate gametogenic calcite below their averaged living depth and commonly cooler waters, which should result in calcite with lower Mg/Ca values (Nürnberg et al., 1996; Erez, 2003; Sadekov et al., 2005, Rebotim et al., 2017). In the culture experiments, the rate of gametogenesis was significantly influenced by salinity. As shown in different culture experiments of Bijma et al. (1990b), around 40 % of foraminifera successfully underwent

gametogenesis at salinities between 41 and 44, but the reproduction frequency decreases ~10-fold above a salinity of 45. However, Nürnberg et al. (1996) already noted that GAM calcite, when secreted at the same temperatures, is enriched in Mg relative to ontogenetic calcite, because their biomineralization mechanisms could be fundamentally different (Bentov and

Erez, 2006; Hamilton et al., 2008). Hamilton et al. (2008) proposed different calcification processes in the microenvironment of foraminifera prior to the gamete formation, compared to the calcite precipitation of ontogenetic calcite (see Sect. 4.4).

Similarly, our study points to geochemical differences between GAM and ontogenetic calcite with reduced Na/Ca values in GAM calcite at a salinity of 44 (Figures 3, 4). This agrees with findings that the outer gametogenic calcite layer has lower Mg/Ca values (e.g. Brown and Elderfield, 1996; Hathorne et al., 2003; Sadekov et al., 2005). Although we could show differences between GAM and ontogenetic calcite, it needs to be considered, in this respect, that foraminifera cultured in the laboratory do not migrate to depth while they precipitate GAM calcite, as foraminifera do in the field (Bé, 1980, Hemleben

et al., 1989; Erez, 2003).

## 4.2 Na/Ca as a proxy for salinity

For *T. sacculifer*, Na incorporation increases significantly ($p$ <0.005) by 2.25 % per salinity unit during culture experiments (Salinity = (Na/Ca$_{foram}$ − 0.97)/0.115) (our study, Figure 4). Within each experiment Na/Ca values are normally distributed (Shapiro-Wilk test, Appendix A). Our results are broadly consistent with previous culture studies (Figure 7), showing a

positive correlation between Na/Ca and seawater salinity for the benthic foraminifer *Ammonia tepida* and increasing Na/Ca by 3 % per salinity unit (Wit et al., 2013). Allen et al. (2016) observed a Na/Ca increase of 1 % per salinity unit in cultured *G. ruber* (pink), while there was no such salinity relationship in their *T. sacculifer* over a salinity range from 33 to 40 (Fig. 7). As the slope of the Na/Ca versus salinity regression appears steeper for *G. ruber* than for *T. sacculifer*, Allen et al. (2016) suggested that the sensitivity of Na-incorporation in response to salinity change is species-specific, and higher for *G. ruber*.

Conversely, no differences were observed in Na/Ca values between *G. ruber* and *T. sacculifer* within the field calibration study from the Red Sea (Mezger et al., 2016; Fig. 7). In that study Na/Ca values were also markedly higher compared to Wit et al. (2013) and Allen et al. (2016) (Fig. 7). Na/Ca increases by ~7.5−8 % per salinity unit in *G. ruber* (w) and *T. sacculifer*, respectively (Mezger et al., 2016). The authors attributed this deviation to region-specific environmental and oceanographic conditions compared to the open ocean (Mezger et al., 2016; see detailed discussion in Sect. 4.3). Overall, this demonstrates

the necessity for establishing species-specific (and regional dependent) calibrations, similar to foraminiferal Mg/Ca paleothermometry.

With respect to paleosalinity reconstructions, our results from culture experiments and surface sediment samples indicate that foraminiferal Na/Ca may serve as a direct proxy for ocean salinities, but has limitations, as listed below. We demonstrate for cultured *T. sacculifer* that a change of ±1 salinity unit results in a Na/Ca change of 2.25 %, i.e. ±0.12 mmol

mol$^{-1}$ in absolute terms. This change definitely exceeds the analytical uncertainty of the electron microprobe and the ICP−OES measurements (0.2−0.4 % RSD), providing the ability to resolve past SSS changes of this magnitude. Although the intra- and inter-test Na/Ca variability of cultured *T. sacculifer* shown here is up to 10 times higher (Sect. 4.1) than the absolute change per salinity unit, averaged Na/Ca values of specimens for each treatment fit with the Na/Ca−salinity calibration and lie within the 95% confidence interval (Figure 4). The respective average Na/Ca uncertainties, expressed as

the standard error (SE), vary between ~0.05 to 0.37 mmol mol$^{-1}$ per salinity condition, which would correspond to a minimum salinity sensitivity of 0.14 units (Appendix C). As already demonstrated for Mg/Ca, which has strong intra- and inter-test heterogeneity, the deviation from the mean would decrease as the number of total specimens used for analyses increases (Sadekov et al., 2008; De Nooijer et al., 2014a). To resolve a salinity change of 0.1 (SE: 0.036 mmol mol$^{-1}$), 40 to 50 specimens are statistically necessary, based on our Na/Ca-salinity calibration (Appendix C). Sodium is known to be heterogeneously distributed in foraminiferal calcite (Branson et al., 2016) and Na/Ca intra- and inter-test heterogeneity is also observed here with single electron microprobe maps on chamber wall cross sections within one specimen having a standard error of 0.08–0.61 mmol mol$^{-1}$. The latter standard error corresponds to a minimum salinity sensitivity of ~0.7 units. To resolve a salinity change of 0.1 (SE: 0.01 mmol mol$^{-1}$) with single spot analysis on one specimen, 30–60 measurements (S 26–36) or up to 80 measurements (> S 41) are required (Appendix C). The inter-test heterogeneity of Na/Ca in cultured *T. sacculifer* observed here is rather low (~5.5–6.5 RSD % for salinity experiments) compared to the inter-test variability of ~14 RSD % observed by Mezger et al. (2016) (see Sect. 4.1). Since we exclusively analyzed chamber walls, and avoided measuring other parts of the test with a distinct morphology and (potentially) different chemical composition (e.g. GAM calcite, crust, spines), Na/Ca may be more variable than reported here. Nonetheless, the Na/Ca range between single foraminifera of all culture experiments measured by the electron microprobe (±0.17–0.6 mmol mol$^{-1}$) is rather similar to that averaged from 20–30 *T. sacculifer* specimens (which all have various amounts of gametogenic calcite), measured by the ICP–OES (Caribbean surface sediment samples: ±0.18–0.56 mmol mol$^{-1}$, Gulf of Guinea surface sediment samples: ±0.03–0.14 mmol mol$^{-1}$) (Figures 4, 6a). Both field studies suggest at least 20 specimens are necessary to resolve past salinity changes of 0.5 to 1 (SE: 0.18–0.36 mmol mol$^{-1}$, Appendix C).

With a sensitivity of 0.5 salinity units, reconstructions of paleo-river discharge (e.g. see Weldeab et al., 2007) as well as glacial meltwater discharge (e.g. in Flower et al., 2004) should become possible, independent from stable oxygen isotope approaches. Therefore, the application of Na/Ca$_{foram}$ in multi-proxy approaches, as already suggested in Vetter et al. (2017) to combine e.g. foraminiferal Mg/Ca–$\delta^{18}$O with Ba/Ca, could infer more confidence of such reconstructions by providing absolute salinity values instead of previous indirect estimations.

## 4.3 The reliability of the foraminiferal Na/Ca-salinity proxy

Temperature effect on Na incorporation

Before accepting Na/Ca as a robust paleoceanographic tool, other factors than salinity potentially affecting Na incorporation need to be qunatified. The statistically insignificant correlation of Na/Ca to temperature changes of cultured *T. sacculifer* in our study (Figures 3, 5) is in agreement with results reported by Allen et al. (2016) (Figure 8). In contrast, the Red Sea plankton pump study of Mezger et al. (2016) showed a negative correlation between Na/Ca values and temperature for *G. ruber* ($R^2 = 0.84$, *p* <0.001) and *T. sacculifer* ($R^2 = 0.95$, *p* <0.001) (Figure 8). It is important to note that increasing temperatures (~26–29 °C) in these samples are paralleled by decreasing salinities (from ~40 to 37; Mezger et al., 2016, Fig.

8) and possibly other factors, such as changes in carbonate chemistry, in the Red Sea. Therefore, it was not possible to separate effects from each other. The absence of a temperature effect on Na incorporation in foraminiferal tests shown here, would suggest that the apparent correlation observed by Mezger et al. (2016) was probably not due to temperature. Also a putative impact of the extreme salinity in the Red Sea can be disregarded as the culture study here extended even beyond the very high salinities found in the Red Sea (45 versus 41).


Although *T. sacculifer* prefers higher temperatures (optimum 28 °C) and is mostly absent below 23 °C (Žarić et al., 2005), Na/Ca$_{foram}$ changes are minor in our study within temperature changes (19.5–29.5 °C) (Fig. 3b). Hence, there is quite robust support that variations in foraminiferal Na/Ca are not primarily driven by temperature.


Further potential factors controlling Na incorporation

Although temperature shows no statistically significant relationship with foraminiferal Na/Ca, aberrant Na/Ca values between the same and amongst other planktonic and benthic foraminiferal species, need to be discussed. Beside salinity as the potential dominant factor controlling Na incorporation into calcite tests, changes in seawater carbonate chemistry, biological processes, different sample preparation techniques and/or diagenetic altering of fossil tests may affect the geochemical signature of foraminifera, which will be discussed below.


Averaged Na/Ca values of cultured *T. sacculifer* (our study) are lower by ~1–2 mmol mol$^{-1}$ than those reported for *A. tepida* in Wit et al. (2013) and *G. ruber* and *T. sacculifer* in Allen et al. (2016) for the overlapping salinity interval of 30 to 38 in culture experiments (Figure 7). Foraminiferal samples from the Red Sea are even higher by up to 4 mmol mol$^{-1}$, with Na/Ca values of 7–14 mmol mol$^{-1}$ (±2–6 mmol mol$^{-1}$) in *G. ruber* (white) and *T. sacculifer* within the salinity range of 37–40 (Mezger et al., 2016; Fig. 7). Mezger et al. (2016) related the extremely high Na/Ca values to many co-varying factors (e.g. carbonate chemistry, salinity, temperature) in the Red Sea, which may differ from open ocean conditions. Increasing dissolved inorganic carbon (DIC), pH, and the calcite saturation state ($\Delta[CO_3^{2-}]$) are likely promoting higher calcite precipitation rates and hence, the uptake of trace elements, as observed for cultured benthic foraminifera (Bentov et al., 2009; De Nooijer et al., 2014b). Beside that, Allen et al. (2016) did not note a significant relation of Na/Ca with $[CO_3^{2-}]$ (or pH) and DIC concentration and could not see a clear relationship between trace element partitioning and calcification rates in *G. ruber* (pink) and *T. sacculifer*. Hence, they assumed that calcification rates are not important factors for Na$^+$ incorporation (Allen et al., 2016). This could indicate that differences in Na/Ca within one species could be region-specific due to indirect environmental controls on Na-incorporation.



Although changes in seawater pH probably have a negligible effect on Na incorporation (Allen et al., 2016), pH changes in the microenvironment of foraminiferal tests, which are mainly regulated or biased by the photosynthetic activity of their symbionts, could impact Na variations instead. The symbiont activity depends on nutrient concentrations and light intensities and thus can actively change the pH of the ambient microenvironment (Rink et al., 1998; Wolf-Gladrow et al., 1999). Higher photosynthetic rates can thereby influence the test geochemistry, which has been demonstrated by boron isotopes in


symbiont-bearing species, such as *T. sacculifer* and *O. universa* (Hönisch et al., 2003; Allen et al., 2011). In contrast, the Mg/Ca heterogeneity of these species is neither related to seawater pH (Dueñas-Bohórquez et al., 2011b) nor microenvironment pH changes due to symbiont photosynthetic/respiration activity, and instead is linked to the diurnal day/night cycles (Spero et al., 2015). Hence, as Na incorporation is presumably not affected by pH ($[CO_3^{2-}]$) (Allen et al., 2016), other factors are influencing the Na variability. The inter- and intra-specimen variations of Na/Ca in the calcite tests

may be related to the active biological control on $Na^+$ incorporation during different biomineralization processes causing ontogenetic variations, as described extensively for Mg/Ca in previous studies (Erez, 2003; Bentov and Erez, 2006; De Nooijer et al., 2014b) (see Sect. 4.4 for a detailed discussion).

However, all factors mentioned previously do not explain the differences in the amount of Na/Ca in *T. sacculifer* between our study and Allen et al. (2016), especially since specimens of both studies were collected in the Caribbean for culture

experiments. We assume that the absolute Na/Ca difference could be related to different cleaning techniques applied prior to analyses, which would need to be confirmed and quantified by additional experiments. The preparation for analysis is essential as has been exemplarily demonstrated by Rosenthal et al. (2004) for foraminiferal Mg/Ca and its reference materials. Furthermore, observed differences could also be explained by different measurement approaches. The studies of Allen et al. (2016) and Mezger et al. (2016) provided a mixed Na/Ca signal by measuring the entire newly calcified chamber

during culture (including ontogenetic, gametogenic calcite and/or spines), while we took care to avoid parts of the test with a distinct morphology and (potentially) different chemical composition. The wet chemical Na/Ca analyses (ICP-OES) with 20 to 30 specimens per surface sediment sample (Caribbean, Gulf of Guinea) match the single foraminiferal analyses by electron microprobe (Figures 4, 6), which further supports the representativeness of our data.

In addition to the parameters discussed above, which could affect and regulate $Na^+$ incorporation in living foraminifera, we

also need to address the preservation state of sodium in the tests over time. It is widely accepted that early diagenetic processes can significantly alter the geochemical composition of fossil foraminifera. For instance, planktonic foraminiferal Mg/Ca values start to decline linearly below the critical $\Delta[CO_3^{2-}]$ threshold of ~21.3 µmol kg$^{-1}$ (Regenberg et al., 2006, 2014, see Sect. 2.3 for details). Evidently, we cannot see the same trend for Na/Ca$_{T. sacculifer}$ values from Caribbean surface sediments, albeit Na/Ca seems to decline at 3.5 to 4 km water depth, below a $\Delta[CO_3^{2-}]$ level of <13 µmol kg$^{-1}$ (Figure 6b). In

the Gulf of Guinea, the critical $\Delta[CO_3^{2-}]$ threshold value is at even shallower water depth (~2.5 km), but no significant Na/Ca change with water depth is obvious, matching the Caribbean values (Table 1, Figure 6b). Therefore, Na/Ca seems to be less affected by dissolution than Mg/Ca, which could be linked to different incorporation mechanisms of both elements into foraminiferal calcite (see Sect. 4.4). The issue of Na/Ca removal due to calcite dissolution at greater water depth, nonetheless, needs further investigation, because of the limited sample set of our study.

Although the calcite saturation state is quite different in bottom waters and differs regionally in depth (Brown and Elderfield, 1996; Dekens et al., 2002; Regenberg et al., 2006, 2014), we suspect that foraminiferal Na/Ca, similar to foraminiferal Mg/Ca dissolution, is not changing above the critical calcite saturation state of >~21 µmol kg$^{-1}$ in the Caribbean and the Gulf


of Guinea at least over the last ~3000 years (ages of Caribbean and Gulf of Guinea surface sediments; Regenberg et al., 2009). Although lowest Na/Ca values of surface sediments are possibly affected by calcite dissolution at respective water depths below 3.5 km, all foraminiferal Na/Ca values match the 95% confidence interval of culture experiments, not affected by dissolution (Fig. 4, 6a). This confirms the robustness of Na/Ca as paleo-salinity proxy over time.

## 4.4 Incorporation of $Na^+$ in foraminiferal calcite

The high variations of Na/Ca between and within benthic and planktonic species (Fig. 7), and the heterogeneity within chamber walls, suggest a biological control of $Na^+$ incorporation during calcification. To examine this hypothesis, the

apparent partition coefficient for Na in organic or inorganic calcite is expressed as $D_{Na} = (Na/Ca_{foraminifer})/(Na/Ca_{ambient\ sw})$ in mmol $mol^{-1}$. Partition calculations are based in our study on $Na/Ca_{seawater} = 44.7$ mol $mol^{-1}$ as given in Delaney et al. (1985). Foraminifera in Delaney et al. (1985) were collected from the same sampling site as the cultured specimens of our study. In our study, $D_{Na}$ of *T. sacculifer* is $D_{Na} = 0.11–0.13 \cdot 10^{-3}$, which is in agreement with previously reported $D_{Na}$: Allen et al. (2016) for *T. sacculifer*, *G. ruber*: $D_{Na} = 0.1 \cdot 10^{-3}$; Wit et al. (2013) for *A. tepida*: $D_{Na} = 0.12–0.16 \cdot 10^{-3}$; Delaney et al. (1985)

for *T. sacculifer*: $D_{Na} = 0.14–0.15 \cdot 10^{-3}$. The highest partition coefficient was found in the Red Sea field calibration study from Mezger et al. (2016) for *G. ruber* and *T. sacculifer*: $D_{Na}$ of $0.18–0.25 \cdot 10^{-3}$. Notably, in inorganic calcite the partitioning of Na varies between $D_{Na} = 0.3–0.8 \cdot 10^{-3}$ (Busenberg and Plummer, 1985; White, 1978), and is therefore markedly higher compared to foraminiferal calcite. The generally two times lower foraminiferal $D_{Na}$ values compared to inorganic calcite suggest a biological control on Na incorporation in foraminiferal calcite, as it is also known for Mg incorporation (e.g.

Bentov and Erez, 2006; De Nooijer et al., 2014b). The partition coefficient of Mg in planktonic foraminiferal calcite (e.g.: $D_{Mg}$ (*T. sacculifer*, *G. ruber*) = ~$0.7–1.0 \cdot 10^{-3}$ in Allen et al. (2016), $D_{Mg}$ (*O. universa*) = 0.002 in Lea et al. (1999)) could be 5 to 10 times lower than observed for inorganic precipitated calcite (0.01) (Busenberg and Plummer, 1989). Additionally, $Mg/Ca_{foram}$ increases by 8–10 % per °C, and only 3 %/°C in inorganic calcite, demonstrating a strong biological control over any kinetic and thermodynamic influence on test Mg incorporation (Nürnberg et al., 1996; Lea et al., 1999; Allen et al.,

2016). Recent results of Allen et al. (2016) show an increase of $D_{Mg}$ with increasing calcifications rates, but not for $D_{Na}$ of *T. sacculifer* and *G. ruber*. This supports the assumption of a biological control on Na incorporation, as Busenberg and Plummer (1985) report increasing Na/Ca values with increasing calcite growth rates of inorganic calcite, thought to result from increasing lattice distortions providing space for Na incorporation.

It is known from previous studies that perforate foraminifera are able to control the elemental composition of their calcite

tests, regulated in the microenvironment of their cells (Erez, 2003; Bentov and Erez, 2006; De Nooijer et al., 2014b). Different calcification pathways could possibly bias the element transport to the calcification site via various transport mechanism, which could affect $Na^+$ incorporation. There are different mechanisms proposed where ions, needed for the calcification process, derive from. The first model of Erez (2003) suggested the uptake of ambient seawater, transported and modified via vacuoles into the cell, from which the calcification starts. Alternatively, the ions taken up by seawater

vacuolization may be stored in intracellular reservoirs before being used during calcification (Bentov and Erez, 2006; De Nooijer et al., 2014b). Foraminifera are also able to directly uptake ions from the ambient seawater by using a selective ion transport via ion pumps or channels during calcification. Just considering transport systems with $Na^+$ involved, the most abundant mechanism is a $2Na^+/Mg^{2+}$ exchanger that specifically counter transports one $Mg^{2+}$ ion across the lower gradient into the cell, in exchange with 2 $Na^+$ ions (Bentov and Erez, 2006). On the other hand, $Mg^{2+}$ ions could also be removed

from the parent solution in the cell via trans-membrane channels, using the electrochemical gradient, which is generated by a $K^+/H^+$ or $Na^+/H^+$ antiporter. These antiporter basically regulate the pH of the calcifying fluid and hence increase calcite precipitation (Erez, 2003; Bentov and Erez, 2006; De Nooijer et al., 2009; Toyofuku et al., 2017). In conclusion, $Na^+$ ions potentially have an important function for element incorporation and calcification processes in the cell of perforate foraminifera, which could strongly influence Na partitioning. Hence, biomineralization processes could be fundamentally

different between species (Bentov and Erez, 2006; De Nooijer et al., 2014b).

     The next unknown in this process is the incorporation mechanism of Na in calcite. Due to the similar size of the ionic radii of both $Na^+$ and $Ca^{2+}$, Yoshimura et al. (2017) suggested that Na is likely to directly substitute for calcium ions as well as other divalent cations like $Mg^{2+}$ or $Sr^{2+}$ in spite of the difference in charge. In this case, two $Na^+$ ions paired with anion vacancies (e.g. generated by $CO_3^{2-}$) as a possible charge compensator, would replace two $Ca^{2+}$ ions (White, 1977). Alternatively one

$Na^+$ ion accompanied by anions, such as $Cl^-$ or $SO_4^{2-}$, which induce calcite lattice distortions, would replace $Ca^{2+}$ and $CO_3^{2-}$ ions (Yoshimura et al., 2017). However, this incorporation mechanism is only described for inorganic precipitation experiments of aragonites, so far, in which alkali metals (e.g. $Na^+$, $K^+$ or $Li^+$) directly substitute for $Ca^{2+}$, whereas alkali metals occupy interstitial positions in inorganic calcite lattice defects due to a different crystal structure (Ishikawa and Ichikuni, 1984; Okumura and Kitano, 1986). This may play a role in the different Na contents of biogenic calcite, such as

foraminifera (Na: ~1000 ppm, Na/Ca: ~5 mmol mol$^{-1}$), and aragonite, such as corals (Na: ~5000 ppm, Na/Ca: ~20 mmol mol$^{-1}$) (Yoshimura et al., 2017).

     Despite the likehood that Na is not competing with Ca for lattice sites, Wit et al. (2013) suggested that $Na^+$ incorporation into foraminiferal calcite is mainly driven by the activity ratio of free $Na^+$ to free $Ca^{2+}$ in seawater, which is related to their concentrations in seawater and increases with salinity (Ishikawa and Ichikuni, 1984; Zeebe and Wolf-Gladrow, 2001). In

contrast, $Na^+$ and $Ca^{2+}$ activities in seawater are only affected by temperature changes to a limited extent, and hence temperature has a minimal influence on Na incorporation (Ishikawa and Ichikuni, 1984; Delaney et al., 1985; Zeebe and Wolf-Gladrow, 2001). Instead, lower temperatures could cause enhanced solubility of calcium carbonate, resulting in an increase of free $Ca^{2+}$ (Zeebe and Wolf-Gladrow, 2001). This could induce lower Na/Ca values in foraminiferal calcite at higher temperatures and would explain the slightly, but insignificant, negative response of Na/Ca$_{foram}$ to temperature.

## 5 Conclusions

Foraminiferal Na/Ca of cultured *T. sacculifer* correlates positively (R = 0.97, *p* <0.005) with increasing sea surface salinity between 26 to 45 salinity units: Salinity = ((Na/Ca$_{foram}$ − 0.97)/0.115). A temperature control on Na incorporation in foraminiferal calcite can be excluded by our data, supporting previous culture studies (Allen et al., 2016). In our experiments, we observed a rather low Na/Ca inter- and intra-test variability of around 2−16 %, half of that determined for Mg/Ca ($\geq$30 %) in symbiont-bearing foraminiferal species (Sadekov et al., 2005; Dueñas-Bohórquez et al., 2011b). The reduced Na/Ca variability compared to higher Mg/Ca variations implies less pronounced Na/Ca banding. As neither a relationship between Na partitioning and calcification rates, nor with changes in the carbonate chemistry (pH, ([$CO_3^{2-}$]), DIC) was observed in foraminiferal calcite (Allen et al., 2016), there is quite robust support that variations in Na/Ca$_{foram}$ are primarily driven by salinity.

According to our culture studies, a change of ±1 salinity unit would result in a Na/Ca change in *T. sacculifer* calcite of 2.25 %, equivalent to ±0.12 mmol mol$^{-1}$. This is in agreement with Na/Ca values of *T. sacculifer* selected from both the Caribbean and the Gulf of Guinea surface sediments. We observed no significant calcite dissolution effect on Na/Ca in foraminiferal calcite with changes in the calcite saturation state above 2.5 km water depth in the Gulf of Guinea and 3.5 to 4 km in the Caribbean, while Na/Ca values decline below. However, Na/Ca is not changing above the critical $\Delta$[$CO_3^{2-}$] threshold of ~21 µmol kg$^{-1}$, established for the onset of Mg/Ca dissolution in planktonic foraminifera (Regenberg et al., 2006, 2014). Further, Na/Ca values of surface sediment samples lie within the 95 % confidence interval of our salinity culture experiments, which are unaffected by calcite dissolution. Therefore, our new data support that foraminiferal Na/Ca potentially serves as a direct and reliable proxy for significant changes in ocean salinities, e.g. exceeding 0.5 salinity units.

However, the sensitivity of Na/Ca to salinity changes is species specific and possibly regional dependent, as demonstrated by the offset of 1−3 mmol mol$^{-1}$ between various planktonic and benthic species, suggesting a dominant biological control (Wit et al., 2013; Allen et al., 2016). Further improvement and proxy development could be achieved by a multi-proxy approach, e.g. combining Na/Ca paired with Mg/Ca$_{foram}$, $\delta^{18}O_{sw}$ and Ba/Ca$_{foram}$, to support the confidence of past salinity reconstructions.

### Sample and Data availability

Sample material used is archived at the GEOMAR Helmholtz Centre for Ocean Research Kiel (contact: dnuernberg@geomar.de). All data used in this study are listed in the manuscript and the appendices.

**Author contributions**

JB and DN designed this study. Data analyses and preparation of the manuscript was done by JB, assisted by DN and EH. Culture experiments described in this study were carried out by JBj. Measurements of Gulf of Guinea foraminiferal samples were accomplished by SN. GJR, EM, LDN, EH, JS, JBj and MK contributed to data interpretation and significantly improved the manuscript with intense discussions and text iterations.

**Competing interests**

The authors declare that they have no conflict of interest.

**Acknowledgments**

We gratefully thank Jan Fietzke, Mario Thöner, Thor Hansteen and Nicolaas Glock for their technical support and improving measurement techniques at the electron microprobe at GEOMAR Helmholtz Centre for Ocean Research Kiel. Special thanks are to Nadine Gehre for assistance in the laboratory. We benefited from two anonymous reviewers, which greatly improved the manuscript. Gert-Jan Reichart and Lennart de Nooijer acknowledge NESSC funding. Jacqueline Bertlich was funded by a PhD fellowship of the Helmholtz Research School for Ocean System Science and Technology (HOSST) at GEOMAR Helmholtz Centre for Ocean Research Kiel (VH-KO-601).

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

**Tables**

**Table 1.** Na/Ca values of *T. sacculifer* from surface sediment samples, obtained from different core locations in the Gulf of Guinea[a] (Lutz et al., 1988) and the Caribbean[b] (Nürnberg et al., 2003). Foraminifera were selected from the uppermost sediment layer (0−1 cm). The salinity and temperature data present averaged annual values at 30 m water depth with ±variations of these parameters between 0−60 m water depth (WOA13), shown as the standard error of the mean, SEM ($\sigma/\sqrt{n}$). Na/Ca values[a,b] were measured by an ICP−OES (±SEM of replicated measurements). Calcite saturation states $\Delta[CO_3^{2-}]$ at bottom waters of respective sampling locations were taken from Regenberg
et al. (2006, Caribbean) and Weldeab (2012, Gulf of Guinea). Additional data were calculated with the program CO2SYS by Pierrot et al. (2006) (see Sect. 2.3 for detailed description). Radiocarbon (AMS[14]C) ages before present (BP) are published in Regenberg et al. (2009).

| Cruise Station | Latitude (°N) | Longitude (°E) | Water depth (m) | $\Delta[CO_3^{2-}]$ ($\mu$mol kg$^{-1}$) | [14]C Age years BP | Salinity | Temperature (°C) | Na/Ca$_{average}$ (mmol mol$^{-1}$) |
|---|---|---|---|---|---|---|---|---|
| ***Gulf of Guinea (M6−5)[a]*** | | | | | | | | |
| GIK 16808−2 | 3° 48.6' | 2° 51.5' | 2234 | 44.3 | | 35.31 ±0.17 | 25.20 ±1.32 | 4.92 ±0.09 |
| GIK 16860−1 | 3° 43.7' | −6° 29.9' | 202 | 81.2 | | 34.96 ±0.62 | 25.42 ±1.35 | 5.01 ±0.07 |
| GIK 16864−1 | 3° 09.3' | −6° 17.1' | 1495 | 65.5 | | 34.98 ±0.54 | 25.72 ±1.34 | 4.92 ±0.10 |
| GIK 16865−1 | 2° 40.3' | −6° 03.3' | 2492 | 35.5 | | 35.01 ±0.45 | 25.87 ±1.32 | 4.77 ±0.03 |
| GIK 16868−2 | −0° 13.7' | −6° 01.1' | 1284 | 37.7 | | 35.51 ±0.31 | 23.65 ±1.21 | 4.92 ±0.14 |
| GIK 16869−1 | −0° 12.7' | −6° 00.8' | 1837 | 41.8 | | 35.49 ±0.32 | 23.90 ±1.23 | 5.12 ±0.12 |
| ***Caribbean (SO164)[b]*** | | | | | | | | |
| 01−3 | 13° 50.195' | 74° 09.028' | 4026 | 13.3 | | 35.97 ±0.05 | 27.23 ±0.10 | 4.81 ±0.28 |
| 02−3 | 15° 18.29' | 72° 47.06' | 2977 | 28.6 | 2205 ±25 | 35.71 ±0.05 | 27.45 ±0.05 | 5.24 ±0.30 |
| 03−3 | 16° 32.40' | 72° 12.31' | 2744 | 34.1 | | 35.63 ±0.06 | 27.50 ±0.04 | 5.54 ±0.27 |
| 04−2 | 17° 16.38' | 71° 39.09' | 1013 | 42.4 | | 35.50 ±0.06 | 27.53 ±0.03 | 5.32 ±0.37 |

| | | | | | | | | |
|---|---|---|---|---|---|---|---|---|
| 07−3 | 21° 19.46' | 74° 08.76' | 2722 | 34.1 | 720 ±35 | 36.37 ±0.02 | 26.96 ±0.15 | 5.67 ±0.17 |
| 18−1 | 21° 13.61' | 74° 21.0' | 1629 | 45.3 | 1350 ±25 | 36.27 ±0.03 | 27.09 ±0.15 | 5.68 ±0.30 |
| 20−2 | 16° 45.49' | 71° 29.22' | 3357 | 28.4 | | 35.57 ±0.05 | 27.49 ±0.05 | 5.13 ±0.26 |
| 21−3 | 16° 06.0' | 70° 30.0' | 3995 | 15.6 | | 35.63 ±0.06 | 27.49 ±0.06 | 5.06 ±0.30 |
| 22−2 | 15° 24.0' | 68° 12.0' | 4506 | 3.4 | | 35.60 ±0.08 | 27.61 ±0.07 | 4.79 ±0.18 |
| 23−3 | 15° 34.01' | 65° 08.09' | 4328 | 6.8 | | 35.42 ±0.13 | 27.60 ±0.08 | 4.64 ±0.25 |
| 24−3 | 14° 11.89' | 63° 25.43' | 1545 | 47.1 | | 35.37 ±0.15 | 27.55 ±0.12 | 5.51 ±0.33 |
| 25−3 | 14° 41.25' | 59° 44.48' | 2720 | 40.5 | 1915 ±30 | 35.45 ±0.16 | 27.45 ±0.10 | 5.32 ±0.25 |
| 48−2 | 15° 57.02' | 60° 55.0' | 1286 | 49.6 | | 35.59 ±0.14 | 27.34 ±0.09 | 5.40 ±0.56 |
| 50−3 | 15° 21.25' | 59° 16.94' | 4002 | 15.4 | | 35.55 ±0.15 | 27.39 ±0.10 | 5.19 ±0.17 |

[a] Foraminifera were handpicked from the 300−400 µm size fraction. Each Na/Ca value derives from the average of 30 specimens and is the mean of 3−5 repeated ICP−OES measurements.

[b] Foraminifera were selected from the 315−400 µm size fraction. Every ICP−OES measurement was repeated 5 times. One sample contains 20−30 specimens.

**Table 2.** Reference materials used for electron microprobe analysis. Precision (Prec.) and accuracy (Accu.) is expressed as the relative standard deviation (RSD) and the relative error in %. Measurements were performed with 10 µm x 10 µm maps (number of pixels: n = 100).

| Standard | Reference | Ca (wt%) | Prec. (%) | Accu. (%) | Na (wt%) | Prec. (%) | Accu. (%) |
|---|---|---|---|---|---|---|---|
| Calcite (USNM 136321) | Jarosewich, 2002 | 40.11 | 0.27 | 0.02 | − | − | − |
| Dolomite (USNM 10057) | Jarosewich, 2002 | 21.85 | 1.97 | 0.55 | − | − | − |
| Albite (131705, Natural Plagioclase Feldspar) | Amelia Co., Virginia, USA | − | − | − | 8.71 | 0.17 | -0.95 |
| Glass, Basalt (USNM 111240 VG-2) | Jarosewich, 2002 | 7.95 | 2.16 | 4.75 | 1.94 | 1.2 | -11.66 |
| Modern Coral A2 | Ohde et al., 2003 | 37.84 | 0.83 | 2.56 | 0.47 | 8.73 | -25.78 |

**Table 3.** Individual Na/Ca values of cultured *T. sacculifer* under variable salinity and constant temperature (26.5 °C ±0.25 °C), measured by the electron microprobe. Each value results from maps, ranging in size from 5 to 20 µm, on inner chamber wall cross sections of newly grown calcite (F, F-1) during the experiments. Each pixel represents 1 µm x 1 µm. Averaged values comprise all Na/Ca values of one single foraminifer. Sample labels correspond to those used in Nürnberg et al. (1996). Chambers grown in the natural environment before culturing are marked with F-3 and F-4 and are averaged separately from those grown in culture. Annual salinity data were taken from WOA13 (Zweng et al., 2013). Individual and averaged Na/Ca uncertainty values are based on ±standard error of the mean ($\sigma/\sqrt{n}$).

| Sample # | Salinity | Pixel n | Na/Ca$_{individual}$ (mmol mol$^{-1}$) | Na/Ca$_{average}$ (mmol mol$^{-1}$) | Sample # | Salinity | Pixel n | Na/Ca$_{individual}$ (mmol mol$^{-1}$) | Na/Ca$_{average}$ (mmol mol$^{-1}$) |
|---|---|---|---|---|---|---|---|---|---|
| 7912 | 26 | 35 | 4.77 ±0.22 | **4.36 ±0.20** | 7703 | 44 | 21 | 5.40 ±0.25 | |
| 7912 | 26 | 25 | 4.15 ±0.20 | | 7703 | 44 | 24 | 4.62 ±0.22 | |
| 7912 | 26 | 25 | 3.73 ±0.25 | | 7703 | 44 | 18 | 3.96 ±0.27 | |
| 7912 | 26 | 25 | 4.82 ±0.39 | | 7703 | 44 | 15 | 4.62 ±0.47 | |

| | | | | | | | | | |
|---|---|---|---|---|---|---|---|---|---|
| 7912 | 26 | 72 | 4.34 ±0.16 | | 7703 | 44 | 20 | 4.53 ±0.40 | |
| | | | | | 7703 | 44 | 19 | 5.32 ±0.34 | |
| 7913 | 26 | 144 | 4.36 ±0.09 | **3.96 ±0.14** | | | | | |
| 7913 | 26 | 24 | 3.84 ±0.31 | | 7704 | 44 | 72 | 5.51 ±0.19 | **5.27 ±0.16** |
| 7913 | 26 | 36 | 4.18 ±0.23 | | 7704 | 44 | 43 | 4.36 ±0.16 | |
| 7913 | 26 | 33 | 3.51 ±0.22 | | 7704 | 44 | 42 | 6.50 ±0.33 | |
| 7913 | 26 | 36 | 3.27 ±0.16 | | 7704 | 44 | 91 | 5.39 ±0.13 | |
| 7913 | 26 | 35 | 3.88 ±0.16 | | 7704 | 44 | 21 | 5.12 ±0.27 | |
| 7913 | 26 | 20 | 4.34 ±0.37 | | 7704 | 44 | 22 | 4.83 ±0.24 | |
| 7913 | 26 | 21 | 4.28 ±0.24 | | 7704 | 44 | 21 | 4.43 ±0.34 | |
| | | | | | 7704 | 44 | 20 | 5.45 ±0.34 | |
| 7914 | 26 | 94 | 3.81 ±0.16 | **3.86 ±0.17** | 7704 | 44 | 22 | 5.62 ±0.45 | |
| 7914 | 26 | 48 | 4.04 ±0.20 | | 7704 | 44 | 48 | 4.85 ±0.20 | |
| 7914 | 26 | 47 | 4.71 ±0.28 | | 7704 | 44 | 45 | 5.64 ±0.21 | |
| 7914 | 26 | 35 | 3.56 ±0.20 | | 7704 | 44 | 48 | 5.43 ±0.20 | |
| 7914 | 26 | 30 | 4.04 ±0.31 | | 7704 | 44 | 105 | 5.37 ±0.14 | |
| 7914 | 26 | 34 | 3.42 ±0.19 | | | | | | |
| 7914 | 26 | 25 | 3.46 ±0.19 | | 8301 | 45 | 47 | 5.91 ±0.25 | **6.40 ±0.19** |
| | | | | | 8301 | 45 | 49 | 6.15 ±0.25 | |
| 8135 | 41 | 23 | 5.97 ±0.33 | **5.66 ±0.20** | 8301 | 45 | 47 | 6.20 ±0.28 | |
| 8135 | 41 | 18 | 4.97 ±0.33 | | 8301 | 45 | 44 | 6.71 ±0.31 | |
| 8135 | 41 | 19 | 4.74 ±0.48 | | 8301 | 45 | 46 | 6.02 ±0.20 | |
| 8135 | 41 | 22 | 5.61 ±0.42 | | 8301 | 45 | 45 | 5.90 ±0.21 | |
| 8135 | 41 | 135 | 6.18 ±0.16 | | 8301 | 45 | 131 | 6.39 ±0.13 | |
| 8135 | 41 | 143 | 5.01 ±0.10 | | 8301 | 45 | 43 | 5.62 ±0.21 | |
| 8135 | 41 | 140 | 4.83 ±0.09 | | 8301 | 45 | 100 | 7.55 ±0.13 | |
| 8135 | 41 | 22 | 4.24 ±0.32 | | 8301 | 45 | 100 | 7.68 ±0.18 | |
| 8135 | 41 | 45 | 6.33 ±0.25 | | 8301 | 45 | 186 | 6.84 ±0.12 | |
| 8135 | 41 | 70 | 6.46 ±0.24 | | 8301 | 45 | 98 | 5.87 ±0.17 | |
| 8135 | 41 | 74 | 6.20 ±0.21 | | | | | | |
| 8135 | 41 | 69 | 6.12 ±0.30 | | | | | | |
| 8135 | 41 | 79 | 6.48 ±0.29 | | | | | | |
| 8135 | 41 | 45 | 6.04 ±0.22 | | 4.1 | 35.9 | 219 | 5.16 ±0.12 | **5.27 ±0.12** |
| | | | | | 4.1 | 35.9 | 263 | 4.96 ±0.08 | |
| 8137 | 41 | 24 | 5.04 ±0.25 | **5.49 ±0.16** | 4.1 | 35.9 | 257 | 5.06 ±0.08 | |
| 8137 | 41 | 19 | 4.84 ±0.29 | | 4.1 | 35.9 | 151 | 5.53 ±0.14 | |
| 8137 | 41 | 22 | 5.46 ±0.29 | | 4.1 | 35.9 | 195 | 5.73 ±0.10 | |
| 8137 | 41 | 22 | 4.30 ±0.21 | | 4.1 | 35.9 | 195 | 5.18 ±0.09 | |
| 8137 | 41 | 21 | 5.20 ±0.28 | | | | | | |
| 8137 | 41 | 23 | 5.43 ±0.34 | | 2.3 | 35.9 | 105 | 5.49 ±0.14 | **5.43 ±0.18** |
| 8137 | 41 | 64 | 5.85 ±0.18 | | 2.3 | 35.9 | 58 | 5.55 ±0.17 | |
| 8137 | 41 | 21 | 5.32 ±0.40 | | 2.3 | 35.9 | 65 | 5.51 ±0.19 | |
| 8137 | 41 | 23 | 5.33 ±0.29 | | 2.3 | 35.9 | 72 | 5.47 ±0.18 | |
| 8137 | 41 | 25 | 6.47 ±0.49 | | 2.3 | 35.9 | 98 | 5.38 ±0.19 | |
| 8137 | 41 | 21 | 5.20 ±0.23 | | 2.3 | 35.9 | 108 | 5.05 ±0.12 | |
| 8137 | 41 | 24 | 6.45 ±0.30 | | 2.3 | 35.9 | 66 | 5.55 ±0.22 | |
| 8137 | 41 | 20 | 6.09 ±0.35 | | | | | | |
| 8137 | 41 | 23 | 5.88 ±0.35 | | 7704 | 35.9 | 109 | 5.43 ±0.14 | **5.48 ±0.04** |
| | | | | | 7704 | 35.9 | 68 | 5.60 ±0.17 | |
| 8138 | 41 | 46 | 4.41 ±0.21 | **5.08 ±0.15** | 7704 | 35.9 | 99 | 5.47 ±0.13 | |
| 8138 | 41 | 46 | 5.42 ±0.35 | | 7704 | 35.9 | 72 | 5.51 ±0.15 | |
| 8138 | 41 | 49 | 5.16 ±0.26 | | 7704 | 35.9 | 78 | 5.33 ±0.15 | |
| 8138 | 41 | 49 | 5.40 ±0.29 | | 7704 | 35.9 | 96 | 5.51 ±0.12 | |

In situ grown chambers (F-3, F-4)

| Sample | T | Pixel | Na/Ca_individual | Na/Ca_average | Sample | T | Pixel | Na/Ca_individual | Na/Ca_average |
|---|---|---|---|---|---|---|---|---|---|
| 8138 | 41 | 48 | 4.90 ±0.20 | | | | | | |
| 8138 | 41 | 49 | 5.16 ±5.19 | | 7703 | 35.9 | 21 | 5.40 ±0.25 | **5.34 ±0.08** |
| | | | | | 7703 | 35.9 | 24 | 5.10 ±0.24 | |
| 7703 | 44 | 19 | 5.03 ±0.40 | **4.83 ±0.18** | 7703 | 35.9 | 45 | 5.16 ±0.18 | |
| 7703 | 44 | 24 | 4.60 ±0.33 | | 7703 | 35.9 | 39 | 5.58 ±0.18 | |
| 7703 | 44 | 46 | 6.06 ±0.24 | | 7703 | 35.9 | 62 | 5.57 ±0.17 | |
| 7703 | 44 | 23 | 4.72 ±0.26 | | 7703 | 35.9 | 66 | 5.25 ±0.17 | |
| 7703 | 44 | 24 | 4.23 ±0.25 | | | | | | |


**Table 4.** Individual Na/Ca values of cultured *T. sacculifer* under variable temperatures and a constant salinity of 36, measured by the electron microprobe. Each value results from maps, ranging in size from 5 to 20 µm, on inner chamber wall cross sections of newly grown calcite (F, F-1) during the experiments. Each pixel represents 1µm x 1 µm. Averaged values comprise all Na/Ca values of one single foraminifer. Sample labels correspond to those used in Nürnberg et al. (1996). Individual and averaged Na/Ca values are based on ±standard error of the mean ($\sigma/\sqrt{n}$).


| Sample # | T (°C) | Pixel n | Na/Ca_individual (mmol mol$^{-1}$) | Na/Ca_average (mmol mol$^{-1}$) | Sample # | T (°C) | Pixel n | Na/Ca_individual (mmol mol$^{-1}$) | Na/Ca_average (mmol mol$^{-1}$) |
|---|---|---|---|---|---|---|---|---|---|
| 1.2 | 19.5 | 36 | 4.47 ±0.34 | **4.24 ±0.19** | 3.1 | 26.5 | 46 | 4.69 ±0.17 | **4.62 ±0.15** |
| 1.2 | 19.5 | 44 | 4.64 ±0.43 | | 3.1 | 26.5 | 48 | 4.26 ±0.18 | |
| 1.2 | 19.5 | 47 | 3.75 ±0.17 | | 3.1 | 26.5 | 48 | 4.57 ±0.25 | |
| 1.2 | 19.5 | 48 | 4.04 ±0.20 | | 3.1 | 26.5 | 45 | 4.97 ±0.22 | |
| 1.2 | 19.5 | 24 | 3.49 ±0.24 | | | | | | |
| 1.2 | 19.5 | 15 | 4.11 ±0.35 | | 3.2 | 26.5 | 13 | 4.76 ±0.24 | **4.86 ±0.10** |
| 1.2 | 19.5 | 19 | 5.06 ±0.46 | | 3.2 | 26.5 | 15 | 4.54 ±0.37 | |
| 1.2 | 19.5 | 19 | 5.17 ±0.28 | | 3.2 | 26.5 | 100 | 4.71 ±0.24 | |
| 1.2 | 19.5 | 47 | 4.68 ±0.34 | | 3.2 | 26.5 | 100 | 5.20 ±0.19 | |
| 1.2 | 19.5 | 48 | 3.80 ±0.17 | | 3.2 | 26.5 | 49 | 4.32 ±0.17 | |
| 1.2 | 19.5 | 47 | 3.68 ±0.22 | | 3.2 | 26.5 | 45 | 4.89 ±0.21 | |
| | | | | | 3.2 | 26.5 | 45 | 4.91 ±0.16 | |
| 1.3 | 19.5 | 25 | 4.43 ±0.30 | **4.78 ±0.24** | 3.2 | 26.5 | 49 | 4.80 ±0.20 | |
| 1.3 | 19.5 | 24 | 4.47 ±0.32 | | 3.2 | 26.5 | 48 | 4.59 ±0.18 | |
| 1.3 | 19.5 | 24 | 4.58 ±0.32 | | | | | | |
| 1.3 | 19.5 | 22 | 4.43 ±0.31 | | 3.4 | 26.5 | 100 | 5.40 ±0.20 | **5.00 ±0.07** |
| 1.3 | 19.5 | 16 | 4.83 ±0.35 | | 3.4 | 26.5 | 100 | 4.98 ±0.13 | |
| 1.3 | 19.5 | 15 | 5.95 ±0.87 | | 3.4 | 26.5 | 49 | 4.72 ±0.22 | |
| | | | | | 3.4 | 26.5 | 45 | 5.11 ±0.23 | |
| 1.4 | 19.5 | 19 | 4.76 ±0.42 | **5.09 ±0.20** | 3.4 | 26.5 | 49 | 5.09 ±0.24 | |
| 1.4 | 19.5 | 13 | 4.07 ±0.56 | | 3.4 | 26.5 | 47 | 4.92 ±0.19 | |
| 1.4 | 19.5 | 20 | 5.97 ±0.61 | | 3.4 | 26.5 | 49 | 4.81 ±0.19 | |
| 1.4 | 19.5 | 22 | 4.39 ±0.31 | | 3.4 | 26.5 | 20 | 4.89 ±0.32 | |
| 1.4 | 19.5 | 45 | 4.68 ±0.17 | | 3.4 | 26.5 | 22 | 5.06 ±0.34 | |
| 1.4 | 19.5 | 48 | 5.32 ±0.25 | | | | | | |
| 1.4 | 19.5 | 49 | 5.07 ±0.28 | | 3.5 | 26.5 | 112 | 5.46 ±0.13 | **5.49 ±0.05** |
| 1.4 | 19.5 | 121 | 5.08 ±0.12 | | 3.5 | 26.5 | 102 | 5.38 ±0.15 | |
| 1.4 | 19.5 | 49 | 5.49 ±0.25 | | 3.5 | 26.5 | 75 | 5.62 ±0.21 | |
| 1.4 | 19.5 | 22 | 6.03 ±0.49 | | 3.5 | 26.5 | 89 | 5.51 ±0.14 | |
| 2.1 | 23.5 | 20 | 4.25 ±0.25 | **4.74 ±0.16** | 4.1 | 29.5 | 100 | 4.86 ±0.14 | **4.80 ±0.08** |

| Sample # | T (°C) | Pixel n | Na/Ca$_{individual}$ (mmol mol$^{-1}$) | Na/Ca$_{average}$ (mmol mol$^{-1}$) | Sample # | T (°C) | Pixel n | Na/Ca$_{individual}$ (mmol mol$^{-1}$) | Na/Ca$_{average}$ (mmol mol$^{-1}$) |
|---|---|---|---|---|---|---|---|---|---|
| 2.1 | 23.5 | 24 | 4.83 ±0.27 | | 4.1 | 29.5 | 48 | 4.60 ±0.17 | |
| 2.1 | 23.5 | 23 | 5.06 ±0.22 | | 4.1 | 29.5 | 42 | 5.11 ±0.25 | |
| 2.1 | 23.5 | 24 | 5.06 ±0.23 | | 4.1 | 29.5 | 48 | 4.91 ±0.19 | |
| 2.1 | 23.5 | 21 | 4.51 ±0.31 | | 4.1 | 29.5 | 49 | 4.64 ±0.13 | |
| | | | | | 4.1 | 29.5 | 23 | 4.39 ±0.24 | |
| 2.2 | 23.5 | 149 | 5.65 ±0.13 | **5.41 ±0.06** | 4.1 | 29.5 | 25 | 4.60 ±0.32 | |
| 2.2 | 23.5 | 100 | 5.40 ±0.14 | | 4.1 | 29.5 | 64 | 4.97 ±0.15 | |
| 2.2 | 23.5 | 100 | 5.15 ±0.11 | | 4.1 | 29.5 | 64 | 4.70 ±0.17 | |
| 2.2 | 23.5 | 142 | 5.63 ±0.14 | | 4.1 | 29.5 | 104 | 5.22 ±0.18 | |
| 2.2 | 23.5 | 178 | 5.39 ±0.10 | | | | | | |
| 2.2 | 23.5 | 149 | 5.17 ±0.10 | | 4.3 | 29.5 | 16 | 4.06 ±0.35 | **3.96 ±0.23** |
| 2.2 | 23.5 | 97 | 5.57 ±0.15 | | 4.3 | 29.5 | 19 | 3.73 ±0.20 | |
| 2.2 | 23.5 | 98 | 5.23 ±0.16 | | 4.3 | 29.5 | 17 | 3.60 ±0.21 | |
| 2.2 | 23.5 | 99 | 5.46 ±0.15 | | 4.3 | 29.5 | 17 | 3.62 ±0.22 | |
| | | | | | 4.3 | 29.5 | 23 | 4.80 ±0.23 | |
| 2.3 | 23.5 | 25 | 4.53 ±0.23 | **4.99 ± 0.33** | | | | | |
| 2.3 | 23.5 | 25 | 4.05 ±0.27 | | 4.4 | 29.5 | 25 | 4.05 ±0.27 | **4.26 ±0.12** |
| 2.3 | 23.5 | 25 | 4.45 ±0.17 | | 4.4 | 29.5 | 25 | 3.93 ±0.25 | |
| 2.3 | 23.5 | 47 | 6.19 ±0.38 | | 4.4 | 29.5 | 25 | 4.33 ±0.22 | |
| 2.3 | 23.5 | 49 | 5.66 ±0.21 | | 4.4 | 29.5 | 49 | 4.26 ±0.23 | |
| 2.3 | 23.5 | 48 | 5.05 ±0.16 | | 4.4 | 29.5 | 48 | 3.86 ±0.16 | |
| | | | | | 4.4 | 29.5 | 45 | 3.94 ±0.18 | |
| 2.4 | 23.5 | 24 | 4.52 ±0.29 | **4.51 ± 0.37** | 4.4 | 29.5 | 49 | 4.34 ±0.15 | |
| 2.4 | 23.5 | 48 | 5.03 ±0.23 | | 4.4 | 29.5 | 96 | 4.82 ±0.15 | |
| 2.4 | 23.5 | 42 | 4.22 ±0.19 | | 4.4 | 29.5 | 104 | 4.84 ±0.13 | |
| 2.4 | 23.5 | 44 | 4.28 ±0.20 | | | | | | |

**Table 5.** Individual Na/Ca values of cultured *T. sacculifer* under variable temperatures and a constant salinity of 33, measured by the electron microprobe. Each value results from maps, ranging in size from 5 to 20 μm, on inner chamber wall cross sections of newly grown calcite (F, F-1) during the experiments. Each pixel represents 1μm x 1 μm. Averaged values comprise all Na/Ca values of one single foraminifer. Sample labels correspond to those used in Nürnberg et al. (1996). Individual and averaged Na/Ca values are based on ±standard error of the mean (σ/√n).

| Sample # | T (°C) | Pixel n | Na/Ca$_{individual}$ (mmol mol$^{-1}$) | Na/Ca$_{average}$ (mmol mol$^{-1}$) | Sample # | T (°C) | Pixel n | Na/Ca$_{individual}$ (mmol mol$^{-1}$) | Na/Ca$_{average}$ (mmol mol$^{-1}$) |
|---|---|---|---|---|---|---|---|---|---|
| 5.1 | 19.5 | 34 | 5.29 ±0.26 | **5.37 ±0.10** | 7.1 | 26.5 | 23 | 3.97 ±0.24 | |
| 5.1 | 19.5 | 42 | 5.41 ±0.22 | | 7.1 | 26.5 | 23 | 4.46 ±0.38 | |
| 5.1 | 19.5 | 48 | 5.64 ±0.19 | | 7.1 | 26.5 | 20 | 4.52 ±0.29 | |
| 5.1 | 19.5 | 39 | 5.27 ±0.20 | | 7.1 | 26.5 | 20 | 4.54 ±0.23 | |
| 5.1 | 19.5 | 33 | 5.40 ±0.26 | | 7.1 | 26.5 | 17 | 4.48 ±0.28 | |
| 5.1 | 19.5 | 33 | 5.65 ±0.26 | | 7.1 | 26.5 | 22 | 4.60 ±0.29 | |
| 5.1 | 19.5 | 20 | 4.90 ±0.35 | | 7.1 | 26.5 | 24 | 4.18 ±0.29 | |
| | | | | | | | | | |
| 5.2 | 19.5 | 63 | 4.91 ±0.17 | **5.13 ±0.15** | 7.3 | 26.5 | 100 | 5.00 ±0.18 | **5.05 ±0.09** |
| 5.2 | 19.5 | 63 | 4.90 ±0.15 | | 7.3 | 26.5 | 47 | 5.05 ±0.20 | |
| 5.2 | 19.5 | 100 | 5.69 ±0.15 | | 7.3 | 26.5 | 23 | 5.23 ±0.26 | |
| 5.2 | 19.5 | 99 | 5.62 ±0.16 | | 7.3 | 26.5 | 43 | 4.64 ±0.19 | |
| 5.2 | 19.5 | 21 | 4.57 ±0.29 | | 7.3 | 26.5 | 85 | 5.45 ±0.15 | |
| 5.2 | 19.5 | 21 | 4.76 ±0.27 | | 7.3 | 26.5 | 46 | 4.81 ±0.19 | |

| | | | | |
|---|---|---|---|---|
| 5.2 | 19.5 | 20 | 4.56 ±0.21 | |
| 5.2 | 19.5 | 19 | 5.64 ±0.47 | |
| 5.2 | 19.5 | 16 | 4.70 ±0.32 | |
| 5.2 | 19.5 | 17 | 5.99 ±0.39 | |
| 5.2 | 19.5 | 23 | 5.13 ±0.25 | |
| 5.5 | 19.5 | 46 | 4.56 ±0.20 | **5.05 ±0.09** |
| 5.5 | 19.5 | 37 | 5.06 ±0.32 | |
| 5.5 | 19.5 | 40 | 5.35 ±0.20 | |
| 5.5 | 19.5 | 42 | 5.01 ±0.19 | |
| 5.5 | 19.5 | 23 | 5.42 ±0.29 | |
| 5.5 | 19.5 | 17 | 5.03 ±0.43 | |
| 5.5 | 19.5 | 25 | 5.07 ±0.32 | |
| 5.5 | 19.5 | 25 | 4.89 ±0.27 | |
| 6.1 | 23.5 | 85 | 5.68 ±0.17 | **5.53 ±0.07** |
| 6.1 | 23.5 | 92 | 5.42 ±0.17 | |
| 6.1 | 23.5 | 89 | 5.73 ±0.17 | |
| 6.1 | 23.5 | 43 | 5.13 ±0.20 | |
| 6.1 | 23.5 | 24 | 5.71 ±0.38 | |
| 6.1 | 23.5 | 45 | 5.46 ±0.18 | |
| 6.1 | 23.5 | 79 | 5.57 ±0.17 | |
| 6.1 | 23.5 | 98 | 5.55 ±0.13 | |
| 6.2 | 23.5 | 54 | 4.58 ±0.16 | **4.53 ±0.12** |
| 6.2 | 23.5 | 21 | 4.74 ±0.24 | |
| 6.2 | 23.5 | 25 | 4.39 ±0.24 | |
| 6.2 | 23.5 | 25 | 4.80 ±0.21 | |
| 6.2 | 23.5 | 25 | 4.27 ±0.24 | |
| 6.2 | 23.5 | 25 | 4.04 ±0.28 | |
| 6.2 | 23.5 | 24 | 4.86 ±0.24 | |
| 6.3 | 23.5 | 18 | 5.24 ±0.21 | **5.16 ±0.21** |
| 6.3 | 23.5 | 18 | 5.10 ±0.28 | |
| 6.3 | 23.5 | 17 | 4.67 ±0.31 | |
| 6.3 | 23.5 | 21 | 4.86 ±0.40 | |
| 6.3 | 23.5 | 20 | 5.92 ±0.39 | |
| 6.5 | 23.5 | 47 | 6.05 ±0.24 | **5.78 ±0.11** |
| 6.5 | 23.5 | 41 | 5.47 ±0.20 | |
| 6.5 | 23.5 | 42 | 6.12 ±0.20 | |
| 6.5 | 23.5 | 35 | 6.02 ±0.22 | |
| 6.5 | 23.5 | 44 | 5.71 ±0.25 | |
| 6.5 | 23.5 | 24 | 6.12 ±0.24 | |
| 6.5 | 23.5 | 44 | 5.17 ±0.22 | |
| 6.5 | 23.5 | 98 | 6.14 ±0.14 | |
| 6.5 | 23.5 | 134 | 5.97 ±0.12 | |
| 6.5 | 23.5 | 93 | 5.25 ±0.12 | |
| 6.5 | 23.5 | 93 | 5.59 ±0.14 | |
| 7.1 | 26.5 | 22 | 3.90 ±0.24 | **4.33 ±0.10** |

| | | | | |
|---|---|---|---|---|
| 7.3 | 26.5 | 45 | 5.09 ±0.21 | |
| 7.3 | 26.5 | 44 | 5.14 ±0.23 | |
| 7.4 | 26.5 | 90 | 5.24 ±0.15 | **5.05 ±0.14** |
| 7.4 | 26.5 | 41 | 4.60 ±0.16 | |
| 7.4 | 26.5 | 52 | 4.87 ±0.19 | |
| 7.4 | 26.5 | 46 | 4.99 ±0.23 | |
| 7.4 | 26.5 | 43 | 5.04 ±0.19 | |
| 7.4 | 26.5 | 45 | 4.86 ±0.19 | |
| 7.4 | 26.5 | 96 | 5.75 ±0.16 | |
| 7.5 | 26.5 | 19 | 4.19 ±0.34 | **4.61 ±0.20** |
| 7.5 | 26.5 | 25 | 4.83 ±0.30 | |
| 7.5 | 26.5 | 14 | 5.02 ±0.27 | |
| 7.5 | 26.5 | 17 | 4.68 ±0.35 | |
| 7.5 | 26.5 | 17 | 5.06 ±0.47 | |
| 7.5 | 26.5 | 16 | 3.85 ±0.32 | |
| 8.1 | 29.5 | 24 | 4.98 ±0.24 | **4.39 ±0.13** |
| 8.1 | 29.5 | 24 | 4.12 ±0.27 | |
| 8.1 | 29.5 | 22 | 4.66 ±0.31 | |
| 8.1 | 29.5 | 24 | 4.35 ±0.29 | |
| 8.1 | 29.5 | 20 | 3.90 ±0.20 | |
| 8.1 | 29.5 | 18 | 4.58 ±0.25 | |
| 8.1 | 29.5 | 22 | 4.52 ±0.26 | |
| 8.1 | 29.5 | 16 | 4.00 ±0.44 | |
| 8.3 | 29.5 | 15 | 3.95 ±0.36 | **4.10 ±0.13** |
| 8.3 | 29.5 | 14 | 3.67 ±0.31 | |
| 8.3 | 29.5 | 17 | 4.14 ±0.25 | |
| 8.3 | 29.5 | 15 | 3.96 ±0.23 | |
| 8.3 | 29.5 | 14 | 3.79 ±0.29 | |
| 8.3 | 29.5 | 45 | 4.13 ±0.19 | |
| 8.3 | 29.5 | 50 | 3.82 ±0.15 | |
| 8.3 | 29.5 | 45 | 4.63 ±0.22 | |
| 8.3 | 29.5 | 44 | 4.78 ±0.17 | |
| 8.4 | 29.5 | 47 | 4.80 ±0.17 | **4.43 ±0.14** |
| 8.4 | 29.5 | 38 | 4.68 ±0.18 | |
| 8.4 | 29.5 | 35 | 4.48 ±0.22 | |
| 8.4 | 29.5 | 20 | 4.04 ±0.23 | |
| 8.4 | 29.5 | 23 | 4.11 ±0.28 | |
| 8.4 | 29.5 | 17 | 3.93 ±0.31 | |
| 8.4 | 29.5 | 42 | 4.33 ±0.38 | |
| 8.4 | 29.5 | 23 | 5.04 ±0.35 | |
| 8.5 | 29.5 | 61 | 4.81 ±0.16 | **5.31 ±0.21** |
| 8.5 | 29.5 | 63 | 5.14 ±0.18 | |
| 8.5 | 29.5 | 59 | 5.57 ±0.20 | |
| 8.5 | 29.5 | 42 | 5.72 ±0.26 | |

**Figures**

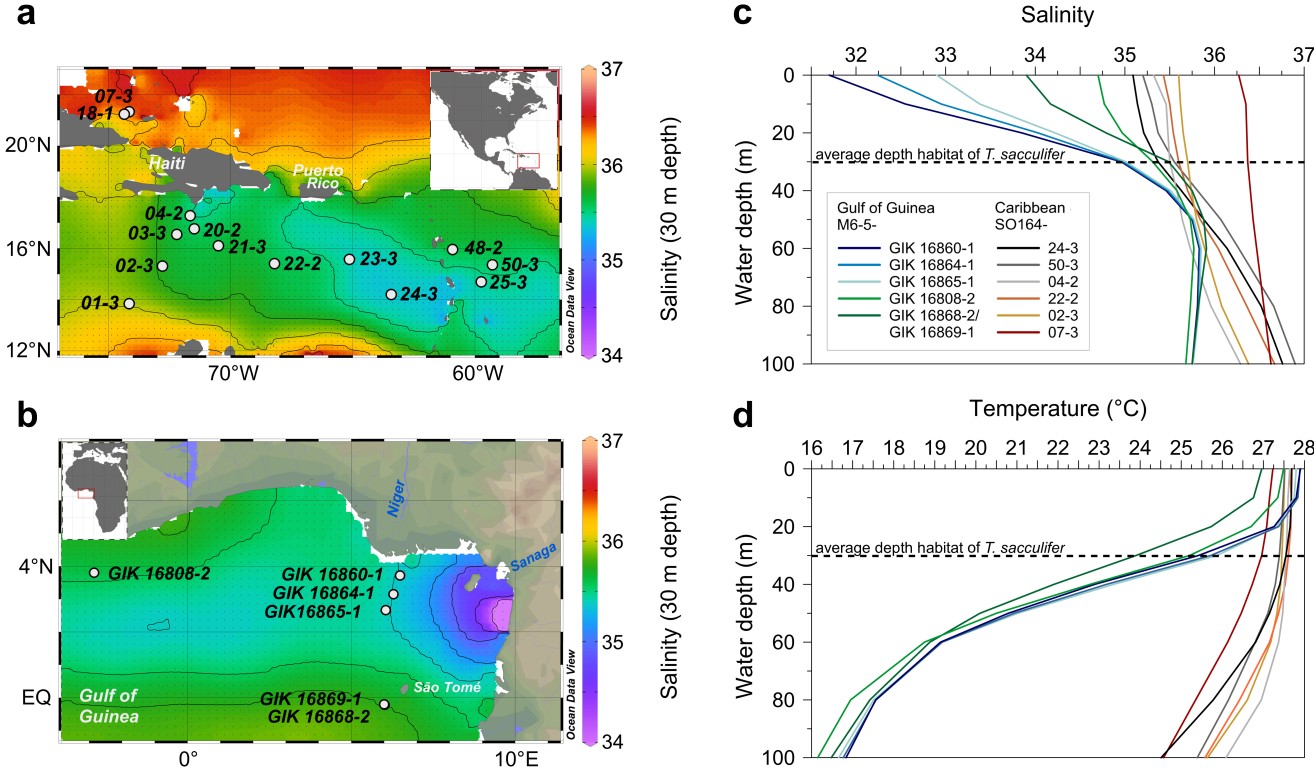

**Figure 1.** Sediment surface samples used for Na/Ca calibration studies (a-b) and modern hydrography in the study areas (c-d). **(a)** Annual sea surface salinity in the Caribbean at 30 m depth (World Ocean Atlas 2013; Zweng et al., 2013). Surface sediments (white dots) were taken during the RV *SONNE* cruise SO164 (Nürnberg et al., 2003). **(b)** Annual sea surface salinity in the Gulf of Guinea at 30 m depth (World Ocean Atlas 2013; Zweng et al., 2013). Sediment samples (white dots) are from RV *METEOR* cruise M 6−5 (Lutze et al., 1988). Station numbers are indicated (Table 2). Maps were created with Ocean Data View (Schlitzer, 2015). **(c)** Annual range of salinities and **(d)** temperatures from 0 to 100 m water depth in the Caribbean and the Gulf of Guinea (World Ocean Atlas 2013; Zweng et al., 2013; Locarnini et al., 2013). Sample stations are color-coded (see legend). Stations of Caribbean surface sediments, which are not shown, vary within the black and yellow lines. Station SO164−18−1 equals station −07−3. Data are listed in Table 2.

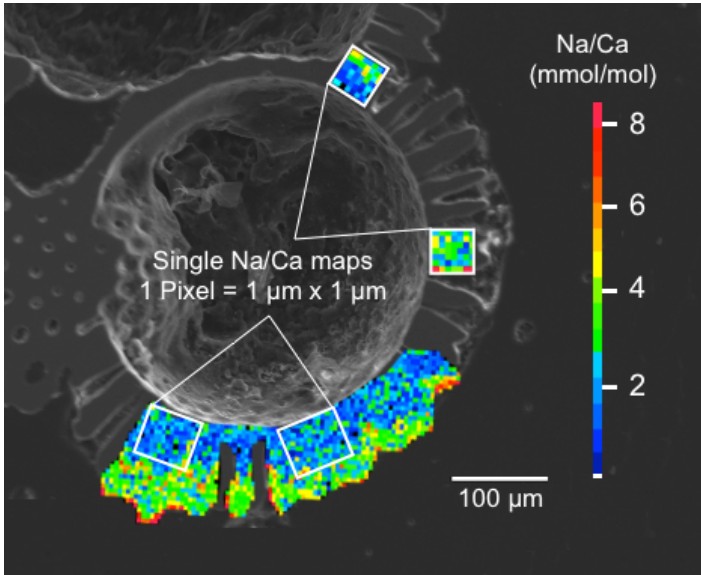

**Figure 2.** Secondary electron image with overlying maps of Na/Ca values in chamber wall cross sections of cultured *T. sacculifer*, measured by the electron microprobe. This is an example of single maps within one chamber. One pixel is 1 μm x 1 μm and the total map sizes vary from 5 μm to 20 μm, depending on chamber wall's thickness. Averaged Na/Ca maps are used to present results of both salinity and temperature experiments. Na and Ca intensities deviating more than 2σ from maximum intensities within a map, were discarded.

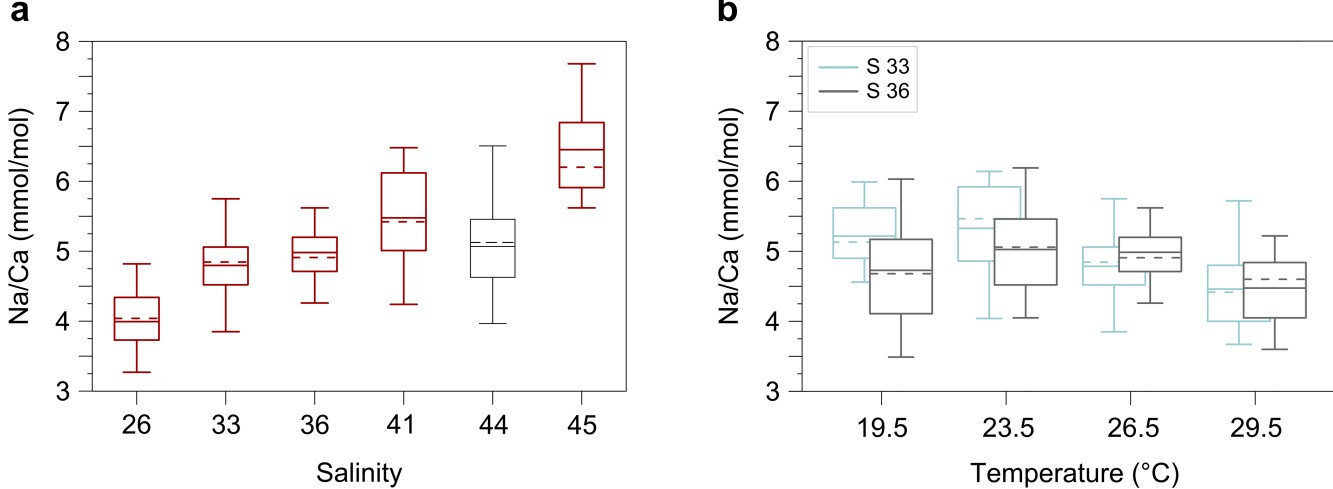

**Figure 3.** Box and whisker plots of the range and distribution of Na/Ca in newly grown test chambers of *T. sacculifer* during culture experiments. The boxes include all data within the first and third quartile. The horizontal solid line in the center of each box demonstrates the mean and the dashed line represents the median. Minimum and maximum values are marked by vertical lines (whiskers). Color-coding is similar to Figures 4 and 5. **(a)** Na/Ca within test chambers grown at varying salinities (26−45) and constant temperature (26.5 °C ±0.25 °C). Datasets for T = 26.5 °C of both temperature experiments (S 36, S 33) were added (Tables 4, 5). Number of electron microprobe maps for each experiment from lowest to highest salinity: n = 20, 29, 26, 34, 24, 12. The black colored box shows results of the 44-salinity experiment, in which cultured foraminifera underwent gametogenesis (Nürnberg et al., 1996) and where therefore excluded from the

regression in Figure 4. **(b)** Na/Ca within test chambers grown at varying temperatures (19.5 °C–29.5 °C) in two salinity groups at either constant salinity of 33 (S 33) or 36 (S 36). Number of electron microprobe maps for each experiment from lowest to highest temperatures: n (S 33) = 26, 31, 29, 29; n (S 36) = 26, 24, 26, 24.

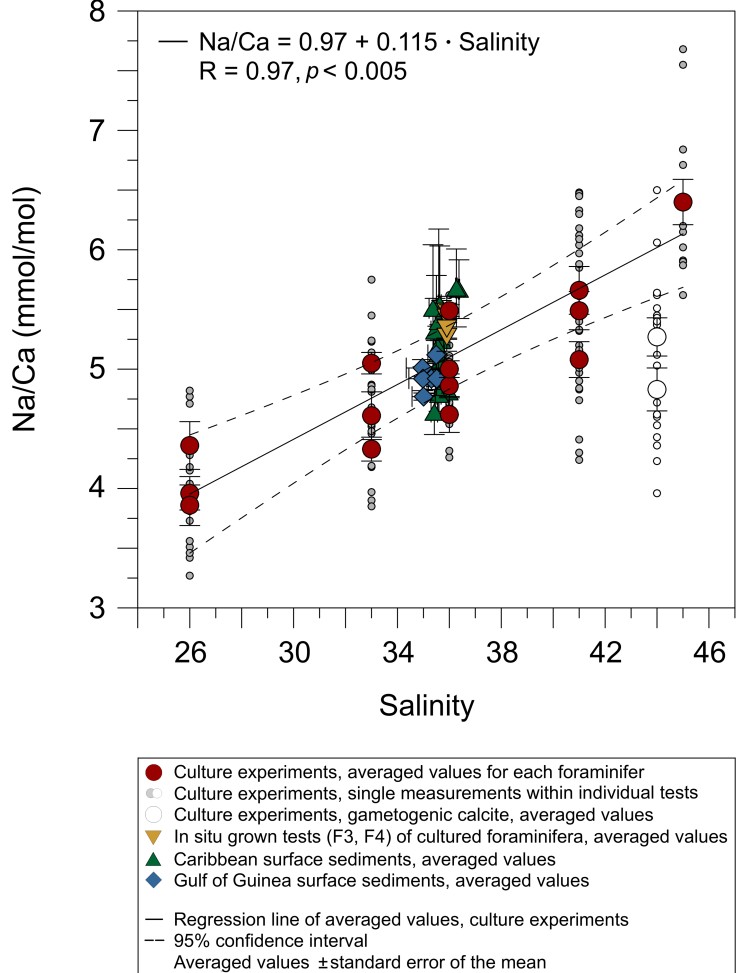

Na/Ca = 0.97 + 0.115 · Salinity
R = 0.97, p < 0.005

● Culture experiments, averaged values for each foraminifer
◐ Culture experiments, single measurements within individual tests
○ Culture experiments, gametogenic calcite, averaged values
▽ In situ grown tests (F3, F4) of cultured foraminifera, averaged values
▲ Caribbean surface sediments, averaged values
◆ Gulf of Guinea surface sediments, averaged values

— Regression line of averaged values, culture experiments
-- 95% confidence interval
   Averaged values ±standard error of the mean

**Figure 4.** Na/Ca measurements of cultured *T. sacculifer* increase significantly with increasing salinity, while the temperature remained constant (26.5 °C ±0.25 °C). Each grey data point comprises averaged Na/Ca values of single map electron microprobe analysis along inner chamber wall cross sections of newly grown calcite (F, F-1) during the experiments. Red dots indicate averaged Na/Ca values of one foraminifer. Related data of individual measurements of the salinity experiments are given in Table 3. Datasets for T = 26.5 °C of both temperature experiments (S 36, S 33) were added (Tables 4, 5). Within each experiment Na/Ca values are normally distributed (Shapiro-Wilk test, Appendix A). The solid line marks the regression line of red colored average values. Pearson's correlation coefficient R and its significance (*p*) is based on these averaged values. Dashed lines indicate the 95 % confidence interval. White data points represent results of the 44-salinity experiment, in which cultured foraminifera underwent gametogenesis (Nürnberg et al., 1996) and hence excluded from the regression. Yellow reversed triangles indicate Na/Ca$_{foram}$ values from in situ grown chambers (F-3, F-4) before culturing (annual salinity: ~35.9, temperature: ~27.4 °C, WOA13, Table 3). Blue squares mark Na/Ca values of *T. sacculifer* from Gulf of Guinea surface sediments (Table 1). Green triangles are data from Caribbean surface sediments (Table 1). Data marked by yellow, blue and green symbols are not included into the calculation of the regression. All vertical error bars are based on the standard error of the mean. Horizontal error

bars of samples from the Gulf of Guinea and the Caribbean (here as small as symbol sizes) demonstrate the salinity gradient between 0–60 m, reflecting the habitat range of *T. sacculifer*, expressed as standard error of the mean.

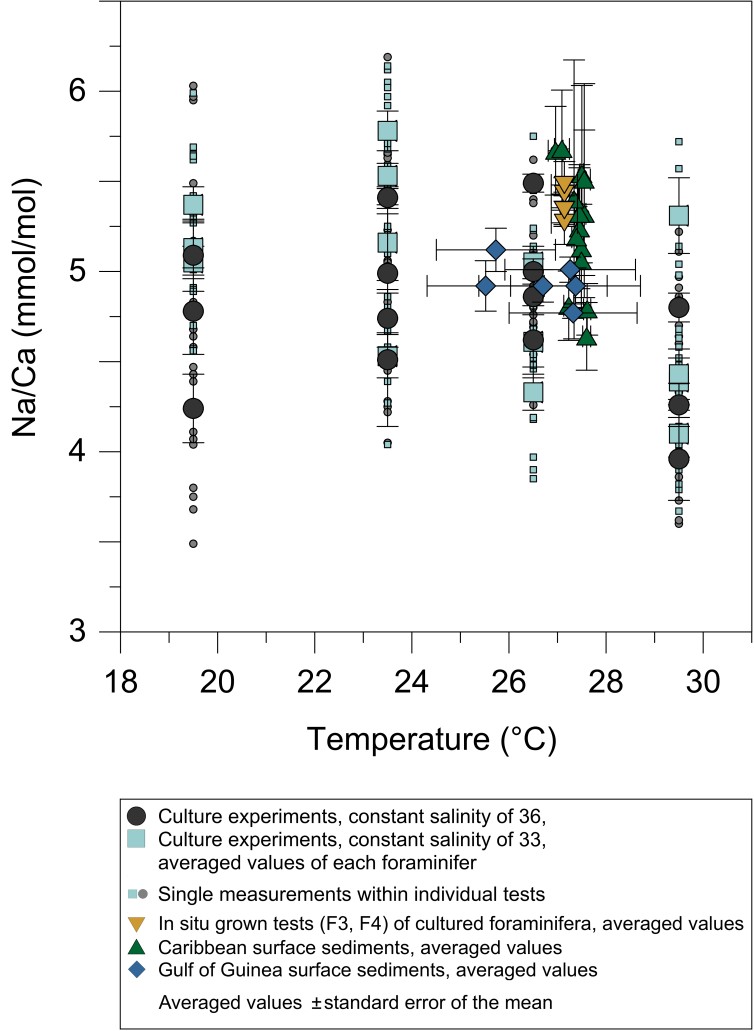

**Figure 5.** Na/Ca values of cultured *T. sacculifer* versus temperature show no significant correlation with increasing temperature (19.5 °C–29.5 °C ±0.25 °C) in two different settings at either a constant salinity of 33 (S 33, *p* <0.14 – grey dots) or 36 (S 36, *p* <0.69 – light blue squares). Each small light grey and light blue symbol comprises averaged Na/Ca values of single map electron microprobe analysis along inner chamber wall cross sections of newly grown calcite (F, F-1) during the experiments. Bigger symbols of the same latter colors indicate averaged Na/Ca values of one foraminifer. Related data are given in Tables 4 and 5. All vertical error bars are based on the standard error of the mean. Yellow reverse triangles indicate Na/Ca values from in situ grown chambers before culturing (annual salinity: ~35.9, temperature: ~27.4 °C, WOA13, Table 3). Blue diamonds reflect Na/Ca values of *T. sacculifer* from Gulf of Guinea surface sediments and green triangles are data from Caribbean surface sediments (Table 1). Horizontal error bars of surface sediments, which are as small as symbols for Caribbean samples, demonstrate the temperature gradient between 0-60 m. The gradient reflects the habitat range of *T. sacculifer*, expressed as the standard error of the mean.

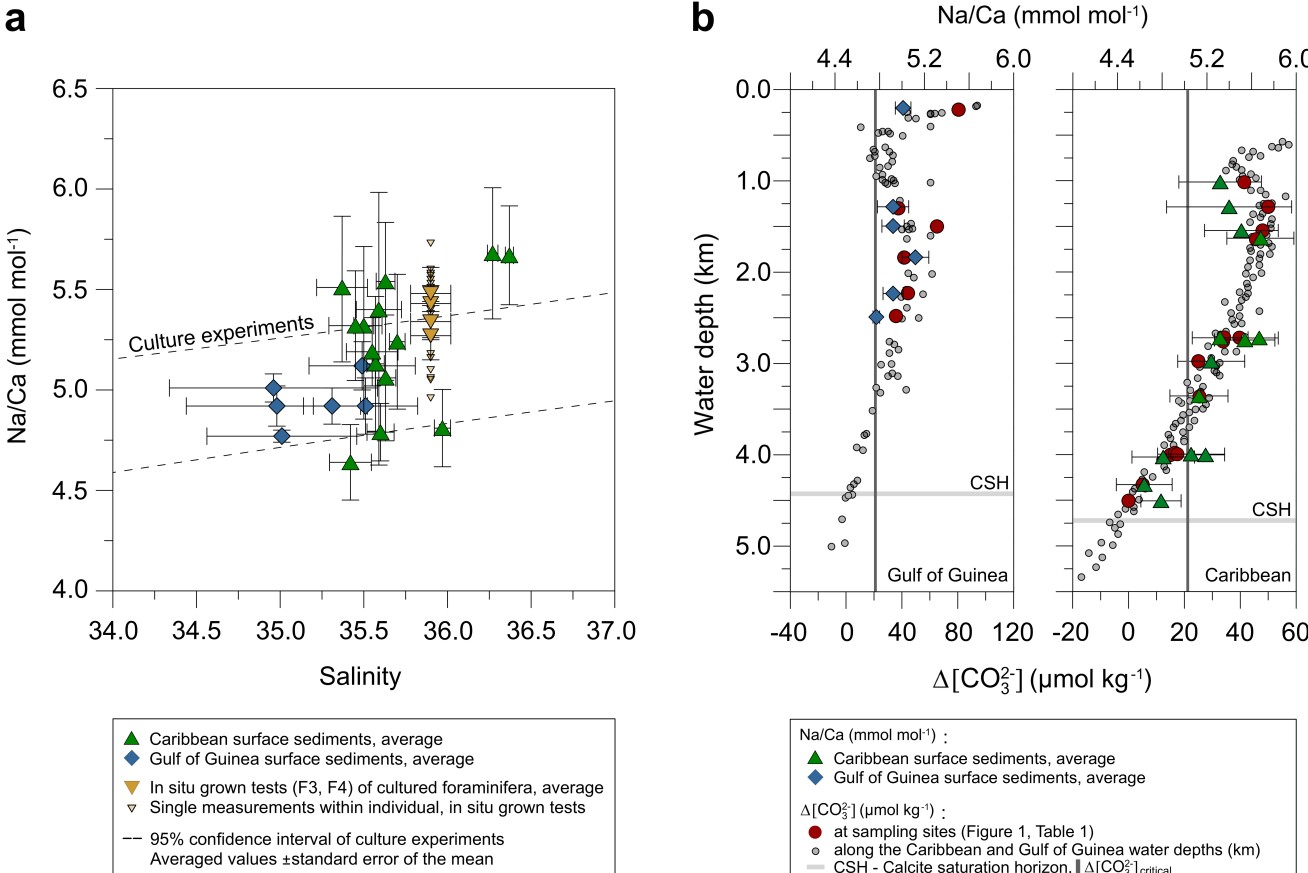

**Figure 6.** Averaged Na/Ca values of *T. sacculifer* from surface sediment samples collected in the Caribbean (stations SO164−01−3 to −50−3, green triangles) and the Gulf of Guinea (stations M6−5 GIK 16808−16869, blue diamonds) (Figure 1, Table 2). **(a)** Dashed lines
indicate the 95 % confidence interval derived from culture experiments (Figure 3) for comparison. Vertical error bars denote the standard error of the mean derived from replicate measurements (Table 2). Horizontal error bars visualize the salinity range from 0−60 m at the respective core locations (Figure 1), representing the possible habitat migration of *T. sacculifer* (Hemleben et al., 1987; Schmuker and Schiebel, 2002). **(b)** $\Delta[CO_3^{2-}]$ versus water depth at respective sampling stations (red dots), presented in Table 1, and the entire depth profile (light grey dots) in the Gulf of Guinea and the Caribbean. $\Delta[CO_3^{2-}]$ is defined as the difference between the in situ carbonate-ion
concentration ($[CO_3^{2-}]$) and $[CO_3^{2-}]$ at calcite saturation. Respective calcite saturation states for the Caribbean were taken from Regenberg et al. (2006) due to overlapping sample stations and for the Gulf of Guinea from Weldeab (2012) and Weldeab et al. (2016). Additional data were calculated with the program CO2SYS by Pierrot et al. (2006) (see Sect. 2.3 for detailed description). The horizontal calcite saturation horizon line marks the top of the present day lysocline, where $\Delta[CO_3^{2-}]$ is 0 µmol kg$^{-1}$ (Regenberg et al., 2006). The vertical line demonstrates the critical $\Delta[CO_3^{2-}]$ threshold of 21.3 µmol kg$^{-1}$, the onset of Mg/Ca decline in all planktonic foraminifera due to calcite
dissolution (Regenberg et al., 2006, 2014, Sect. 2.3).

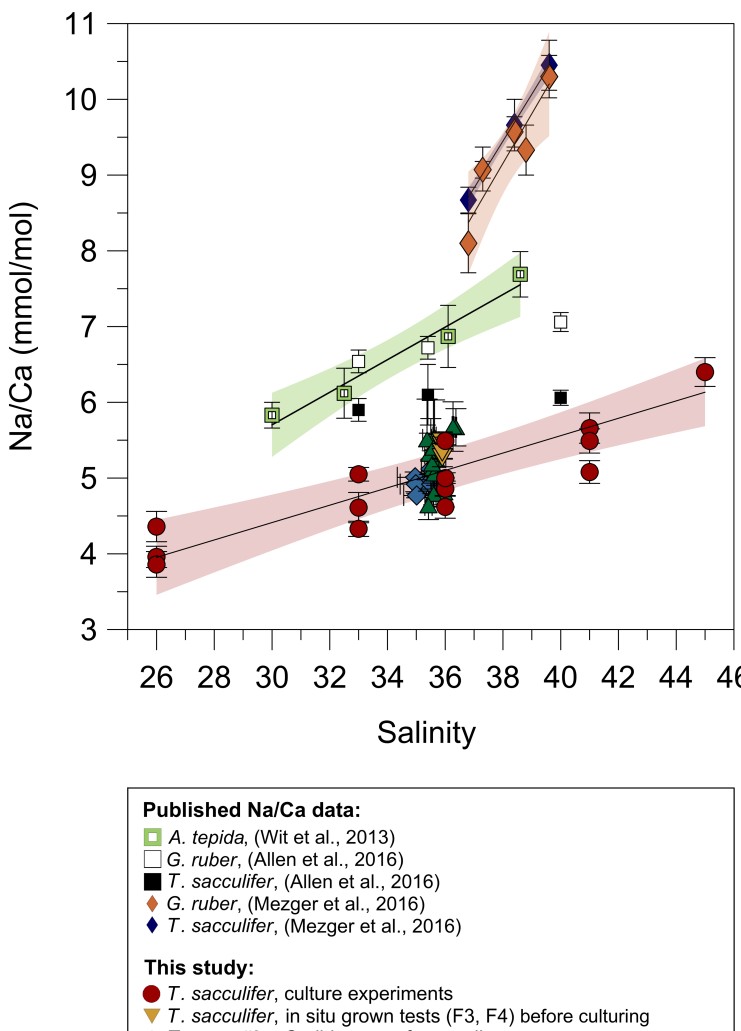

**Published Na/Ca data:**
☐ *A. tepida*, (Wit et al., 2013)
☐ *G. ruber*, (Allen et al., 2016)
■ *T. sacculifer*, (Allen et al., 2016)
◆ *G. ruber*, (Mezger et al., 2016)
◆ *T. sacculifer*, (Mezger et al., 2016)

**This study:**
● *T. sacculifer*, culture experiments
▼ *T. sacculifer*, in situ grown tests (F3, F4) before culturing
▲ *T. sacculifer*, Caribbean surface sediments
◆ *T. sacculifer*, Gulf of Guinea surface sediments

— Regression line; shaded areas: 95% confidence interval
  Averaged values ±standard error of the mean

**Figure 7.** Comparison of foraminiferal Na/Ca values in response to salinity changes between this study's *T. sacculi*fer from culture experiments and surface sediments with published Na/Ca data of benthic (Wit et al., 2013) and planktonic foraminifera (Allen et al., 2016; Mezger et al., 2016). All Na/Ca values are average values from 25–40 specimens per analysis, except for our culture experiments. Here, each red and yellow symbol represents average Na/Ca values of one foraminifer. Solid lines mark the regression of averaged values and shaded areas define the 95 % confidence interval for each study. Regression lines are based on the following equations: Wit et al. (2013): $\mathrm{Na/Ca}_{A.\ tepida} = 0.22 \cdot \mathrm{S} - 0.75$ ($R^2 = 0.96$, $p$ <0.01); Allen et al. (2016): $\mathrm{Na/Ca}_{G.\ ruber\ (pink)} = 0.074\ (\pm0.006) \cdot \mathrm{S} + 4.1\ (\pm0.2)$, ($R = 0.99$, $p$ <0.01), no significant relationship of Na/Ca with salinity in *T. sacculifer*; Mezger et al. (2016): $\mathrm{Na/Ca}_{G.\ ruber\ (w)} = 0.57 \cdot \mathrm{S} - 12.38$ ($R^2 = 0.91$, $p$ <0.001); $\mathrm{Na/Ca}_{T.\ sacculifer} = 0.60 \cdot \mathrm{S} - 13.49$ ($R^2 = 0.99$, $p$ <0.001); Our study: $\mathrm{Na/Ca}_{T.\ sacculifer} = 0.97 + 0.115 \cdot \mathrm{Salinity}$ ($R = 0.97$, $p$ <0.005). Vertical error bars are based on the standard error of the mean and horizontal error bars of surface sediments (as big as symbols for Caribbean samples) demonstrate the salinity gradient between 0–60 m water depth, the possible habitat range of *T. sacculifer*. Different regression lines for the same species, e.g. *T. sacculifer*, demonstrate various potential factors, and not only salinity, controlling Na incorporation in foraminiferal calcite (discussed in Sect. 4.3). However, our results of culture experiments correspond to fossil samples of surface sediments from two regions suggesting this regression is the most reliable for paleo-reconstructions.

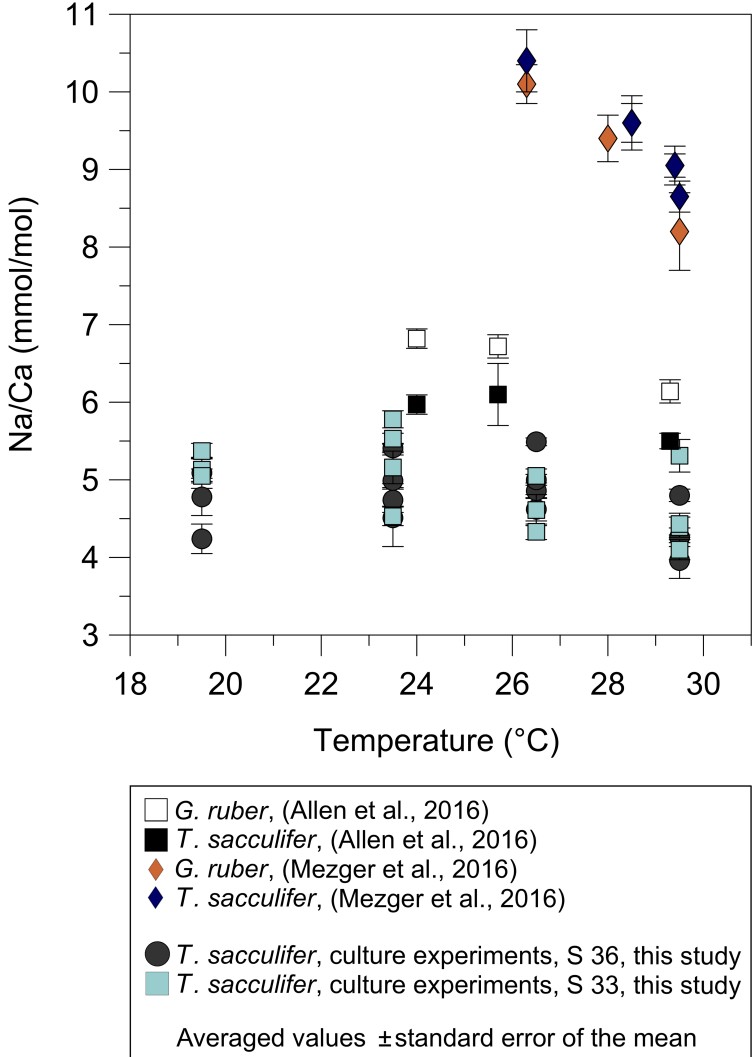

**Figure 8.** Comparison of foraminiferal Na/Ca values in response to temperature changes between this study´s *T. sacculifer* from culture experiments with published Na/Ca data of planktonic foraminifera from Allen et al. (2016) and Mezger et al. (2016). Averaged Na/Ca values of our study are shown in two different settings at either a constant salinity of 33 (grey dots) or 36 (light blue squares). During culture experiments from Allen et al. (2016) the salinity slightly decreased from 35.6 to 35.2 with increasing temperatures from 24 °C to 29.3 °C. Although in the field calibration study of Mezger et al. (2016) Na/Ca values apparently decrease with increasing temperature, corresponding salinities also decrease from 39.6 to 36.8. Vertical error bars are based on the standard error of the mean. Except for specimens from the Red Sea, influenced by co-varying factors, (*G. ruber*: $R^2 = 0.84$, *p* <0.001; *T. sacculifer*: $R^2 = 0.95$, *p* <0.001; Mezger et al., 2016), all Na/Ca data for *T. sacculifer* and *G. ruber* show no significant response to temperature changes (our study; Allen et al., 2016).

## Appendices

### Appendix A

**Table A1.** Na/Ca values generated for each experiment were tested for normality with the Shapiro-Wilk W-test and its corresponding *p*-value. If *p* is ≥ 0.05 (95 % confidence interval), the data are normally distributed. The number of measurements is indicated with n. Salinity and temperature (T) values derive from culture experiments listed in Tables 3, 4 and 5.

| Salinity | T (°C) | n | W | *p* |
|---|---|---|---|---|
| 26 | 26.5 | 20 | 0.962 | 0.587 |
| 41 | 26.5 | 34 | 0.953 | 0.590 |
| 44 | 26.5 | 24 | 0.972 | 0.709 |
| 45 | 26.5 | 12 | 0.878 | 0.083 |
| 33 | 19.5 | 26 | 0.960 | 0.399 |
| 33 | 23.5 | 31 | 0.952 | 0.183 |
| 33 | 26.5 | 29 | 0.971 | 0.590 |
| 33 | 29.5 | 29 | 0.941 | 0.108 |
| 36 | 19.5 | 27 | 0.964 | 0.462 |
| 36 | 23.5 | 24 | 0.960 | 0.440 |
| 36 | 26.5 | 26 | 0.975 | 0.767 |
| 36 | 29.5 | 24 | 0.951 | 0.286 |

### Appendix B

**Table B1.** Inter- and intra-test Na/Ca variability of cultured *T. sacculifer* with varying experimental settings. The Na/Ca intra- and inter-test variability is calculated from single averaged electron microprobe maps (Tables 3, 4, 5) within one foraminifer (#) and between foraminifera (Na/Ca$_{average}$ ±standard error of the mean) from the same experiment. The variability is expressed as the relative standard deviation (RSD) in %. Annual salinity and temperature data for in situ grown foraminiferal calcite (F-3, F-4) in the open ocean were taken from WOA13 (Zweng et al., 2013; Locarnini et al., 2013).

| Sample # | **Intra-test variabilty RSD (%)** | **Inter-test variabilty RSD (%)** | Temp (°C) | Salinity | Na/Ca$_{average}$ (mmol mol$^{-1}$) |
|---|---|---|---|---|---|
| **Salinity experiments** | | | | | |
| 7912 | 10.39 | 6.52 | 26.5 | 26 | 4.36 ±0.20 |
| 7913 | 10.23 | | 26.5 | 26 | 3.96 ±0.14 |
| 7914 | 11.74 | | 26.5 | 26 | 3.86 ±0.17 |
| 8135 | 13.27 | 5.51 | 26.5 | 41 | 5.66 ±0.20 |
| 8137 | 11.06 | | 26.5 | 41 | 5.49 ±0.16 |
| 8138 | 7.44 | | 26.5 | 41 | 5.08 ±0.15 |
| 7703 | 12.22 | 6.16 | 26.5 | 44 | 4.83 ±0.18 |
| 7704 | 10.73 | | 26.5 | 44 | 5.27 ±0.16 |

| | | | | | |
|---|---|---|---|---|---|
| 8301 | 10.40 | – | 26.5 | 45 | 6.40 ±0.19 |

**Temperature experiments (constant S 36)**

| | | | | | |
|---|---|---|---|---|---|
| 1.2 | 14.18 | 9.15 | 19.5 | 36 | 4.24 ± 0.19 |
| 1.3 | 12.38 | | 19.5 | 36 | 4.78 ± 0.24 |
| 1.4 | 12.57 | | 19.5 | 36 | 5.09 ± 0.20 |
| | | | | | |
| 2.1 | 7.50 | 7.84 | 23.5 | 36 | 4.74 ± 0.16 |
| 2.2 | 3.53 | | 23.5 | 36 | 5.41 ± 0.06 |
| 2.3 | 16.23 | | 23.5 | 36 | 4.99 ± 0.33 |
| 2.4 | 8.17 | | 23.5 | 36 | 4.51 ± 0.37 |
| | | | | | |
| 3.1 | 6.36 | 7.35 | 26.5 | 36 | 4.62 ± 0.15 |
| 3.2 | 5.32 | | 26.5 | 36 | 4.86 ± 0.10 |
| 3.4 | 3.98 | | 26.5 | 36 | 5.00 ± 0.07 |
| 3.5 | 1.83 | | 26.5 | 36 | 5.49 ± 0.05 |
| | | | | | |
| 4.1 | 5.38 | 9.81 | 29.5 | 36 | 4.80 ± 0.08 |
| 4.3 | 12.71 | | 29.5 | 36 | 3.96 ± 0.23 |
| 4.4 | 8.60 | | 29.5 | 36 | 4.26 ± 0.12 |

**Temperature experiments (constant S 33)**

| | | | | | |
|---|---|---|---|---|---|
| 5.1 | 4.76 | 3.21 | 19.5 | 33 | 5.37 ± 0.10 |
| 5.2 | 9.96 | | 19.5 | 33 | 5.13 ± 0.15 |
| 5.5 | 5.27 | | 19.5 | 33 | 5.05 ± 0.09 |
| | | | | | |
| 6.1 | 3.58 | 10.35 | 23.5 | 33 | 5.53 ± 0.07 |
| 6.2 | 6.72 | | 23.5 | 33 | 4.53 ± 0.12 |
| 6.3 | 9.29 | | 23.5 | 33 | 5.16 ± 0.21 |
| 6.5 | 6.28 | | 23.5 | 33 | 5.78 ± 0.11 |
| | | | | | |
| 7.1 | 6.35 | 7.43 | 26.5 | 33 | 4.33 ± 0.10 |
| 7.3 | 4.91 | | 26.5 | 33 | 5.05 ± 0.09 |
| 7.4 | 7.23 | | 26.5 | 33 | 5.05 ± 0.14 |
| 7.5 | 10.54 | | 26.5 | 33 | 4.61 ± 0.20 |
| | | | | | |
| 8.1 | 8.35 | 11.47 | 29.5 | 33 | 4.39 ± 0.13 |
| 8.3 | 9.24 | | 29.5 | 33 | 4.10 ± 0.13 |
| 8.4 | 8.91 | | 29.5 | 33 | 4.43 ± 0.14 |
| 8.5 | 7.80 | | 29.5 | 33 | 5.31 ± 0.21 |

In situ grown chambers (F-3, F-4)

| | | | | | |
|---|---|---|---|---|---|
| 4.1 | 5.63 | 1.69 | 27.4 | 35.9 | 5.27 ±0.12 |
| 2.3 | 3.25 | | 27.4 | 35.9 | 5.43 ±0.18 |
| 7704 | 1.66 | | 27.4 | 35.9 | 5.48 ±0.04 |
| 7703 | 3.85 | | 27.4 | 35.9 | 5.34 ±0.08 |

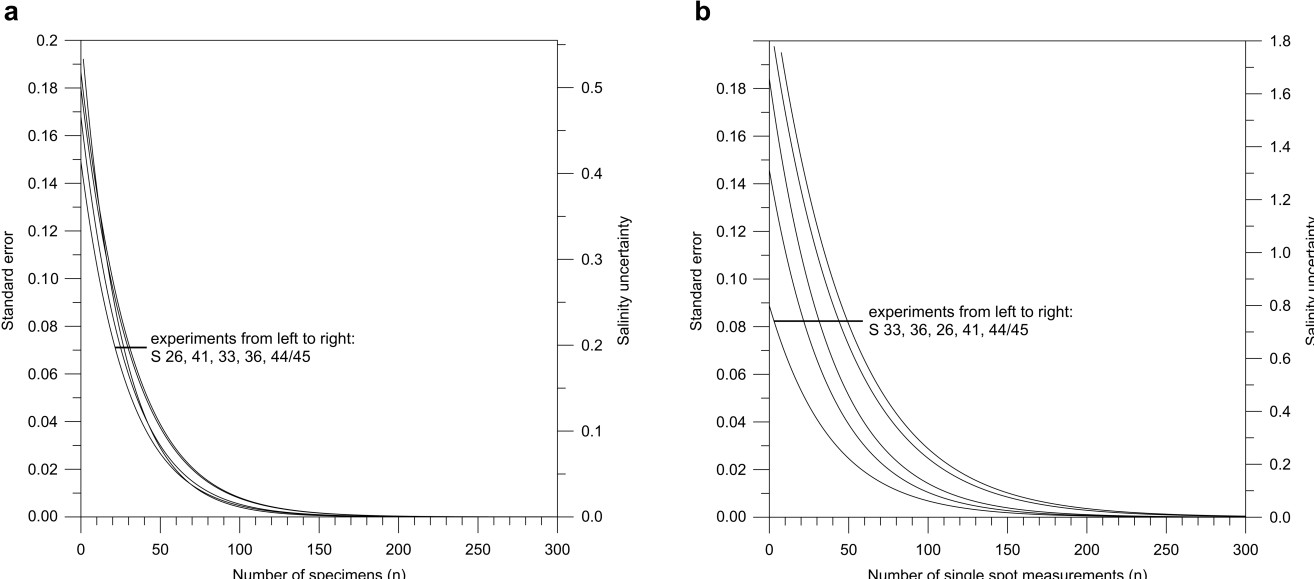

**Figure C1.** Standard errors (standard deviation $\sigma/\sqrt{n}$) of Na/Ca values from cultured *T. sacculifer* for each salinity experiment (S) related to salinity uncertainties. Salinity calculations are based on our calibration (Salinity = ((Na/Ca$_{T.\ sacculifer}$ − 0.97)/0.115)) **(a)** Number of total specimens needed to resolve a certain salinity range. **(b)** Number of single spot measurements by electron microprobe analysis within chamber wall cross sections are required to resolve a certain salinity.

1070