# Peer review of "Salinity control on Na incorporation into calcite tests of the planktonic foraminifera *Trilobatus sacculifer* – Evidence from culture experiments and surface sediments"

_Biogeosciences, 2018_

## Referee Comment (RC1) · Anonymous Referee #1 · 15 May 2018

Bertlich et al., make use of earlier studies that introduced Na/Ca ratios from foraminifera as a proxy for surface seawater salinity, and they assess the reliability of Na/Ca derived from T. sacculifer as a salinity proxy. They conclude that Na/Ca can be applied as a reliable proxy for reconstructing sea surface salinity and furthermore that species-specific calibrations are necessary. General comments: This study is one step closer towards establishing Na/Ca as a proxy for SSS. It is a well-designed study making use of culture experiments, wherein they vary salinity keeping temperature constant or vice versa and measure the Na/Ca to understand the salinity and temperature

effects on Na/Ca. They further corroborate the culture studies with surface sediments from two different locations, Caribbean and Gulf of Guinea. This study has the potential to highlight the efficiency of Na/Ca as a salinity proxy which may soon be followed by other studies that may target different foraminiferal species and oceanic locations so as to develop this proxy further into a robust one. This proxy can then be utilized in unison with $\delta$18O and in some cases, Ba/Ca, to get a better handle on absolute salinity values, as each proxy has its own limitations. Here, the authors have touched a few important environmental variables that may affect and are needed in proxy validation. However I urge the authors to consider the following issues that I have raised and incorporate them in the manuscript. Specific comments: • The authors mention that in order to avoid dissolution they have chosen sampling sites where $\Delta[CO3^{2-}]$ is >30 $\mu$mol/kg. But, if samples from water depths deeper than 2.7 km (see Table 1) are plotted against Na/Ca a clear depth dependency is noticed, Na/Ca decreasing with increasing depth. The authors should make this explicit to the reader by providing the $\Delta[CO3^{2-}]$ for each sample/depth and a separate paragraph discussing the depth dependency/dissolution effect on Na/Ca. • The range in spatial salinity distribution at the surface sediment locations is ∼1 to 1.5 salinity units. However the vertical distribution in salinity which may be encountered by T. sacculifer at its habitat, especially at the Gulf of Guinea is large, reaching ∼ 4 salinity units. Planktic foraminifera live for few weeks and so may encounter ambient conditions which are seasonal during their life cycle (eg: Honisch et al., 2013 GCA). So when applying the Na/Ca proxy to down core samples to environments such as the Gulf of Guinea, how one does take into account such large vertical variations. This signal will also be mixed with the seasonal signal. Ideally surface samples should have covered a larger range in surface salinity.

Technical corrections: Line 74: Do you mean G. sacculifer? Line 89: 'related' should rather be 'correlated'? Line 92: SCUBA is written in upper case, is it an acronym? Line 95: Salinity experiments were done at salinities of 23, 26, 41, 44 and 45, which do not encompass the entire salinity range of 23 to 45. Why are the mid ranges not included? Line 100: '26.5°C to 29.5°C' OR '26.5°C and 29.5°C'? Line

114 and 133: any reason for using the two different size fractions? 315 to 400$\mu$ and 300 to 400$\mu$? Line 138: 'The annual SSS varies from east to.........'. The variation of SSS from 32 to 35.9 is spatial or with depth? Line 140 to 141: $\Delta CO3=$ should be calculated for respective depths and included in tabular form. Line 221: Na/Ca increases by 0.12 mmol/mol per salinity unit. However the error on Na/Ca measurements at many instances is beyond this value (see table 1). Line 273: 'Nevertheless, it is still a ....noticeable'. Should it read as.......' .....which is noticeable?'
* * *
Figure1b: As seen from the figure, the GIK samples seem to fall within 35 and 36 salinity units. However the figure 1c shows a larger range in salinity, from $\sim$ 31.7 to 34.7 units. Figure 3a: Why is the 5th box, at salinity value of 44, grey in color? The text describes the salinity values for culture experiments of 23, 26, 41, 44 and 45 salinity units. The figure however shows 26, 33, 36, 41, 44 and 45 salinity units.
* * *
References: Line 33, 35 and 36: Rohling and Bigg, 1989 is listed as Rohling and Bigg, 1998 in the reference list. Line 78: The reference Lin et al., 2004 is Lin et al., 2014 in reference list. Line 193: Barker et al., 2003 is not listed in the references. Lines 297, 300 and 487: Dueñas-Bohórquez et al 2011 mentioned in text is Dueñas-Bohórquez et al 2011b in the reference list. Line 469: Busenberg and Plummer, 1989 is not included in the reference list. Lines 538 and 540: Bijma et al., 1990 are two different papers and should be labelled as 1990a and 1990b, in text and reference list. Line 682: The reference Spero, 1988 is not found in the text.

---

## Referee Comment (RC2) · Anonymous Referee #2 · 18 May 2018

**Review for Bertlich et al. „Salinity control on Na incorporation into calcite tests of the planktonic foraminifera *Trilobatus sacculifer* – Evidence from culture experiments and surface sediments"**

Bertlich and co-authors present an interesting and important study for proxy development by using Na/Ca ratios of the planktonic foraminifera *Trilobatus sacculifer* to reconstruct salinity using electron microprobe analyses. This study is well designed as the authors present a foraminiferal Na/Ca-salinity calibration curve which is based on culture experiments and compare this curve with actual surface sediment samples from two different tropical Atlantic regions. The culture experiments comprise a wide range of salinity and also test the influence of temperature on Na incorporation into foraminiferal tests. Further controlling factors of Na incorporation are also explored briefly. The results of the surface sediments agree well with the culture experiments which validate the calibration curve presented in this study.

The manuscript is mostly well written, logically organised and clear. The figures are overall nice and clear. I think that this work is an interesting and important contribution and therefore suitable to be published in Biogeosciences. Nevertheless, I would like to see the points below addressed by the authors.

Main points

1) You mention in the manuscript that your intra-test variability of the temperature experiments is less than half than that of the salinity experiments. I am extremely sorry, but I strongly disagree. According to Table B1, the variability between both experiments is similar. The average intra-test variability of the salinity experiments is 10.82% RSD, the half of this is 5.41% RSD, but only 9 out of 27 individual foraminifera results of the temperature experiments have actually lower RSDs than 5.41%. So please change those sentences (L260-262, L382-384) or make clear what you mean.

2) Although the data is good and relevant, the authors did not convince me of their results in the discussion, especially Sections 4.3 and 4.4. The authors introduce an interesting train of thought but then did not think it through, instead they start a new line of thought or connect one idea with another seemingly unrelated. I really like the first section of the discussion because the influence of biological processes/response is brought into focus which is important for this study, considering that Na$^+$ ions are essential for ion channels/pumps and for the general functioning of cells and could therefore influence/bias the incorporation into calcite tests. However, Section 4.3 did not address these and other biological aspects further and the reader is left with a lot unanswered questions. Furthermore, the authors jump back and forth in this section which makes it hard to follow each line of thought. Your data is great! You just need to sell it better.

3) Your calibration indicates an increase in Na/Ca ratios by 2.25% per salinity unit. However, you intra- and inter-test variability is much higher than that ranging from 1.83-16.21% and 3.19-11.51% (according to Table B1) when using electron microprobe (or other micro-analytical techniques). Could you please give an estimate on how many individual measurements are needed to distinguish $\pm$ 0.5-1 salinity units analytically and statistically?

4) You say that dissolution and early diagenetic processes do not influence the foraminiferal Na/Ca ratio of your surface sediments, however, you state in the Material and Method section that you only chose sites, where $\Delta[CO_3^{2-}]$ is >30 µmol/kg, which in turn is above the critical calcite saturation value given by Regenberg et al. (2006, 2009). Therefore, please explain how you can address the effect of dissolution on Na/Ca ratios.

In this regard, it would be of benefit if you could insert the $\Delta[CO_3^{2-}]$ values for your stations in Table 1. Have you tried to plot your Na/Ca values from Table 1 against $\Delta[CO_3^{2-}]$ to see if there is a relationship?

Further, did you check (visually, by SEM) the tests of the surface sediments for any signs of dissolution/early diagenesis?

I guess, this is difficult to address with EMP, but do you see differences in test wall thickness, test size or thickness of primary calcite between specimens from shallower and deeper stations in the Caribbean Sea (similar to the results from Johnstone et al. 2011, Sadekov et al. 2010)? Assuming you did measure some tests by using EMP…

Minor points

L24-25 better to say "…indicating salinity to be/is one of the dominant factors…" as I think the biological component on Na incorporation into calcite is still unclear.

L26-27 Considering that the biological influence is still unclear, maybe replace "reliable proxy" with "potential proxy".

L74-76 It would be interesting to see if *T. sacculifer* and *T. triloba* actually have a similar geochemical signature. Did you test this? If so, could you please provide these data in the supplements? This would then be the first time such a comparison has been done.

L91-94 Were the (SCUBA dive-) collected specimens of *T. sacculifer* juveniles or young adults? Please indicate.

L104-105 Your "slight" increase in salinity of 0.5-0.8 units during the 29.5°C experiment is actually the salinity range you say you can resolve with your calibration! Please delete "slight". What does this mean for your error bars in Figure 5 if you include the related uncertainty?

L113 Please insert that the water depth is also given in Figure 1 and Table 1 as this is important concerning the preservation state of the tests. As mentioned in the main points, could you please include the $\Delta[CO_3^{2-}]$ values for your stations in Table 1?

L196-198 Can you please give the concentration of your hydrazine solution?

L211-212 This sentence reads odd. Do you mean your measured value is $19.79 \pm 0.51$ mmol/mol? If not, which reference did you use and what is your analysed value for Na of the JCp-1?

L213-214 When you say "every sample solution was measured 5 times", do mean your foraminifera samples only or also the JCp-1? Please be more precise.

L233-234 Reference Figure 3a to be precise. You also have a comparably large range in your temperature experiments 19°C < x < 24°C for salinity 36 according to Figure 3b and Table B1. Please mention this here or in the section about the temperature experiments.

L260-262 As mentioned in the main points, please delete this sentence as this is not true according to Table B1: Only 9 experiments out of 27 have actually lower intra-test variability.

L269-270 The value of $4.71 \pm 0.21$ mmol/mol is not listed in Table 1, the lowest value in Table 1 is $4.64 \pm 0.25$ mmol/mol.

L281-283 Please explain briefly what the salinity uncertainty of the Gulf of Guinea surface sediments means, illustrated by horizontal error bars, as this is only mentioned in the caption of Figure 4. Why is there no such uncertainty for the sediment samples from the Caribbean Sea? Is the error bar as big as the symbol? If yes, please say so.

L285-287 Could you please give an explanation why the stations GIK 16860-1/16864-1 have higher Na/Ca values than station GIK 16865-1, although they are all close to the river mouths, similar to your explanation in L270-273?

L293-294 According to Table B1 your Na/Ca intra- and inter-test variability varies between about 2-16% (lowest value: 1.83%).

L300-303 You can also reference Hathorne et al. 2009 here.

L303-305 Please reference Spero et al. 2015 here.

L306-309 I do not think that Figure 4 shows that the lowest intra-test variation is obtained at conditions close to those at the natural habitat. Reference only Figure 3 here.

L306-314 I really like that you discuss here the possible biological response to increased stress levels. You say that increased stress levels could lead to higher intra-test variability. Could that have influenced your Na/Ca values at low and high salinities (like seen in Figure 3 – high variability at T < 24°C and at S ≥ 41)? Could this, in turn, result in a lower or steeper slope of your calibration curve?

L328-330 Mention here, that you assume that the observation of Branson et al. (2016) could also be valid for *T. sacculifer* and other planktonic species.

L347-349 Please indicate here how many individual measurements are necessary to achieve resolving a salinity change of about 1 unit using EMPA, as addressed in main point 3).

L350-351 What do you exactly mean with "little small-scale variability"? The intra- or inter-test variation itself or compared to the higher variability of Mg/Ca?

L356-357 As said before and in main point 3): Please state the minimum number of individual analyses to actually resolve a salinity change of $\pm 0.5$-1 units?

L379-380 I would not be so hasty, the study of Mezger et al. was a field calibration which means there are most likely several factors that induce stress or otherwise influence the Na$^+$ incorporation compared to a culture study with rather stable conditions. The different results between the field and the culture studies implies to me that there are other controlling factors than salinity alone. The results from Mezger et al. could also be region-specific or relate to the variation in salinity within one transect as the authors mentioned. But this needs to be discussed in more detail as mentioned in main point 2).

L382-384 As said in main point 1), your intra-test variations are similar between the salinity and temperature experiments. Please delete this sentence.

L433-435 As mentioned in the main point 4), you state in L140-143 that you have only used sediments from sites with $\Delta[CO_3^{2-}] > 30$ µmol/kg, which is higher than the critical calcite saturation level. So, how do you know that dissolution did not affect your Na/Ca ratios? Did you check those samples for any signs of dissolution/diagenesis (visually or by SEM)?

L435-437 I think what you mean here, is that your surface sediment samples fit nicely to your calibration curve, but the sediment samples themselves form a cloud if you look at Figure 6. Therefore, these data do not show any trend or even a slope as you state here. Please rephrase this sentence. I also think that you should reference Figure 5 here as well, as this figure shows a good comparison of your culture data and the sediment samples including their similar variability.

L449-451 Can you please give examples of which kind of biogenic calcite is meant here? A specific species or a species group? And was this relationship found a specific (regional) setting?

L461-463 Can you please briefly explain what you mean by the biological control on Na incorporation is smaller than for Mg? Or delete the last part of this sentence as you compare and discuss $D_{Na}$ and $D_{Mg}$ in L468-473.

L464-465 Can you please explain this in more detail? Up to this point, you always said that the Na/Ca values at salinity 44 are low because of the precipitation of gametogenic calcite. So, what other biological aspects can influence the Na incorporation?

L464-467 You said that you only found gametogenic calcite at salinity 44 and that such high salinities could impact the growth of the test and therefore, the Na incorporation. So, what does that mean for your highest salinity experiment at S = 45? Do you see gametogenic calcite in those samples and if not, do you have an idea what makes S = 44 so special? Or why do the samples at S = 45 fit to your calibration and those at S = 44 not, if high salinity has an impact on Na/Ca?

L488-490 What about all the other possible controlling factors? In Section 4.3, you discuss that the cleaning procedure, the photosynthetic rate of symbionts, thermal stress and potentially dissolution can impact the Na/Ca values. Can you rule out some of the factors? If so, please mention this in Section 4.3.

L493-494 the same as in L433-435.

Table 1 Is the salinity and temperature record of the World Ocean Atlas (2013) that imprecise that your values have an error of $\pm 30$ m? Please include the $\Delta[CO_3^{2-}]$ value for each station, as mentioned in main point 4).

Figure 7 You show 3 different regression lines for *T. sacculifer*, which suggests that there could be other influences on Na/Ca than only salinity, in my opinion. You give several potential explanations in Section 4.3, but could you please explore the potential (unknown) impact of biological processes on Na/Ca as mentioned in main point 2)? What makes your calibration the most reliable one? Consider your big advantage of culture and surface sediment samples.

Table B1 I am confused. Why does this table give different values and uncertainties for the individual measurements than Tables 4 and 5. Not all values are the same as listed in Tables 4 and 5. Does this result in slightly different intra- and inter-test variabilities?

Style and language:

1) Please check your grammar and word order in your sentences. It is sometimes hard to follow what you mean. And please try to avoid nested sentences, they are confusing, e.g. L64-67. It is better to split those sentences into two to be more clear. For more examples, see also minor points below.

2) The discussion, especially Sections 4.3 and 4.4, needs to be better structured. As mentioned before, the arguments should be linked better. Additionally, the authors jump back and forth in this section which makes it hard to follow each line of thought.

L40-41 In the entire manuscript, you cite the references in chronological order but here alphabetically. Please change this to be consistent.

L58-60 better to say "The high spatial resolution of the microprobe technique…"

L64-67 As mentioned above, please avoid such nested sentences and make 2 sentences here as this sentence is confusing and somehow reads more like a list.

L82-84 better: "Prior to gametogenesis at the end of its life cycle, a process related to reproduction, *T. sacculifer…*"

L118-120 I would delete the part of the sentence about the Subtropical Underwater mass as I do not think you need this information, since *T. sacculifer* lived in the Caribbean Surface Water mass.

L120-122 I think you mean: "…entering the NE Caribbean through the Lesser Antilles."

L130-131 If you reference more than one figure, please use plural, e.g. Figures 1b-d. This is valid for all your references to your figures (e.g., L231, L239, L253, L267, L269, L270, L275, L280, L287,…).

L138-139 Please reference Figures 1b and c here which show salinity at 30 m and with increasing depth as you say in this sentence.

L152-153 If I am not mistaken the abbreviation is "EMPA".

L154-155 "…were covered by a 20 nm carbon coating…" better to say "…were impregnated with carbon coating…"?

L161-162 Please use either "shell" or "test" to be consistent. You mostly use "test" in the manuscript.

L170-171 better to say "Absolute Na/Ca ratios were calculated using…" instead of "quantified"

L170-171 Please delete $I_{back}$ and $I_{total}$ as the abbreviations are not needed, you do not use them again in the entire manuscript.

L171-172 I know what you mean but it sounds wrong to compare intensities with concentrations. I think it is better to state: "The obtained net intensities were converted to concentrations and corrected to known concentrations (in wt %) of referenced materials."

L198-199 "The samples were thoroughly rinsed to remove the chemicals…"

L201-203 Delete "…and dissolved…", it does not really fit with the rest of the sentence.

L205-206 „The machine is..." device?

L206-208 "…these wavelengths were 70.60 nm for Ca, 279.55 nm for Mg, and 589.59 nm for Na."

L208-209 Delete one "with".

L209-210 Delete "Hence" at the beginning of sentence.

L211-212 You give only one values, so please use the singular. Also, provide the value together with its uncertainty using "±". Please use the past tense in this sentence. Please be also consistent in how you report units. Here you use "mmol/mol" whereas you mainly use "mmol mol$^{-1}$" in the manuscript. This is also valid for the tables in which the units are reported in one or the other version. You always use "mmol/mol" for your plots, but could you also change the unit to "mmol mol$^{-1}$" to be consistent throughout the entire manuscript?

L227-228 Space between 13.2 and %. Please be consistent, whether you use a space between numbers and the % symbol or not.

L230-231 I do not think that Figure 4 represents the low variability of the experiment at 26.5°C and salinity 36. I guess, it would be better to cite Section 2.3 here where you describe the natural habitats.

L237-239 Reference Figures 3a and 4.

L251-252 Delete "…within the same temperature interval.". The interval mentioned in the sentence before is your entire range of T experiments, so this bit of text is not needed, on the contrary it is confusing.

L257-258 This information is already said in L228-230, however, fits better here in the T experiment section. Hence, delete sentence in L228-230.

L259-260 Reference Figure 3b here.

L272-273 You are always very precise with reporting differences between Na/Ca values, so please say here "± 0.16 mmol/mol" or say "about 0.2 mmol/mol". Further move "noticeable" to after "increase of about 0.2 mmol/mol".

L278-280 better to say "…as the inter-test variability indicates of specimens from…"

L290-291 Please decide whether you use "shell" or "test" in your manuscript. I think, it is better to stick to one expression to be consistent. You mainly use the term "test".

L306-308 I would rephrase this sentence a little bit: "Transferred to our study, higher Na/Ca intra-test variations may occur between maps within single wall cross sections of each salinity experiment compared to the temperature experiments (Table B1) due to higher stress levels of (individual) foraminifera in culture."

L315-316 Delete "more or less". I think it is not important here how much GAM calcite is produced, only that it is precipitated.

L318-320 Delete "probably" and say instead: "…their biomineralisation mechanisms could be fundamentally different."

L320-321 Insert "at salinity 44" at the end of the sentence to make it clear. Also "Figures 3, 4".

L327-328 Rephrase this nested, confusing sentence, e.g.: "…dark layers in benthic foraminiferal calcite, most likely comprising the primary organic sheet, which in turn is located between the primary and secondary calcite."

L339-341 In your plots and other parts of the manuscript, the word "versus" is spelled out and not written in italics. Please be consistent throughout the manuscript. Further, maybe insert "their" between "for" and "*T. sacculifer*" to make clear that only Allen et al.'s study is meant here.

L341-342 Reference Figure 7 at the end of the sentence.

L342-344 Also reference Figure 7, best right after "Allen et al. (2016)" and reference Section 4.3 at the end of the sentence as you discuss in detail the difference between those previous studies in that section.

L351-353 As mentioned before, please use either "test" or "shell" in the manuscript.

L360-362 Please spell "YD" out as it is only used this once in the entire manuscript.

L362-363 "…a salinity decrease of ~2-4  units." I am not sure if you need the "salinity" in front of "units" as you already said it is about salinity.

L364-366 Please make 2 sentences out of this nested sentence as it is confusing. You also used "$\delta^{18}O_{ivf\text{-}sw}$" before instead of "$\delta^{18}O_{ivc\text{-}sw}$". Please stick to one term.

L366-367 Desirable instead of "helpful"?

L367-369 "Instead, the addition of Na/Ca…"

L374-376 "In contrast, the Red Sea study…"

L377-379 As mentioned before, please use either "test" or "shell" in the manuscript.

L381-382 Reference Figure 3b here.

L387-438 As said in point 2) (Style and Language) please re-structure this section. I think it might be easier to list all potential factors first and then explain them in detail, as it is not clear until L392-393 what you mean with the first sentences. Please make separate paragraphs when you start discussing another potential controlling factor.

In L394-400 you discuss that several carbonate system parameter could influence the Na/Ca values and then in L406-409 you talk again about the influence of carbonate system parameters on Na/Ca ratios. Please combine these parts.

You also need to link your sentences better in this section, as sometimes (see listed below) it is not clear what you mean.

L390-392 Better to move "for the overlapping salinity intervals of 30 to 38" to the end of the sentence and insert the reference to Figure 7 right after "Allen et al. (2016)".

Start a new paragraph after "…further study." in L394 as you then start discussing the influence of the carbonate system parameter on Na/Ca ratios and not the impact of cleaning procedures.

L395-398 What are those "many co-varying factors"? Please give some example. Is it pH, DIC? Say this here and you have a nice link to your next sentence.

L398-400 Please insert "in the Red Sea" between "…foraminiferal Na/Ca ratios" and "(Mezger et al., 2016)" to make clear this still relates to the study in the Red Sea. Also start a new paragraph after this sentence as you go on about Na heterogeneity in tests.

L403-405 Please rephrase this sentence, it sounds odd. E.g., "Previous studies possibly provide a mixed Na/Ca signal because chamber wall calcite and spines were analysed together for elemental composition of foraminifera tests."

L406-407 Delete "also" after "…confidence interval)". Further, link this sentence better to the next as you jump from increasing [$CO_3^{2}$], DIC and pH to element partitioning and calcification rates. Hence, it is best to combine this paragraph (L406-409) with the paragraph in L394-400 where you actually say that an increase in DIC, pH, [$CO_3^{2}$] could cause higher precipitation rates.

L410-412 There is also the link between the 2 sentences missing. Here is a suggestion: "…micro-environment of foraminiferal tests, driven by photosynthetic activity of symbionts. This activity depends on nutrient concentration and light intensities and thus can actively change …"

L412-414 This sentence reads also weird. Maybe: "Higher photosynthetic rates influence test geochemistry which has been demonstrated by(?) boron…species, such as *O. universa*…"

L414-417 This sentence needs to be restructured. The part "although within…error of the mean" should go to the end of the sentence (between "specimens" and "(this study,..)"). Also, put a comma after *T. sacculifer*. However, I do not really understand what you mean to say here. Do you suspect different photosynthetic rates to explain that offset between culture experiments and surface sediments/in situ grown chambers? Then start a new paragraph after this sentence, you then discuss the influence of light intensities. Or better link it to this paragraph (L410-417).

L420-421 If the discussion of the light intensities belongs to the part before, start now a new paragraph as you talk about thermal stress. If the part about the intensities is meant to go with the following paragraph (L420-425) you need to better link these ideas, as I do not see a connection between light intensities and thermal stress.

Also, insert this entire sentence "We speculate…culture experiments." after the next sentence "Previous studies…(Edgar et al., 2013)." and maybe connect it with "Hence, we speculate…". Then maybe start the next sentence "This warming/thermal stress impacts the…".

L426-428 Delete "for example" and change "or" into "…Cd/Ca, and Ba/Ca…".

L428-430 Use a comma instead of semicolon to cite more than 1 publication of the same author (here, Regenberg). I would change this sentence a little bit "…report a decline in planktonic foraminiferal Mg/Ca below ~2500-3000 m…"

L431-438 Here, you do not need a new paragraph as you continue to discuss the effect of calcite dissolution. Again, use comma for "Regenberg et al. 2009, 2014)".

L432-433 This is no full sentence, there is the main clause missing or do not start this sentence with "as". Further, move "(± 0.25 mmol/mol)" to the end of sentence, just before the reference to your figure and table.

L433-435 better: "Although the calcite saturation state is quite different in bottom waters, we suspect…" Full stop is missing here.

L441-444 It is better to split this sentence into 2, using "alternatively" as the beginning of the second sentence. Better: "In this case, two…", "…accompanied by anions, such as Cl⁻…"

L444-447 Put "so far" after "…experiments of aragonite, so far, in which…"

L447-449 missing commas "…biogenic calcite, such as foraminifera…~5 mmol/mol), and aragonite, such as corals…" Further, link this sentence better to the next, there seems to be no connection between what you just said and the next sentence.

L451-453 "…with increasing(?) growth rates of inorganic calcite…" Do you mean it that way?

L454-456 It is better to give the $D_{Na}$ value of foraminifera first before you compare it to the inorganic value. Or start your next sentence saying that the inorganic $D_{Na}$ value is higher than your foraminifera equivalent and then give your calculated value.

L456-457 better: "…, using Na/Ca$_{seawater}$…"

L457-458 Who do you mean by "their"? Delaney et al. or Busenberg and Plummer? Please specify. Also: "…and of this study."

L458-460 Delete "listed in the following" and maybe put a colon after each $D_{Na}$ value instead of a comma, like this: "$D_{Na}$ = 0.1x10$^{-3}$: *T. sacculifer, G. ruber*…"

L460-461 Rephrase this sentence to: "The highest $D_{Na}$ of 0.18-0.25x10$^{-3}$ were found in the field calibration study…" or "Mezger et al. (2016) found the highest $D_{Na}$ of 0.18-0.25x10$^{-3}$ in their field calibration study…".

L463-467 Can you please restructure these 3 sentences? I was confused about what you mean until you mention the influence of the gametogenic calcite in the last sentence (L466-467). Maybe start with this last sentence and then that you deduced from the presence of the gametogenic calcite that there has to be a biological control on the Na incorporation.

L468-469 "In comparison to foraminiferal $D_{Na}$, the partition …" Please give a value or a range for the $D_{Mg}$.

L474 Maybe better to use "requires" instead of "affords".

L477-479 There is a verb missing in the last part of that sentence. Please rephrase, e.g., "…in seawater are only affected by temperature changes to a limited extent, and hence temperature has minimal influence(?) on Na incorporation…" if I understand your last part correctly.

L484-485 Delete opening parenthesis before 2016: "(Allen et al. 2016)".

L485-487 "In our experiments, we…" I also think your intra- and inter-test variability is 2-16%.

L520-719 References: Maybe this is an issue of converting your file into pdf, but could you please use indentation to separate individual references? The way it is at the moment is hard to read.

Table 1 Can you please change mmol/mol into mmol mol$^{-1}$ to be consistent?

Table 3 Please rephrase the sentence "Chambers grown in situ…" because it sounds like that the chambers that grew in situ are different to those which grew in the natural environment. But they are the same, that is at least what you say in the manuscript text.

Figure 1 Can you please insert a line at 30 m in your plots 1c, 1d to make a direct comparison easier between the hydrography plots and surface maps.

Figure 3 caption, better: "(19.5 – 29.5°C)" or "(19.5°C – 29.5°C)" Further, please mention here that the experiment at salinity 44 is excluded because of the precipitation of GAM calcite.

Figure 4 "Related raw data of salinity experiments…" better to say "Related data of individual measurements of the salinity experiments…" because the data presented in Table 3 are not raw data per se (Raw data would be the measured intensities for each element.) but your final, calculated data.

Please say: "Datasets for T = 26.5°C of both temperature experiments (S 33, S 36) were added (Tables 4, 5)."

Insert: "Horizontal error bars of samples from the Gulf of Guinea demonstrate…" But why only for those sediments? What about the surface sediment samples from the Caribbean Sea? Is the error as big as the symbol size? If yes, please say so.

Move your sentence "Data marked by yellow,…" to after you explained those symbols, that is "…Caribbean surface sediments (Table 1)." Those data is not listed in Table 2. Also reference Table 3 for the in situ chambers and Table 1 for the Gulf of Guinea samples.

Figure 5 Similar to Figure 4: better to say "Related data of individual measurements…" instead of "Related raw data…". Also, "Tables 4 and 5"

The same as in Figure 4: "Horizontal error bars of samples from the Gulf of Guinea demonstrate…" Why only for those sediments?

Figure 6 Can you please label the dashed line in your plot with "95% confidence interval" to make it easier to read this plot?

Figures 4-6 This is a very personal thing, but maybe consider to present the numbers on your axes with the same decimal digits, e.g. 8.0; 7.5; 7.0;…

Figure 7 Please be consistent and use as font style regular instead of italics for "versus", which you otherwise use in the entire manuscript. Further move "versus salinity" in the first sentence to the end of the sentence, as it is confusing, or incorporate "versus salinity" into a new sentence, e.g., delete from the first sentence and then say "Foraminiferal Na/Ca values are shown versus/against salinity."

Better "Regression lines are based on the following equations: Wit et al.,…"

The equations are given in R and $R^2$. Is this correct?

Figure 8 Please give the salinity of the other studies (Allen et al., Mezger et al.) in the legend of this plot for a direct comparison.

Similar to Figure 7: Move "versus temperature" in the first sentence to the end of this sentence or make a new sentence, e.g., delete from the first sentence and then say "Foraminiferal Na/Ca values are plotted versus/against temperature."

Table A1 caption "…listed in Tables 3, 4, and 5."

Table B1 caption "…averaged maps (Tables 3, 4, 5)…" Further, please be consistent and use "Intra-test" and "Inter-test" in the header of your table, like you do in the text.

It would improve the readability of this table if you insert a header for the salinity and the temperature experiments like you do for the in situ grown chamber and maybe separate the different experiments/data by horizontal lines.

---

## Author Comment (AC1) · 20 Jul 2018

Dear Editor,

We kindly thank two anonymous reviewers for their positive feedback, very constructive and thoughtful comments, which greatly helped to improve our manuscript. Below you find our responses to each point the reviewer's addressed and how we incorporated all suggestions in our revised manuscript, thereby following the given structure of their comments. Referee comments are written in italics and the respective answers are

given in normal font.

Yours sincerely, Jacqueline Bertlich and on behalf of all co-authors

Answers to anonymous Referee #1

Summary and general comments:

Bertlich et al., make use of earlier studies that introduced Na/Ca ratios from foraminifera as a proxy for surface seawater salinity, and they assess the reliability of Na/Ca derived from T. sacculifer as a salinity proxy. They conclude that Na/Ca can be applied as a reliable proxy for reconstructing sea surface salinity and furthermore that species-specific calibrations are necessary.

This study is one step closer towards establishing Na/Ca as a proxy for SSS. It is a well-designed study making use of culture experiments, wherein they vary salinity keeping temperature constant or vice versa and measure the Na/Ca to understand the salinity and temperature effects on Na/Ca. They further corroborate the culture studies with surface sediments from two different locations, Caribbean and Gulf of Guinea. This study has the potential to highlight the efficiency of Na/Ca as a salinity proxy which may soon be followed by other studies that may target different foraminiferal species and oceanic locations so as to develop this proxy further into a robust one. This proxy can then be utilized in unison with d18O and in some cases, Ba/Ca, to get a better handle on absolute salinity values, as each proxy has its own limitations. Here, the authors have touched a few important environmental variables that may affect and are needed in proxy validation. However I urge the authors to consider the following issues that I have raised and incorporate them in the manuscript.

Specific comments:

The authors mention that in order to avoid dissolution they have chosen sampling sites where $\Delta[CO_3^{2-}]$ is >30 $\mu$mol/kg. But, if samples from water depths deeper than 2.7 km (see Table 1) are plotted against Na/Ca a clear depth dependency is noticed, Na/Ca de-

creasing with increasing depth. The authors should make this explicit to the reader by providing the Δ[CO32–] for each sample/depth and a separate paragraph discussing the depth dependency/dissolution effect on Na/Ca.

We totally agree with the referee and see the need to additionally present the Δ[CO32–] values for each sampling location to examine in detail a possible effect of calcite dissolution on foraminiferal Na/Ca. This point was also addressed in the second review. Therefore, we added the calcite saturation state for each sampling location in Table 1 (L903) and also plotted Na/Ca values with increasing water depth and the respective Δ[CO32–] values for the Caribbean and the Gulf of Guinea (Figure 6b – L1006). According to our new graphs, we would claim to see a clear depth dependence of foraminiferal Na/Ca, since error bars of all locations above 4 km intersect with each other. The issue of selective Na+-removal due to dissolution at greater water depth, nonetheless, needs further investigation, because of the limited sample set of our study. All points mentioned above are additionally added to section 2.3 in Material and Methods (L154-179) and in the results (341-363) and discussion (L532-542).

The range in spatial salinity distribution at the surface sediment locations is ∼1 to 1.5 salinity units. However the vertical distribution in salinity which may be encountered by T. sacculifer at its habitat, especially at the Gulf of Guinea is large, reaching ∼4 salinity units. Planktic foraminifera live for few weeks and so may encounter ambient conditions, which are seasonal during their life cycle (eg: Honisch et al., 2013 GCA). So when applying the Na/Ca proxy to down core samples to environments such as the Gulf of Guinea, how one does take into account such large vertical variations. This signal will also be mixed with the seasonal signal. Ideally surface samples should have covered a larger range in surface salinity.

This is indeed a very important point the referee mentioned here regarding future applicability of Na/Ca as a salinity proxy for down core records. We totally agree that seasonal shell flux patterns, which are significantly different between various planktonic foraminifera (e.g. see Jonkers and Kučera, 2015), should take into account when

applying Na/Ca as a paleo-salinity proxy. But for T. sacculifer it is well known that this species is present throughout the year and shell fluxes do not vary significantly with seasonal changes in the tropic and sub-tropic water masses, when the annual SST is >25 °C (Bijma et al., 1990a, b; Bijma and Hemleben, 1994; Schmuker and Schiebel, 2002; Lin et al., 2004; Jonkers and Kučera, 2015) (L91-93). Instead, G. ruber shows preferential seasonal peaks (Hönisch et al., 2013 GCA). But especially cold-water species show prominent peaks throughout the year, strongly dependent with seasonality and changing temperatures, respectively (Jonkers and Kučera, 2015). Further, a recent publication of Raitzsch et al. (2018) have demonstrated how information on the population dynamics of foraminiferal species can be used to reconstruct seasonal variability of environmental parameters for future application. Furthermore, the combination of foraminiferal Na/Ca and the Ba/Caforam – sea surface salinity relationship, which is limited to regions close to rivers, could provide additional information about vertical variations, especially in the Gulf of Guinea. But according to the large vertical variations of salinity e.g. in the Gulf of Guinea and also the wide range of vertical migration of foraminifera in their habitat, eventual Na/Ca results potentially represents a depth-averaged salinity, rather than that of the surface. Here, the application of a multi-proxy approach and measurements on a variety of planktonic foraminifera (covering certain depth habitat intervals) could limit the large salinity range.

Technical corrections:

Line 74: Do you mean G. sacculifer?

Actually both terms, G. sacculifer and T. sacculifer, would be correct in this case, as both morphotypes are the same biological species and genetically identical (Hemleben et al., 1987; Bijma et al., 1994; André et al., 2013; Spezzaferri et al., 2015). The previously used species Globigerinoides sacculifer just comprised both, the trilobus and sacculifer morphotype, without and with a sac-like chamber (Bijma et al., 1994; André et al., 2013).

Line 89: 'related' should rather be 'correlated'? We changed it accordingly (now Line 95).

Line 92: SCUBA is written in upper case, is it an acronym?

Uppercase letters were already previously used in the initial description of culture experiments in Bijma et al. (1990b) and Nürnberg et al. (1996). But to avoid misunderstandings, as the word SCUBA does not implicate an acronym, we changed it to "scuba" (L98).

Line 95: Salinity experiments were done at salinities of 23, 26, 41, 44 and 45, which do not encompass the entire salinity range of 23 to 45. Why are the mid ranges not included?

In case you mean here that culture experiments at salinities of $S = 33$ and $S = 36$ are missing, they are not listed at salinity experiments, because initially culture experiments were separated. We additionally included results of temperature experiments from both treatments S 33 and S 36, were foraminifera were cultured at the overlapping temperature of 26.5 °C. We added this information to the text (L253-254).

Line 100: '26.5 °C to 29.5 °C' OR '26.5 °C and 29.5 °C'? We meant 26.5 °C and 29.5 °C and changed it accordingly.

Line 114 and 133: any reason for using the two different size fractions? 315 to 400$\mu$ and 300 to 400$\mu$?

There is no specific reason using two different size fractions for surface sediment samples. This was simply related to slight variations in mesh sizes of sieves between different research groups, as Steffanie Nordhausen measured the Gulf of Guinea sediment surface samples.

Line 138: 'The annual SSS varies from east to.........'. The variation of SSS from 32 to 35.9 is spatial or with depth?

This is spatial, but we also clarified this in the revised manuscript (L149-151).

Line 140 to 141: $\Delta[CO3^2-]$ = should be calculated for respective depths and included in tabular form.

This has been added to the manuscript. Please find the information in Table 1 (starting at L903). We additionally prepared a new Figure (6b, L 1006)

Line 221: Na/Ca increases by 0.12 mmol/mol per salinity unit. However the error on Na/Ca measurements at many instances is beyond this value (see table 1).

This point was also addressed from the second referee and we solved this issue by additionally providing the number of specimens needed to resolve a certain salinity change, based on our Na/Ca to salinity calibration, and added a figure (Appendix C, L1060) to clarify this relationship. Concluding from that, we could proof with our surface sediment samples (Table 1) that at least 20 specimens are necessary to resolve past salinity changes between 0.5 to 1 (see L441-456).

Line 273: 'Nevertheless, it is still a ... noticeable'. Should it read as.....'..... ..which is noticeable?' Yes, we changed this accordingly.
* * *
Figure1b: As seen from the figure, the GIK samples seem to fall within 35 and 36 salinity units. However the figure 1c shows a larger range in salinity, from $\sim$31.7 to 34.7 units.

This is correct, because Figure 1b only presents the annual salinity at 30 m, the averaged water depth habitat we assumed for T. sacculifer in this study. Figure 1c shows instead the entire range of parameters within depth.

Figure 3a: Why is the 5th box, at salinity value of 44, grey in color? The text describes the salinity values for culture experiments of 23, 26, 41, 44 and 45 salinity units. The figure however shows 26, 33, 36, 41, 44 and 45 salinity units.

Maybe this was due to a converting issue, but actually the 5th box in Figure 3a should

be colored in black, equivalent to the color code in Figure 4. But we additionally added this information to the figure caption (L973-974). Further, we deleted the salinity experiment S 23, because the final, in clture newly precipitated chamber broke (too thin) and was removed due to the cleaning procedure for electron microprobe measurements. We also included the information that datasets for T = 26.5°C of both temperature experiments (S33, S36) were added to Figure 3a (L972).
* * *

**Supplement:**

[revised manuscript text omitted]
)}$ = 0.57 · S − 12.38 ($R^2$ = 0.91, $p$ <0.001); Na/Ca$_{T. sacculifer}$ = 0.60 · S − 13.49 ($R^2$ = 0.99, $p$ <0.001); Our study: Na/Ca$_{T. sacculifer}$ = 0.97 + 0.115 · Salinity (R = 0.97, $p$ <0.005). Vertical error bars are based on the standard error of the mean and horizontal error bars of surface sediments (as big as symbols for Caribbean samples) demonstrate the salinity gradient between 0–60 m water depth, the possible habitat range of *T. sacculifer*. Different regression lines for the same species, e.g. *T. sacculifer*, demonstrate various potential factors, and not only salinity, controlling Na incorporation in foraminiferal calcite (discussed in Sect. 4.3). However, our results of culture experiments correspond to fossil samples of surface sediments from two regions suggesting this regression is the most reliable for paleo-reconstructions.

[Figure]

1035

**Figure 8.** Comparison of foraminiferal Na/Ca values in response to temperature changes between this study´s *T. sacculifer* from culture experiments with published Na/Ca data of planktonic foraminifera from Allen et al. (2016) and Mezger et al. (2016). Averaged Na/Ca values of our study are shown in two different settings at either a constant salinity of 33 (grey dots) or 36 (light blue squares). During culture experiments from Allen et al. (2016) the salinity slightly decreased from 35.6 to 35.2 with increasing temperatures from 24 °C to 29.3 °C. Although in the field calibration study of Mezger et al. (2016) Na/Ca values apparently decrease with increasing temperature, corresponding salinities also decrease from 39.6 to 36.8. Vertical error bars are based on the standard error of the mean. Except for specimens from the Red Sea, influenced by co-varying factors, (*G. ruber*: $R^2 = 0.84$, *p* <0.001; *T. sacculifer*: $R^2 = 0.95$, *p* <0.001; Mezger et al., 2016), all Na/Ca data for *T. sacculifer* and *G. ruber* show no significant response to temperature changes (our study; Allen et al., 2016).

1040

**Appendices**

**Appendix A**

**Table A1.** Na/Ca values generated for each experiment were tested for normality with the Shapiro-Wilk W-test and its corresponding $p$-value. If $p$ is $\geq 0.05$ (95 % confidence interval), the data are normally distributed. The number of measurements is indicated with n. Salinity and temperature (T) values derive from culture experiments listed in Tables 3, 4 and 5.

| Salinity | T (°C) | n | W | $p$ |
|---|---|---|---|---|
| 26 | 26.5 | 20 | 0.962 | 0.587 |
| 41 | 26.5 | 34 | 0.953 | 0.590 |
| 44 | 26.5 | 24 | 0.972 | 0.709 |
| 45 | 26.5 | 12 | 0.878 | 0.083 |
| 33 | 19.5 | 26 | 0.960 | 0.399 |
| 33 | 23.5 | 31 | 0.952 | 0.183 |
| 33 | 26.5 | 29 | 0.971 | 0.590 |
| 33 | 29.5 | 29 | 0.941 | 0.108 |
| 36 | 19.5 | 27 | 0.964 | 0.462 |
| 36 | 23.5 | 24 | 0.960 | 0.440 |
| 36 | 26.5 | 26 | 0.975 | 0.767 |
| 36 | 29.5 | 24 | 0.951 | 0.286 |

**Appendix B**

**Table B1.** Inter- and intra-test Na/Ca variability of cultured *T. sacculifer* with varying experimental settings. The Na/Ca intra- and inter-test variability is calculated from single averaged electron microprobe maps (Tables 3, 4, 5) within one foraminifer (#) and between foraminifera (Na/Ca$_{average}$ ±standard error of the mean) from the same experiment. The variability is expressed as the relative standard deviation (RSD) in %. Annual salinity and temperature data for in situ grown foraminiferal calcite (F-3, F-4) in the open ocean were taken from WOA13 (Zweng et al., 2013; Locarnini et al., 2013).

| Sample # | Intra-test variabilty RSD (%) | Inter-test variabilty RSD (%) | Temp (°C) | Salinity | Na/Ca$_{average}$ (mmol mol$^{-1}$) |
|---|---|---|---|---|---|
| **Salinity experiments** | | | | | |
| 7912 | 10.39 | 6.52 | 26.5 | 26 | 4.36 ±0.20 |
| 7913 | 10.23 | | 26.5 | 26 | 3.96 ±0.14 |
| 7914 | 11.74 | | 26.5 | 26 | 3.86 ±0.17 |
| 8135 | 13.27 | 5.51 | 26.5 | 41 | 5.66 ±0.20 |
| 8137 | 11.06 | | 26.5 | 41 | 5.49 ±0.16 |
| 8138 | 7.44 | | 26.5 | 41 | 5.08 ±0.15 |
| 7703 | 12.22 | 6.16 | 26.5 | 44 | 4.83 ±0.18 |
| 7704 | 10.73 | | 26.5 | 44 | 5.27 ±0.16 |

| | | | | | |
|---|---|---|---|---|---|
| 8301 | 10.40 | – | 26.5 | 45 | 6.40 ±0.19 |

**Temperature experiments (constant S 36)**

| | | | | | |
|---|---|---|---|---|---|
| 1.2 | 14.18 | 9.15 | 19.5 | 36 | 4.24 ± 0.19 |
| 1.3 | 12.38 | | 19.5 | 36 | 4.78 ± 0.24 |
| 1.4 | 12.57 | | 19.5 | 36 | 5.09 ± 0.20 |
| 2.1 | 7.50 | 7.84 | 23.5 | 36 | 4.74 ± 0.16 |
| 2.2 | 3.53 | | 23.5 | 36 | 5.41 ± 0.06 |
| 2.3 | 16.23 | | 23.5 | 36 | 4.99 ± 0.33 |
| 2.4 | 8.17 | | 23.5 | 36 | 4.51 ± 0.37 |
| 3.1 | 6.36 | 7.35 | 26.5 | 36 | 4.62 ± 0.15 |
| 3.2 | 5.32 | | 26.5 | 36 | 4.86 ± 0.10 |
| 3.4 | 3.98 | | 26.5 | 36 | 5.00 ± 0.07 |
| 3.5 | 1.83 | | 26.5 | 36 | 5.49 ± 0.05 |
| 4.1 | 5.38 | 9.81 | 29.5 | 36 | 4.80 ± 0.08 |
| 4.3 | 12.71 | | 29.5 | 36 | 3.96 ± 0.23 |
| 4.4 | 8.60 | | 29.5 | 36 | 4.26 ± 0.12 |

**Temperature experiments (constant S 33)**

| | | | | | |
|---|---|---|---|---|---|
| 5.1 | 4.76 | 3.21 | 19.5 | 33 | 5.37 ± 0.10 |
| 5.2 | 9.96 | | 19.5 | 33 | 5.13 ± 0.15 |
| 5.5 | 5.27 | | 19.5 | 33 | 5.05 ± 0.09 |
| 6.1 | 3.58 | 10.35 | 23.5 | 33 | 5.53 ± 0.07 |
| 6.2 | 6.72 | | 23.5 | 33 | 4.53 ± 0.12 |
| 6.3 | 9.29 | | 23.5 | 33 | 5.16 ± 0.21 |
| 6.5 | 6.28 | | 23.5 | 33 | 5.78 ± 0.11 |
| 7.1 | 6.35 | 7.43 | 26.5 | 33 | 4.33 ± 0.10 |
| 7.3 | 4.91 | | 26.5 | 33 | 5.05 ± 0.09 |
| 7.4 | 7.23 | | 26.5 | 33 | 5.05 ± 0.14 |
| 7.5 | 10.54 | | 26.5 | 33 | 4.61 ± 0.20 |
| 8.1 | 8.35 | 11.47 | 29.5 | 33 | 4.39 ± 0.13 |
| 8.3 | 9.24 | | 29.5 | 33 | 4.10 ± 0.13 |
| 8.4 | 8.91 | | 29.5 | 33 | 4.43 ± 0.14 |
| 8.5 | 7.80 | | 29.5 | 33 | 5.31 ± 0.21 |

In situ grown chambers (F-3, F-4)

| | | | | | |
|---|---|---|---|---|---|
| 4.1 | 5.63 | 1.69 | 27.4 | 35.9 | 5.27 ±0.12 |
| 2.3 | 3.25 | | 27.4 | 35.9 | 5.43 ±0.18 |
| 7704 | 1.66 | | 27.4 | 35.9 | 5.48 ±0.04 |
| 7703 | 3.85 | | 27.4 | 35.9 | 5.34 ±0.08 |

**Appendix C**

[Figure]

**Figure C1.** Standard errors (standard deviation $\sigma/\sqrt{n}$) of Na/Ca values from cultured *T. sacculifer* for each salinity experiment (S) related to salinity uncertainties. Salinity calculations are based on our calibration (Salinity = ((Na/Ca$_{T. sacculifer}$ − 0.97)/0.115)) **(a)** Number of total specimens needed to resolve a certain salinity range. **(b)** Number of single spot measurements by electron microprobe analysis within chamber wall cross sections are required to resolve a certain salinity.

1060

1065

---

## Author Comment (AC2) · 20 Jul 2018

Dear Editor,

We kindly thank two anonymous reviewers for their positive feedback, very constructive and thoughtful comments, which greatly helped to improve our manuscript. Below you find our responses to each point the reviewer's addressed and how we incorporated all suggestions in our revised manuscript, thereby following the given structure of their comments. Referee comments are written in italics and the respective answers are given in normal font.

Yours sincerely,
Jacqueline Bertlich and on behalf of all co-authors

**Answers to anonymous Referee #2**

Summary and general comments:

*Bertlich and co-authors present an interesting and important study for proxy development by using Na/Ca ratios of the planktonic foraminifera Trilobatus sacculifer to reconstruct salinity using electron microprobe analyses. This study is well designed as the authors present a foraminiferal Na/Ca-salinity calibration curve which is based on culture experiments and compare this curve with actual surface sediment samples from two different tropical Atlantic regions. The culture experiments comprise a wide range of salinity and also test the influence of temperature on Na incorporation into foraminiferal tests. Further controlling factors of Na incorporation are also explored briefly. The results of the surface sediments agree well with the culture experiments which validate the calibration curve presented in this study.*
*The manuscript is mostly well written, logically organised and clear. The figures are overall nice and clear. I think that this work is an interesting and important contribution and therefore suitable to be published in Biogeosciences. Nevertheless, I would like to see the points below addressed by the authors.*

**Main points**

*"1) You mention in the manuscript that your intra-test variability of the temperature experiments is less than half than that of the salinity experiments. I am extremely sorry, but I strongly disagree. According to Table B1, the variability between both experiments is similar. The average intra-test variability of the salinity experiments is 10.82% RSD, the half of this is 5.41% RSD, but only 9 out of 27 individual foraminifera results of the temperature experiments have actually lower RSDs than 5.41%. So please change those sentences (L260-262, L382-384) or make clear what you mean."*

We agree that differences in the inter- and intra-test variability between salinity and temperature experiments are minor and have removed this false statement. Instead, in the revised manuscript we point out that the smallest intra-test variations are observed for cultured foraminifera close to conditions of their natural environment (L291-293 and L395-397).

*"2) Although the data is good and relevant, the authors did not convince me of their results in the discussion, especially Sections 4.3 and 4.4. The authors introduce an interesting train of thought but then did not think it through, instead they start a new line of thought or connect one idea with another seemingly unrelated. I really like the first section of the discussion because the influence of biological processes/response is brought into focus which is important for this study, considering that Na+ ions are essential for ion channels/pumps and for the general functioning of cells and could therefore influence/bias the incorporation into calcite tests. However, Section 4.3 did not address these and other biological aspects further and the reader is left with a lot unanswered questions. Furthermore, the authors jump back and forth in this section, which makes it hard to follow each line of thought. Your data is great! You just need to sell it better."*

We thank the referee for this insightful comment and have added information in our revised manuscript on how the transport of ions to the calcification site via different mechanisms could also possibly influence Na$^+$ incorporation. Hence, we followed the advice to restructure and clarify discussion Sections 4.3 and 4.4, to more clearly link possible biological and/or environmental factors controlling Na incorporation into calcite tests. These sections are completely rewritten in the revised manuscript (Discussion 4.3 (L468-L542) and 4.4 (L543-600).

*"3) Your calibration indicates an increase in Na/Ca ratios by 2.25% per salinity unit. However, you intra- and inter-test variability is much higher than that ranging from 1.83-16.21% and 3.19-11.51% (according to Table B1) when using electron microprobe (or other micro-analytical techniques). Could you please give an estimate on how many individual measurements are needed to distinguish ± _0.5-1 salinity units analytically and statistically?"*

This is indeed a very important point the reviewer mentioned here, regarding the future applicability of Na/Ca for paleoceanographic reconstructions. The deviation around the mean is strongly dependent on the number of analyses and total specimens used for analysis because of inter- and intra-test variability of the element/Ca values, especially for Mg/Ca e.g. Sadekov et al. (2008) or De Nooijer et al. (2014a). Accordingly, we now provide estimates of the number of specimens needed to resolve a certain salinity change, based on our Na/Ca to salinity calibration, and added a figure (Appendix C, L1060) and discussed it in the text (L441-456) to clarify this relationship.

[Figure]

**Figure C1.** Standard errors (standard deviation σ/√n) of Na/Ca values from cultured *T. sacculifer* for each salinity experiment (S) related to salinity uncertainties. Salinity calculations are based on our calibration (Salinity = ((Na/Ca$_{T. sacculifer}$ − 0.97)/0.115)) **(a)** Number of total specimens needed to resolve a certain salinity range. **(b)** Number of single spot measurements by electron microprobe analysis within chamber wall cross sections are required to resolve a certain salinity.

*"4) You say that dissolution and early diagenetic processes do not influence the foraminiferal Na/Ca ratio of your surface sediments, however, you state in the Material and Method section that you only chose sites, where Δ[CO$_3^{2-}$] is > 30 µmol/kg which in turn is above the critical calcite saturation value given by Regenberg et al. (2006, 2009). Therefore, please explain how you can address the effect of dissolution on Na/Ca ratios."*

As no studies currently exist about the calcite dissolution effect on Na/Ca in fossil tests, we assumed a similar trend to the start of selective ion-removal of Mg$^{2+}$ in planktonic foraminifera below the critical calcite saturation state of ~21 µmol kg$^{-1}$, presented by Regenberg et al. (2006, 2014). But we agree and see the need to additionally present the Δ[CO$_3^{2-}$]values for each sampling location to allow examination of a possible effect of calcite dissolution on foraminiferal Na/Ca. Therefore we extended Section 2.3 in the Material and Methods part (L154-179) and added the according information to the results (L341-363) and discussion (L532-542).

*In this regard, it would be of benefit if you could insert the $\Delta[CO_3^{2-}]$ values for your stations in Table 1. Have you tried to plot your Na/Ca values from Table 1 against $\Delta[CO_3^{2-}]$ to see if there is a relationship?*

We added the calcite saturation state for each sampling location in Table 1 (starting at L903) and also plot Na/Ca values with water depth and the respective $\Delta[CO_3^{2-}]$ values for the Caribbean and the Gulf of Guinea (Figure 6b – L1006).

[Figure]

**Figure 6.** Averaged Na/Ca values of *T. sacculifer* from surface sediment samples collected in the Caribbean (stations SO164−01−3 to −50−3, green triangles) and the Gulf of Guinea (stations M6−5 GIK 16808−16869, blue diamonds) (Figure 1, Table 2). **(a)** Dashed lines indicate the 95 % confidence interval derived from culture experiments (Figure 3) for comparison. Vertical error bars denote the standard error of the mean derived from replicate measurements (Table 2). Horizontal error bars visualize the salinity range from 0-60 m at the respective core locations (Figure 1), representing the possible habitat migration of *T. sacculifer* (Hemleben et al., 1987; Schmuker and Schiebel, 2002). **(b)** $\Delta[CO_3^{2-}]$ versus water depth at respective sampling stations (red dots), presented in Table 1, and the entire depth profile (light grey dots) in the Gulf of Guinea and the Caribbean. $\Delta[CO_3^{2-}]$ is defined as the difference between the in situ carbonate-ion concentration ($[CO_3^{2-}]$) and $[CO_3^{2-}]$ at calcite saturation. Respective calcite saturation states for the Caribbean were taken from Regenberg et al. (2006) due to overlapping sample stations and for the Gulf of Guinea from Weldeab (2012) and Weldeab et al. (2016). Additional data were calculated with the program CO2SYS by Pierrot et al. (2006) (see section 2.3 for detailed description). The horizontal calcite saturation horizon line marks the top of the present day lysocline, where $\Delta[CO_3^{2-}]$ is 0 μmol kg$^{-1}$ (Regenberg et al., 2006). The vertical line demonstrates the critical $\Delta[CO_3^{2-}]$ threshold of 21.3 μmol kg$^{-1}$, the onset of selective Mg$^{2+}$-ion removal due to calcite dissolution in all planktonic foraminifera (Regenberg et al., 2006, 2014, section 2.3).

*Further, did you check (visually, by SEM) the tests of the surface sediments for any signs of dissolution/early diagenesis?*

We only checked foraminifera from surface sediment samples visually with a binocular microscope (L123-125), but did not additionally control calcite tests by a scanning electron microscope or electron microprobe technique. But this should be kept in mind for future studies, applying Na/Ca for paleoceanographic reconstructions.

*I guess, this is difficult to address with EMP, but do you see differences in test wall thickness, test size or thickness of primary calcite between specimens from shallower and deeper stations in the Caribbean Sea (similar to the results from Johnstone et al. 2011, Sadekov et al. 2010)? Assuming you did measure some tests by using EMP . . .*

We did not check additionally surface sediment samples with electron microprobe technique, because we mainly focused on the applicability and reliablity of Na/Ca as a paleo-proxy, were it is very uncommon to use single foraminiferal analysis. But to avoid a possible effect of ontogenetic variations and may increasing amounts of Na/Ca with foraminiferal test size, as it is known for Mg/Ca (e.g. Elderfield et al., 2002), we chose a narrow size fraction (see L122-123).

**Minor points**

*L24-25 better to say "...indicating salinity to be/is one of the dominant factors…" as I think the biological component on Na incorporation into calcite is still unclear.*

We changed this accordingly (L25).

*L26-27 Considering that the biological influence is still unclear, maybe replace "reliable proxy" with "potential proxy".*

We changed this accordingly (L26).

*L74-76 It would be interesting to see if T. sacculifer and T. triloba actually have a similar geochemical signature. Did you test this? If so, could you please provide these data in the supplements? This would then be the first time such a comparison has been done.*

Unfortunately we did not measure both morphotypes, because culture experiments were only carried out with one morphotype.

*L91-94 Were the (SCUBA dive-) collected specimens of T. sacculifer juveniles or young adults? Please indicate.*

We added the information that test length of collected foraminifera thus sampled did not exceed ~110−500 μm, but only specimens with a similar test size were chosen for each experiment (Bijma et al., 1990b) (L100-101).

*L104-105 Your "slight" increase in salinity of 0.5-0.8 units during the 29.5°C experiment is actually the salinity range you say you can resolve with your calibration! Please delete "slight". What does this mean for your error bars in Figure 5 if you include the related uncertainty?*

We omitted "slight", but a salinity increase of 0.5-0.8 at T=29.5 °C did not result in a larger spread of values compared to the other experiments.

*L113 Please insert that the water depth is also given in Figure 1 and Table 1 as this is important concerning the preservation state of the tests.*

This has been added (L120).

*As mentioned in the main points, could you please include the $\Delta[CO_3^{2-}]$ values for your stations in Table 1?*

As already written in Main point 4, we provided the calcite saturation state for each bottom water depth at respective sampling locations in Table 1 (L903).

*L196-198 Can you please give the concentration of your hydrazine solution?*

This has been added, but instead of the concentration we gave the molarity now (L229-231).

*L211-212 This sentence reads odd. Do you mean your measured value is 19.79 ± 0.51 mmol/mol? If not, which reference did you use and what is your analysed value for Na of the JCp-1?*

This has been clarified in the revised manuscript and we clearly separated our measured JCp-1 Na/Ca values in comparison to the reference material (L245-249).

*L213-214 When you say "every sample solution was measured 5 times", do mean your foraminifera samples only or also the JCp-1? Please be more precise.* This has been adjusted accordingly.

This has been clarified in the revised manuscript. The measurement routine repeats 5 times and the RSD of these 5 numbers gives an estimation of internal error (L241-242).

*L233-234 Reference Figure 3a to be precise* (adjusted). *You also have a comparably large range in your temperature experiments 19°C < x < 24°C for salinity 36 according to Figure 3b and Table B1. Please mention this here or in the section about the temperature experiments.*

We changed this accordingly and mentioned it in the section about temperature experiments.

> L291-293: Notably, the highest Na/Ca intra-test distribution (up to 16 %) only occurs in experiments below 24 °C (S 36), which is in contrast to the smallest intra-test variation of 1.8 %, observed for treatments close to the natural habitat conditions of *T. sacculifer* (S 36, 26.5 °C; Figure 3b, Appendix B).

*L260-262 As mentioned in the main points, please delete this sentence as this is not true according to Table B1: Only 9 experiments out of 27 have actually lower intra-test variability.*

We deleted this accordingly.

*L269-270 The value of 4.71 ± 0.21 mmol/mol is not listed in Table 1, the lowest value in Table 1 is 4.64 ± 0.25 mmol/mol.*

We originally averaged the Na/Ca values of both stations mentioned in this line, but we agree that this is confusing and therefore provide the values separately in the revised manuscript (L312).

*L281-283 Please explain briefly what the salinity uncertainty of the Gulf of Guinea surface sediments means, illustrated by horizontal error bars, as this is only mentioned in the caption of Figure 4. Why is there no such uncertainty for the sediment samples from the Caribbean Sea? Is the error bar as big as the symbol? If yes, please say so.*

We clarified that these represent the variability in the upper 0-60 m of the water column, which is the main depth habitat of *T. sacculifer* in the Caribbean (L306-309) and the Gulf of Guinea (L330-332):

*L285-287 Could you please give an explanation why the stations GIK 16860-1/16864-1 have higher Na/Ca values than station GIK 16865-1, although they are all close to the river mouths, similar to your explanation in L270-273?*

We mentioned this observed differences of Na/Ca values from stations proximal to the river mouths in the results (L335-337, L360-363) and discussed possible explanations, related to bottom water depths of sampling sites and their respective calcite saturation state (L537-539).

*L293-294 According to Table B1 your Na/Ca intra- and inter-test variability varies between about 2-16% (lowest value: 1.83%).*

We changed this accordingly (L370).

*L300-303 You can also reference Hathorne et al. 2009 here.*

We added this reference.

*L303-305 Please reference Spero et al. 2015 here.*

We changed this accordingly.

*L306-309 I do not think that Figure 4 shows that the lowest intra-test variation is obtained at conditions close to*

*those at the natural habitat. Reference only Figure 3 here.*

Indeed, this is more obvious just from Figure 3 (now L397-399).

*L306-314 I really like that you discuss here the possible biological response to increased stress levels. You say that increased stress levels could lead to higher intra-test variability. Could that have influenced your Na/Ca values at low and high salinities (like seen in Figure 3 – high variability at T < 24°C and at S ≥ 41)? Could this, in turn, result in a lower or steeper slope of your calibration curve?*

The field generally uses culture experiments to show how planktonic foraminifera react to large or extreme parameter changes and transfer this knowledge to paleo-reconstructions. As results of our culture experiments close to conditions of their natural habitat (T=26.5 °C, S 33-36) agree with results of measurements on in situ grown chambers, we assume that foraminifera would react similar to extreme changes in their natural habitat (caused e.g. by abrupt river or meltwater discharges, glacial- inter-glacial temperature changes). Therefore, we would say that our established calibration curve would not change towards a lower or steeper slope.
This is supported by previous studies, which inferred from culture experiments the possible loss of foraminiferal symbionts due to thermal stress (Van Dam et al., 2012). Edgar et al. (2013) could also proof these observations in the paleorecord, where photosymbiont-bearing planktonic foraminifera were obviously 'bleached', induced by "transient global warming during the Middle Eocene Climatic Optimum".

*L328-330 Mention here, that you assume that the observation of Branson et al. (2016) could also be valid for T. sacculifer and other planktonic species.*

This has been added to the manuscript (L390-392).

*L347-349 Please indicate here how many individual measurements are necessary to achieve resolving a salinity change of about 1 unit using EMPA, as addressed in main point 3).*

We calculated the number of specimens needed to resolve a certain salinity (L441-456).

*L350-351 What do you exactly mean with "little small-scale variability"? The intra- or inter-test variation itself or compared to the higher variability of Mg/Ca?*

We clarified this term (L454-456).

*L356-357 As said before and in main point 3): Please state the minimum number of individual analyses to actually resolve a salinity change of ± 0.5-1 units?*

We added this information accordingly (L462-463)..

*L379-380 I would not be so hasty, the study of Mezger et al. was a field calibration which means there are most likely several factors that induce stress or otherwise influence the Na+ incorporation compared to a culture study with rather stable conditions. The different results between the field and the culture studies implies to me that there are other controlling factors than salinity alone. The results from Mezger et al. could also be region-specific or relate to the variation in salinity within one transect as the authors mentioned. But this needs to be discussed in more detail as mentioned in main point 2).*

We considered your point in the discussion (starting at L495), because Na/Ca values of specimens from the Red See seem to be influenced by many co-varying factors, which could indeed not explained just by temperature or salinity changes.

*L382-384 As said in main point 1), your intra-test variations are similar between the salinity and temperature experiments. Please delete this sentence.*

We deleted this accordingly.

*L433-435 As mentioned in the main point 4), you state in L140-143 that you have only used sediments from sites with $\Delta[CO_3^{2-}] > 30$ μmol/kg, which is higher than the critical calcite saturation level. So, how do you know that dissolution did not affect your Na/Ca ratios? Did you check those samples for any signs of dissolution/diagenesis (visually or by SEM)?*

We changed this statement, as we agree and see the need to additionally present the $\Delta[CO_3^{2-}]$ values for each sampling location to examine in detail a possible effect of calcite dissolution on foraminiferal Na/Ca. Please find the additional information below main point 4). And we checked all foraminiferal samples visually by a binocular microscope, only selecting intact, unbroken and unaltered tests.

*L435-437 I think what you mean here, is that your surface sediment samples fit nicely to your calibration curve, but the sediment samples themselves form a cloud if you look at Figure 6. Therefore, these data do not show any trend or even a slope as you state here. Please rephrase this sentence. I also think that you should reference Figure 5 here as well, as this figure shows a good comparison of your culture data and the sediment samples including their similar variability.*

We agree that we used the wrong word here, as we see no obvious trend in Na/Ca results of surface sediments and changed it accordingly (L332).

*L449-451 Can you please give examples of which kind of biogenic calcite is meant here? A specific species or a species group? And was this relationship found a specific (regional) setting?*

We actually specified biogenic calcite in this respect, but the authors did not mention where the natural calcites, used in their study, derive from. But we decided to delete this paragraph completely as it does not support the discussion. Ishikawa and Ichikuni (1984) did not report about the relationship of Na and salinity in foraminifera.

> Deleted: Further, Ishikawa and Ichikuni (1984) describe a close positive relationship between $Na^+$ and salinity in natural calcites (e.g. limestones, travertines, marine invertebrates as barnacles or oyster shells), albeit this relationship only holds for the low salinity range of 0−10. Data are scattered above a salinity of 10, which the authors explain for marine invertebrates by e.g. different physiological processes or various incorporation mechanisms of Na into the organic matrix or the inclusion of seawater in the cell (Ishikawa and Ichikuni, 1984).

*L461-463 Can you please briefly explain what you mean by the biological control on Na incorporation is smaller than for Mg? Or delete the last part of this sentence as you compare and discuss $D_{Na}$ and $D_{Mg}$ in L468-473.*

We restructured this section and compared the partition coefficients between Mg and Na accordingly (L554-556).

*L464-465 Can you please explain this in more detail? Up to this point, you always said that the Na/Ca values at salinity 44 are low because of the precipitation of gametogenic calcite. So, what other biological aspects can influence the Na incorporation?*
*L464-467 You said that you only found gametogenic calcite at salinity 44 and that such high salinities could impact the growth of the test and therefore, the Na incorporation. So, what does that mean for your highest salinity experiment at S = 45? Do you see gametogenic calcite in those samples and if not, do you have an idea what makes S = 44 so special? Or why do the samples at S = 45 fit to your calibration and those at S = 44 not, if high salinity has an impact on Na/Ca?*

We deleted this sentence in L464-467 and additionally added the information to section 4.1 to avoid repetition about gametogenetic calcite and possible explanations for varying chamber formation at different salinities. We answered your questions as follows:

> L406-414: At the end of their lifecycle foraminifera precipitate gametogenic calcite below their averaged living depth and commonly cooler waters, which should result in calcite with lower Mg/Ca values (Nürnberg et al., 1996; Erez, 2003; Sadekov et al., 2005, Rebotim et al., 2017). In the culture experiments, the rate of gametogenesis was significantly influenced by salinity. As shown in different culture experiments of Bijma et al. (1990b), around 40 % of foraminifera successfully underwent gametogenesis at salinities between 41 and 44, but the reproduction frequency decreases ~10-fold above a salinity of 45. However, Nürnberg et al. (1996) already noted that GAM calcite, when secreted at the same temperatures, is enriched in Mg relative to ontogenetic calcite, because their biomineralization mechanisms could be fundamentally different (Bentov and Erez, 2006; Hamilton et al., 2008). Hamilton et al. (2008) proposed different calcification processes in the microenvironment of foraminifera prior to the gamete formation, compared to the calcite precipitation of ontogenetic calcite (see Sect. 4.4).

*L488-490 What about all the other possible controlling factors? In Section 4.3, you discuss that the cleaning procedure, the photosynthetic rate of symbionts, thermal stress and potentially dissolution can impact the Na/Ca values. Can you rule out some of the factors? If so, please mention this in Section 4.3.*

We restructured the entire section 4.3 and included other possible influencing factors on Na incorporation into foraminiferal calcite, as you suggested in main point 2 (e.g. function of Na in foraminiferal cells and different calcification pathways), but also discussed them in more detail (L513-516).

*L493-494 the same as in L433-435.*

We changed these sentences and adjusted them in respect to the calcite saturation state at respective sampling locations of surface sediment samples (L345-347, L615-616).

*Table 1 Is the salinity and temperature record of the World Ocean Atlas (2013) that imprecise that your values have an error of ± 30 m?*

We clarified this sentence accordingly. Temperature and salinity values are presented at 30 m water depth for respective sampling locations. The error ± indicates the change of these parameters between 0-60 m water depths, expressed as the standard error of the mean. This range implies the habitat migration range of *T. sacculifer* (L905-907).

*Please include the $\Delta[CO_3^{2-}]$ value for each station, as mentioned in main point 4).*
Added to Table 1 (L903).

*Figure 7 You show 3 different regression lines for T. sacculifer, which suggests that there could be other influences on Na/Ca than only salinity, in my opinion. You give several potential explanations in Section 4.3, but could you please explore the potential (unknown) impact of biological processes on Na/Ca as mentioned in main point 2)? What makes your calibration the most reliable one? Consider your big advantage of culture and surface sediment samples.*

We agree and therefore changed it accordingly in the manuscript.

> L1031-1033, Figure 7: Different regression lines for the same species, e.g. *T. sacculifer*, demonstrate various potential factors, and not only salinity, controlling Na incorporation in foraminiferal calcite (discussed in section 4.3). However, our results of culture experiments correspond to fossil samples of surface sediments from two regions suggesting this regression is the most reliable for paleo-reconstructions.

*Table B1 I am confused. Why does this table give different values and uncertainties for the individual measurements than Tables 4 and 5. Not all values are the same as listed in Tables 4 and 5. Does this result in slightly different intra- and inter-test variabilities?*

This was simply due to a converting mistake from our program and thank you spotting this mistake. In this "old" Table we first presented only the standard deviation, but changed it to the standard error of the mean to compare our results with previous studies.

**Style and language:**

*1) Please check your grammar and word order in your sentences. It is sometimes hard to follow what you mean. And please try to avoid nested sentences, they are confusing, e.g. L64-67. It is better to split those sentences into two to be more clear. For more examples, see also minor points below.*

We have tried to take this into account when rewriting the manuscript

*2) The discussion, especially Sections 4.3 and 4.4, needs to be better structured. As mentioned before, the arguments should be linked better. Additionally, the authors jump back and forth in this section which makes it hard to follow each line of thought.*

We restructured and rephrased section 4.3 and 4.4 completely.

We accepted all suggestions below and adjusted those accordingly to the revised manuscript:

*L40-41 In the entire manuscript, you cite the references in chronological order but here alphabetically. Please change this to be consistent.*
*L58-60 better to say "The high spatial resolution of the microprobe technique..."*
*L64-67 As mentioned above, please avoid such nested sentences and make 2 sentences here as this sentence is confusing and somehow reads more like a list.* Now L65-70.
*L82-84 better: "Prior to gametogenesis at the end of its life cycle, a process related to reproduction, T. sacculifer...".* Adjusted.

*L118-120 I would delete the part of the sentence about the Subtropical Underwater mass as I do not think you need this information, since T. sacculifer lived in the Caribbean Surface Water mass.*

Here we would kindly disagree, because from our point of view, it is important to also mention parameter conditions of the Subtropical Underwater mass, because foraminifera descend in the water column prior to gemetogenesis below their averaged living depth (e.g. in Rebotim et al., 2017). We added this information to L406.

*L120-122 I think you mean: "...entering the NE Caribbean through the Lesser Antilles."*
Changed.

*L130-131 If you reference more than one figure, please use plural, e.g. Figures 1b-d. This is valid for all your references to your figures (e.g., L231, L239, L253, L267, L269, L270, L275, L280, L287,...).*
Changed.

*L138-139 Please reference Figures 1b and c here which show salinity at 30 m and with increasing depth as you say in this sentence.*
Changed.

*L152-153 If I am not mistaken the abbreviation is "EMPA".*
The abbreviation we prefer, and is clearly defined, is EMP.

*L154-155 "...were covered by a 20 nm carbon coating..." better to say "...were impregnated with carbon coating..."?*
We respectfully disagree as the coating is only on the top surface.

*L161-162 Please use either "shell" or "test" to be consistent. You mostly use "test" in the manuscript.*
Changed consistently to "test".

*L170-171 better to say "Absolute Na/Ca ratios were calculated using..." instead of "quantified"*
Changed.

*L170-171 Please delete Iback and Itotal as the abbreviations are not needed, you do not use them again in the entire manuscript.*
Changed.

*L171-172 I know what you mean but it sounds wrong to compare intensities with concentrations. I think it is better to state: "The obtained net intensities were converted to concentrations and corrected to known concentrations (in wt %) of referenced materials."*
Changed.

*L198-199 "The samples were thoroughly rinsed to remove the chemicals..."*
Changed.

*L201-203 Delete "...and dissolved...", it does not really fit with the rest of the sentence.*
Changed.

*L205-206 „The machine is..." device?* Right.
Changed.

*L206-208 "...these wavelengths were 70.60 nm for Ca, 279.55 nm for Mg, and 589.59 nm for Na."*

Changed.

*L208-209 Delete one "with".*
Changed.

*L209-210 Delete "Hence" at the beginning of sentence.*
Changed.

*L211-212 You give only one values, so please use the singular. Also, provide the value together with its uncertainty using "±". Please use the past tense in this sentence. Please be also consistent in how you report units. Here you use "mmol/mol" whereas you mainly use "mmol mol-1" in the manuscript. This is also valid for the tables in which the units are reported in one or the other version. You always use "mmol/mol" for your plots, but could you also change the unit to "mmol mol-1" to be consistent throughout the entire manuscript?*
We changed all points mentioned here accordingly.

We changed all to mmol mol$^{-1}$.

*L227-228 Space between 13.2 and %. Please be consistent, whether you use a space between numbers and the % symbol or not.*
Changed.

*L230-231 I do not think that Figure 4 represents the low variability of the experiment at 26.5°C and salinity 36. I guess, it would be better to cite Section 2.3 here where you describe the natural habitats.*
Changed. Now only mentioned in the section about temperature experiments (L291-295).

*L237-239 Reference Figures 3a and 4.*
Changed.

*L251-252 Delete "...within the same temperature interval.". The interval mentioned in the sentence before is your entire range of T experiments, so this bit of text is not needed, on the contrary it is confusing.*
Changed.

*L257-258 This information is already said in L228-230, however, fits better here in the T experiment section. Hence, delete sentence in L228-230.*
Changed.

*L259-260 Reference Figure 3b here.*
Changed.

*L272-273 You are always very precise with reporting differences between Na/Ca values, so please say here "± 0.16 mmol/mol" or say "about 0.2 mmol/mol". Further move "noticeable" to after "increase of about 0.2 mmol/mol".*
Changed.

*L278-280 better to say "...as the inter-test variability indicates of specimens from…"*
Changed.

*L290-291 Please decide whether you use "shell" or "test" in your manuscript. I think, it is better to stick to one expression to be consistent. You mainly use the term "test".*

Agreed on using the term "test" and changed it.

*L306-308 I would rephrase this sentence a little bit: "Transferred to our study, higher Na/Ca intra-test variations may occur between maps within single wall cross sections of each salinity experiment compared to the temperature experiments (Table B1) due to higher stress levels of (individual) foraminifera in culture."*
Changed (L555-557).

*L315-316 Delete "more or less". I think it is not important here how much GAM calcite is produced, only that it is precipitated.*
Changed.

*L318-320 Delete "probably" and say instead: "...their biomineralisation mechanisms could be fundamentally*

*different.”*
Changed (L572).

*L320-321 Insert “at salinity 44” at the end of the sentence to make it clear. Also “Figures 3, 4”.*
Changed.

*L327-328 Rephrase this nested, confusing sentence, e.g.: “...dark layers in benthic foraminiferal calcite, most likely comprising the primary organic sheet, which in turn is located between the primary and secondary calcite.”*

We rephrased this sentence (L388-390).

*L339-341 In your plots and other parts of the manuscript, the word “versus” is spelled out and not written in italics. Please be consistent throughout the manuscript. Further, maybe insert “their” between “for” and “T. sacculifer” to make clear that only Allen et al.’s study is meant here.*
Changed.

*L341-342 Reference Figure 7 at the end of the sentence.*
Changed.

*L342-344 Also reference Figure 7, best right after “Allen et al. (2016)” and reference Section 4.3 at the end of the sentence as you discuss in detail the difference between those previous studies in that section.*
Changed.

*L351-353 As mentioned before, please use either “test” or “shell” in the manuscript.*
Changed.

*L360-362 Please spell “YD” out as it is only used this once in the entire manuscript.*
We deleted this term completely when rephrasing this paragraph.

*L362-363 “...a salinity decrease of ~2-4 salinity units.” I am not sure if you need the “salinity” in front of “units” as you already said it is about salinity.*
We changed the whole paragraph (L464-467).

*L364-366 Please make 2 sentences out of this nested sentence as it is confusing. You also used “δ18Oivf-sw” before instead of “δ18Oivc-sw”. Please stick to one term.*

We shortened this paragraph (L464-467).

*L366-367 Desirable instead of “helpful”?*
We rephrased this paragraph, and therefore deleted “helpful or desirable” (L466-467).

*L367-369 “Instead, the addition of Na/Ca…”*
Changed.

*L374-376 “In contrast, the Red Sea study…”*
Changed (L472).

*L377-379 As mentioned before, please use either “test” or “shell” in the manuscript.*
Changed.

*L381-382 Reference Figure 3b here.*
Changed.

*L387-438 As said in point 2) (Style and Language) please re-structure this section. I think it might be easier to list all potential factors first and then explain them in detail, as it is not clear until L392-393 what you mean with the first sentences. Please make separate paragraphs when you start discussing another potential controlling factor.*

Now we listed all potential controlling factors one by one as suggested.

*In L394-400 you discuss that several carbonate system parameter could influence the Na/Ca values and then in L406-409 you talk again about the influence of carbonate system parameters on Na/Ca ratios. Please combine these parts.*
*You also need to link your sentences better in this section, as sometimes (see listed below) it is not clear what you mean.*

We agree and clearly separated each factor per paragraph (Section 4.3, starting at Line 469).

*L390-392 Better to move "for the overlapping salinity intervals of 30 to 38" to the end of the sentence and insert the reference to Figure 7 right after "Allen et al. (2016)".*
*Start a new paragraph after "...further study." in L394 as you then start discussing the influence of the carbonate system parameter on Na/Ca ratios and not the impact of cleaning procedures.*
This has been adjusted and clearly separated in the revised manuscript (L492-493).

*L395-398 What are those "many co-varying factors"? Please give some example. Is it pH, DIC? Say this here and you have a nice link to your next sentence.*
We added the information respectively.

> L495-496: Mezger et al. (2016) related the extremely high Na/Ca ratios in their study to many co-varying factors (e.g. carbonate chemistry, salinity, temperature) in the Red Sea, which may differ from open ocean conditions.

*L398-400 Please insert "in the Red Sea" between "...foraminiferal Na/Ca ratios" and "(Mezger et al., 2016)" to make clear this still relates to the study in the Red Sea. Also start a new paragraph after this sentence as you go on about Na heterogeneity in tests.*
Adjusted.

*L403-405 Please rephrase this sentence, it sounds odd. E.g., "Previous studies possibly provide a mixed Na/Ca signal because chamber wall calcite and spines were analysed together for elemental composition of foraminifera tests."*
Changed (L456-458).

*L406-407 Delete "also" after "...confidence interval)". Further, link this sentence better to the next as you jump from increasing $\Delta[CO_3^{2-}]$, DIC and pH to element partitioning and calcification rates. Hence, it is best to combine this paragraph (L406-409) with the paragraph in L394-400 where you actually say that an increase in DIC, pH, $\Delta[CO_3^{2-}]$ could cause higher precipitation rates.*
We combined these sentences and structured it clearly.

*L410-412 There is also the link between the 2 sentences missing. Here is a suggestion: "...micro-environment of foraminiferal tests, driven by photosynthetic activity of symbionts. This activity depends on nutrient concentration and light intensities and thus can actively change ...".*
We changed this accordingly (L506-507).

*L412-414 This sentence reads also weird. Maybe: "Higher photosynthetic rates influence test geochemistry which has been demonstrated by(?) boron...species, such as O. universa...".*
This has been adjusted (L506-509).

*L414-417 This sentence needs to be restructured. The part "although within...error of the mean" should go to the end of the sentence (between "specimens" and "(this study,..)"). Also, put a comma after T. sacculifer. However, I do not really understand what you mean to say here. Do you suspect different photosynthetic rates to explain that offset between culture experiments and surface sediments/in situ grown chambers? Then start a new paragraph after this sentence, you then discuss the influence of light intensities. Or better link it to this paragraph (L410-417).*

Since we rephrased the whole section 4.3, we deleted the previous sentence in L414-417 and compared and better linked it to calcite dissolution effects in the last paragraph of section 4.3, starting at L528.

*L420-421 If the discussion of the light intensities belongs to the part before, start now a new paragraph as you talk about thermal stress. If the part about the intensities is meant to go with the following paragraph (L420-425) you need to better link these ideas, as I do not see a connection between light intensities and thermal stress. Also, insert this entire sentence "We speculate . . . culture experiments." after the next sentence "Previous studies…(Edgar et al., 2013)." and maybe connect it with "Hence, we speculate…". Then maybe start the next sentence "This warming/thermal stress impacts the…".*

This has been clarified now and further, we deleted our previous mentioned hypothesis about thermal stress, because as we could demonstrate with our culture temperature experiments, there is no significant trend of Na/Ca in relation to increasing temperatures. And Van Dam et al. (2012) also report that only temperature changes above >30 °C could impact the photosynthetic activity, resulting in the possible loss of chlorophyll a, of benthic foraminifera. This is beyond our temperature experiments, but could be checked in future studies.

*L426-428 Delete "for example" and change "or" into "…Cd/Ca, and Ba/Ca…".* Adjusted.

*L428-430 Use a comma instead of semicolon to cite more than 1 publication of the same author (here, Regenberg). I would change this sentence a little bit "… report a decline in planktonic foraminiferal Mg/Ca below ~2500-3000 m…"*
Changed.

*L431-438 Here, you do not need a new paragraph as you continue to discuss the effect of calcite dissolution. Again, use comma for "Regenberg et al. 2009, 2014)".* Adjusted.

*L432-433 This is no full sentence, there is the main clause missing or do not start this sentence with "as". Further, move "("± 0.25 mmol/mol)" to the end of sentence, just before the reference to your figure and table.*
We rephrased the sentence (L345-347).

*L433-435 better: "Although the calcite saturation state is quite different in bottom waters, we suspect…" Full stop missing here.* Adjusted.

*L441-444 It is better to split this sentence into 2, using "alternatively" as the beginning of the second sentence. Better: "In this case, two…", "…accompanied by anions, such as Cl-…".* Adjusted.

*L444-447 Put "so far" after "…experiments of aragonite, so far, in which…".* Adjusted.

*L447-449 missing commas "…biogenic calcite, such as foraminifera…~5 mmol/mol), and aragonite, such as corals…" Further, link this sentence better to the next, there seems to be no connection between what you just said and the next sentence.*
We restructured the entire section and now this is better linked to previous thoughts (L590).

*L451-453 "…with increasing(?) growth rates of inorganic calcite…" Do you mean it that way?*
 Yes and changed.

*L454-456 It is better to give the $D_{Na}$ value of foraminifera first before you compare it to the inorganic value. Or start your next sentence saying that the inorganic $D_{Na}$ value is higher than your foraminifera equivalent and then give your calculated value.*

We shifted the paragraph about partition coefficients to the beginning of section 4.4, reporting about $D_{Na}$ in planktic foraminifera and then comparing those to inorganic calcite experiments (now L546-556).

*L456-457 better: "…, using Na/Ca$_{seawater}$…".* Adjusted.

*L457-458 Who do you mean by "their"? Delaney et al. or Busenberg and Plummer? Please specify. Also: "…and of this study."*

In this case we meant Delaney et al. and changed it accordingly (L548) and also changed "this study" to "our study".

*L458-460 Delete "listed in the following" and maybe put a colon after each $D_{Na}$ value instead of a comma, like this: "$D_{Na} = 0.1x10-3$: T. sacculifer, G. ruber…"*
*L460-461 Rephrase this sentence to: "The highest $D_{Na}$ of 0.18-0.25x10-3 were found in the field calibration study…" or "Mezger et al. (2016) found the highest $D_{Na}$ of 0.18-0.25x10-3 in their field calibration study…".*
This has been adjusted to the manuscript.

*L463-467 Can you please restructure these 3 sentences? I was confused about what you mean until you mention the influence of the gametogenic calcite in the last sentence (L466-467). Maybe start with this last sentence and then that you deduced from the presence of the gametogenic calcite that there has to be a biological control on the Na incorporation.*
As previously mentioned, we omitted these sentences here and shifted it to section 4.1, reporting in detail about gametogenic calcite (see L 406 …).

*L474 Maybe better to use "requires" instead of "affords".* Adjusted.

*L477-479 There is a verb missing in the last part of that sentence. Please rephrase, e.g., "…in seawater are only affected by temperature changes to a limited extent, and hence temperature has minimal influence(?) on Na incorporation…" if I understand your last part correctly.*
This has been changed.

*L484-485 Delete opening parenthesis before 2016: "(Allen et al. 2016)".* Adjusted.

*L485-487 "In our experiments, we…" I also think your intra- and inter-test variability is 2-16%.* Adjusted.

*L520-719 References: Maybe this is an issue of converting your file into pdf, but could you please use indentation to separate individual references? The way it is at the moment is hard to read.*
This has been changed.

*Table 1 Can you please change mmol/mol into mmol mol-1 to be consistent?* Adjusted.

*Table 3 Please rephrase the sentence "Chambers grown in situ…" because it sounds like that the chambers that grew in situ are different to those which grew in the natural environment. But they are the same, that is at least what you say in the manuscript text.*
Indeed, this was confusing and we changed it (L923-924).

*Figure 1 Can you please insert a line at 30 m in your plots 1c, 1d to make a direct comparison easier between the hydrography plots and surface maps.*

We included a horizontal line, indicating the average depth habitat of *T. sacculifer*.

[Figure]

*Figure 3 caption, better: "(19.5 – 29.5°C)" or "(19.5°C – 29.5°C)" Further, please mention here that the experiment at salinity 44 is excluded because of the precipitation of GAM calcite.*
Adjusted.

> L973-975: The black colored box shows results of the 44-salinity experiment, in which cultured foraminifera underwent gametogenesis (Nürnberg et al., 1996) and where therefore excluded from the regression in Figure 4.

*Figure 4 "Related raw data of salinity experiments…" better to say "Related data of individual measurements of the salinity experiments…" because the data presented in Table 3 are not raw data per se (Raw data would be the measured intensities for each element.) but your final, calculated data.*
*Please say: "Datasets for T = 26.5°C of both temperature experiments (S 33, S 36) were added (Tables 4, 5)."*
We added all points addressed here to the revised manuscript.

*Insert: "Horizontal error bars of samples from the Gulf of Guinea demonstrate…" But why only for those sediments? What about the surface sediment samples from the Caribbean Sea? Is the error as big as the symbol size? If yes, please say so.*

Indeed, horizontal error bars are as small as their symbol sizes and we therefore mentioned this additionally.

*Move your sentence "Data marked by yellow,…" to after you explained those symbols, that is "…Caribbean surface sediments (Table 1)." Those data is not listed in Table 2. Also reference Table 3 for the in situ chambers and Table 1 for the Gulf of Guinea samples.*

We included all points to the manuscript (L990-991).

*Figure 5 Similar to Figure 4: better to say "Related data of individual measurements…" instead of "Related raw data…". Also, "Tables 4 and 5".*

We changed this caption similar to the caption of Figure 4 (L995-1004).

*The same as in Figure 4: "Horizontal error bars of samples from the Gulf of Guinea demonstrate…" Why only for those sediments?*

This was simply a mistake from our side to not mention horizontal error bars for the Caribbean surface sediment samples and therefore added this information (L1002-1003).

*Figure 6 Can you please label the dashed line in your plot with "95% confidence interval" to make it easier to read this plot?*

Here we would say that this is not necessary as it is indicated in the symbol legend, and the figure is less crowded.

*Figures 4-6 This is a very personal thing, but maybe consider to present the numbers on your axes with the same decimal digits, e.g. 8.0; 7.5; 7.0;...*
We changed this to be consistent.

*Figure 7 Please be consistent and use as font style regular instead of italics for "versus", which you otherwise use in the entire manuscript. Further move "versus salinity" in the first sentence to the end of the sentence, as it is confusing, or incorporate "versus salinity" into a new sentence, e.g., delete from the first sentence and then say "Foraminiferal Na/Ca values are shown versus/against salinity."*
*Better "Regression lines are based on the following equations: Wit et al.,..."*
This has been adjusted to the figure caption (L1030-1033).

*The equations are given in R and $R^2$. Is this correct?*
Yes, this is correct. In our case we only used R to directly compare it to the study of Allen et al. (2016), because the authors used R instead of $R^2$.

*Figure 8 Please give the salinity of the other studies (Allen et al., Mezger et al.) in the legend of this plot for a direct comparison.*
This is a very good point and we added the respective salinities to the figure caption (L1038-1041), but also added this information to the main text (L474-477).

*Similar to Figure 7: Move "versus temperature" in the first sentence to the end of this sentence or make a new sentence, e.g., delete from the first sentence and then say "Foraminiferal Na/Ca values are plotted versus/against temperature."*
We changed "versus temperature" to "Comparison of foraminiferal Na/Ca values in response to temperature changes [...]" (L1033-1034).

*Table A1 caption "...listed in Tables 3, 4, and 5."* Adjusted.

*Table B1 caption "...averaged maps (Tables 3, 4, 5)..." Further, please be consistent and use "Intra-test" and "Inter-test" in the header of your table, like you do in the text.* Adjusted.
*It would improve the readability of this table if you insert a header for the salinity and the temperature experiments like you do for the in situ grown chamber and maybe separate the different experiments/data by horizontal lines.*
We added a header for salinity and temperature experiments, respectively.

References used in the author's response

De Nooijer, L. J., Hathorne, E. C., Reichart, G. J., Langer, G., and Bijma, J.: Variability in calcitic Mg/Ca and Sr/Ca ratios in clones of the benthic foraminifer *Ammonia tepida*, Marine Micropaleontology, 107, 32-43, doi:10.1016/j.marmicro.2014.02.002, 2014a.

Edgar, K. M., Bohaty, S. M., Gibbs, S. J., Sexton, P. F., Norris, R. D., and Wilson, P. A.: Symbiont 'bleaching' in planktic foraminifera during the Middle Eocene Climatic Optimum, Geology, 41(1), 15-18, doi:10.1130/G33388.1, 2013.

Elderfield, H., Vautravers, M., and Cooper, M.: The relationship between shell size and Mg/Ca, Sr/Ca, $\delta^{18}$O, and $\delta^{13}$C of species of planktonic foraminifera, Geochem. Geophys. Geosyst., 3(8), doi:10.1029/2001GC000194, 2002.

Regenberg, M., Nürnberg, D., Steph, S., Groeneveld, J., Garbe-Schönberg, D., Tiedemann, R., and Dullo, W. C.: Assessing the effect of dissolution on planktonic foraminiferal Mg/Ca ratios: Evidence from Caribbean core tops, Geochem. Geophys. Geosyst., 7(7), Q07P15, doi:10.1029/2005GC001019, 2006.

Regenberg, M., Regenberg, A., Garbe-Schönberg, D., and Lea, D. W.: Global dissolution effects on planktonic foraminiferal Mg/Ca ratios controlled by the calcite-saturation state of bottom waters, Paleoceanography, 29, 127-142, doi:10.1002/2013PA002492, 2014.

Rebotim, A., Voelker, A. H. L., Jonkers, L., Waniek, J. J., Meggers, H., Schiebel, R., Fraile, I., Schulz, M., and Kucera, M.: Factors controlling the depth habitat of planktonic foraminifera in the subtropical eastern North Atlantic, Biogeosciences, 14, 827-859, https://doi.org/10.5194/bg-14-827-2017, 2017.

Sadekov, A., Eggins, S. M., De Deckker, P., and Kroon, D.: Uncertainties in seawater thermometry deriving from intratest and intertest Mg/Ca variability in *Globigerinoiedes ruber*, Paleoceanography, 23, PA1215, doi:10.1029/2007PA001452, 2008.

Van Dam, J. W., Negri, A. P., Mueller, J. F., Altenburger, R., and Uthicke S.: Additive Pressures of Elevated Sea Surface Temperatures and Herbicides on Symbiont-Bearing Foraminifera, PLoS ONE 7(3), e33900, doi:10.1371 /journal.pone.0033900, 2012.

---

## Author Comment (AC3) · 20 Jul 2018

The comment was uploaded in the form of a supplement:
https://www.biogeosciences-discuss.net/bg-2018-164/bg-2018-164-AC3-supplement.pdf

---

## Author Response (AR1)

Dear Aninda Mazumdar,

We kindly thank you for your effort in reviewing our manuscript. In the following we refer in detail to the three minor issues you finally raised in your last correspondence to us from September 17th, 2018. These comments, which have also been previously addressed by the anonymous reviewers, were carefully considered and answered in detail in the current version of our manuscript. Below you find our point-by-point reply, complemented by the according text passages in the manuscript. We hope that this will find your support, and will help the manuscript to get finally accepted at Biogeosciences. Please let us know in case of further questions.

Yours sincerely,
Jacqueline Bertlich and on behalf of all co-authors

**Answers to the editor's comments:**

A. *I think the effect of alkalinity on surface sediment Na/Ca ratio is quite apparent in case of Caribbean samples. In fact the change in Na/Ca is almost of the same magnitude as in figure-4. This issue may be a little bit more clarified.*

We totally agree that there is a possible effect on foraminiferal Na/Ca values with changes in the carbonate ion concentration and increasing water depth at different locations, especially obvious at Caribbean surface sediment samples below 3.5 to 4 km water depth (Figure 6b). We pointed out this more clearly and as already written in the revised manuscript, the dissolution effect on Na/Ca at deeper water depth definitely needs more study and elaboration due to the limited sample set. Further, Na/Ca values of surface sediments (Fig. 6b) have indeed the same variation/magnitude as in Figure 4, since we used the same values, but just plotted against salinity in Figure 4 and 6a, in comparison to the influence on Na/Ca of changes in the calcite saturation state with increasing water depth.

Line 531-546: For instance, planktonic foraminiferal Mg/Ca values start to decline linearly below the critical $\Delta[CO_3^{2-}]$ threshold of ~21.3 µmol kg$^{-1}$ (Regenberg et al., 2006, 2014, see Sect. 2.3 for details). Evidently, we cannot see the same trend for Na/Ca$_{T. sacculifer}$ values from Caribbean surface sediments, albeit Na/Ca seems to decline at 3.5 to 4 km water depth, below a $\Delta[CO_3^{2-}]$ level of <13 µmol kg$^{-1}$ (Figure 6b). In the Gulf of Guinea, the critical $\Delta[CO_3^{2-}]$ threshold value is at even shallower water depth (~2.5 km), but no significant Na/Ca change with water depth is obvious, matching the Caribbean values (Table 1, Figure 6b). Therefore, Na/Ca seems to be less affected by dissolution than Mg/Ca, which could be linked to different incorporation mechanisms of both elements into foraminiferal calcite (see Sect. 4.4). The issue of Na/Ca removal due to calcite dissolution at greater water depth, nonetheless, needs further investigation, because of the limited sample set of our study.

Although the calcite saturation state is quite different in bottom waters and differs regionally in depth

(Brown and Elderfield, 1996; Dekens et al., 2002; Regenberg et al., 2006, 2014), we suspect that foraminiferal Na/Ca, similar to foraminiferal Mg/Ca dissolution, is not changing above the critical calcite saturation state of $>\sim21$ µmol kg$^{-1}$ in the Caribbean and the Gulf of Guinea at least over the last ~3000 years (ages of Caribbean and Gulf of Guinea surface sediments; Regenberg et al., 2009). Although lowest Na/Ca values of surface sediments are possibly affected by calcite dissolution at respective water depths below 3.5 km, all foraminiferal Na/Ca values match the 95% confidence interval of culture experiments, not affected by dissolution (Fig. 4, 6a). This confirms the robustness of Na/Ca as paleo-salinity proxy over time.

L617-623: We observed no significant calcite dissolution effect on Na/Ca in foraminiferal calcite with changes in the calcite saturation state above 2.5 km water depth in the Gulf of Guinea and 3.5 to 4 km in the Caribbean, while Na/Ca values decline below. However, Na/Ca is not changing above the critical $\Delta[CO_3^{2-}]$ threshold of ~21 µmol kg$^{-1}$, established for the onset of Mg/Ca dissolution in planktonic foraminifera (Regenberg et al., 2006, 2014). Further, Na/Ca values of surface sediment samples lie within the 95 % confidence interval of our salinity culture experiments, which are unaffected by calcite dissolution. Therefore, our new data support that foraminiferal Na/Ca potentially serves as a direct and reliable proxy for significant changes in ocean salinities, e.g. exceeding 0.5 salinity units.

B. *The dispersion (error bar) in the Na/Ca data from surface sediments are quite large (Fig. 6A). Authors may suggest how much error in paleosalinity measurement is expected using T. succulifer based Na/Ca ratio measurements? and how this will be advantageous to Mg/Ca based measurement currently used since the author has suggested multi-proxy approach at the end.*

Both reviewers addressed previously the same points that the Na/Ca variability within specimens is quite high in respect to paleo-salinity reconstructions. Therefore we calculated additionally, based on our culture experiments and surface sediment samples, the smallest salinity range we could reconstruct, while considering the standard error of culture experiments and surface sediments. As written in the manuscript (L462-463), it is necessary to analyze at least 20 specimens to get a standard error of 0.18-0.36 mmol/mol, which in fact would reflect a salinity change of 0.5 to 1 at respective standard errors. Here we would kindly refer to the section 4.2, starting at line 441 onwards.

Further, the advantage of a combined multi-proxy approach in combination with Mg/Ca is simply linked to the fact, that Mg/Ca in combination with $\delta^{18}O$ just give indirect estimations and no absolute salinity values. We clarified this accordingly in the revised manuscript.

L464-468: With a sensitivity of 0.5 salinity units, reconstructions of paleo-river discharge (e.g. see Weldeab et al., 2007) as well as glacial meltwater discharge (e.g. in Flower et al., 2004) should become possible, independent from stable oxygen isotope approaches. Therefore, the application of Na/Ca$_{foram}$ in multi-proxy approaches, as already suggested in Vetter et al. (2017) to combine e.g. foraminiferal Mg/Ca$-\delta^{18}O$ with Ba/Ca, could infer more confidence of such reconstructions by providing absolute salinity values instead of previous indirect estimations.

*C. In a routine measurement, an aliquot of multiple well preserved shells are expected to be used for Na/Ca measurement, which I believe represent an homogenized number than the point specific measurements using EPMA which is bound to show a lot of dispersion. Author may kindly clarify how an EPMA based calibration line would work for aliquot based measurement?*

As already written in lines 458-462, we could show that Na/Ca values of fossil foraminifera (bulk/aliquot samples of around 20 to 30 specimen, homogenized and measured by wet-chemical analysis – ICP-OES) from respective surface sediments reflect the same inter-test variability and have the same standard error as single map analysis by the electron microprobe. This proofs the reliability of our EPMA based calibration line. For our Na/Ca to salinity calibration we averaged Na/Ca values of at least 3 to 5 specimens, so it is comparable to the application routine for paleo-reconstructions. Electron microprobe results just additionally provide insights into the intra-test variability (which is the higher "dispersion" compared to bulk samples) and element distribution patterns with a higher spatial resolution than observed with bulk analysis.